# Mean Field Residual Networks: On the Edge of Chaos

**Greg Yang**[*]
Microsoft Research AI
gregyang@microsoft.com

**Samuel S. Schoenholz**
Google Brain
schsam@google.com

## Abstract

We study randomly initialized residual networks using mean field theory and the theory of difference equations. Classical feedforward neural networks, such as those with tanh activations, exhibit exponential behavior on the average when propagating inputs forward or gradients backward. The exponential forward dynamics causes rapid collapsing of the input space geometry, while the exponential backward dynamics causes drastic vanishing or exploding gradients. We show, in contrast, that by adding skip connections, the network will, depending on the nonlinearity, adopt subexponential forward and backward dynamics, and in many cases in fact polynomial. The exponents of these polynomials are obtained through analytic methods and proved and verified empirically to be correct. In terms of the "edge of chaos" hypothesis, these subexponential and polynomial laws allow residual networks to "hover over the boundary between stability and chaos," thus preserving the geometry of the input space and the gradient information flow. In our experiments, for each activation function we study here, we initialize residual networks with different hyperparameters and train them on MNIST. Remarkably, our *initialization time* theory can accurately predict *test time* performance of these networks, by tracking either the expected amount of gradient explosion or the expected squared distance between the images of two input vectors. Importantly, we show, theoretically as well as empirically, that common initializations such as the Xavier or the He schemes are not optimal for residual networks, because *the optimal initialization variances depend on the depth*. Finally, we have made mathematical contributions by deriving several new identities for the kernels of powers of ReLU functions by relating them to the zeroth Bessel function of the second kind.

## 1 Introduction

Previous works [9, 3, 11] have shown that randomly initialized neural networks exhibit a spectrum of behavior with depth, from stable to chaotic, which depends on the variance of the initializations: the cosine distance of two input vectors converges exponentially fast with depth to a fixed point in [0, 1]; if this fixed point is 1, then the behavior is stable; if this fixed point is 0, then the behavior is chaotic. It has been argued in many prior works [1, 9] that effective computation can only be supported by a dynamical behavior that is on **the edge of chaos**. Too much stability prevents the neural network from telling apart two different inputs. While some chaotic behavior can increase the expressivity of a network, too much chaos makes the neural network think two similar inputs are very different. At the same time, the same initialization variances also control how far gradient information can be propagated through the network; the networks with chaotic forward dynamics will tend to suffer from exploding gradients, while networks with stable forward dynamics will tend to suffer from vanishing gradients.

---

[*]Work done while at Harvard University

These works have focused on vanilla (fully connected) feedforward networks. Here we consider residual networks [6, 7] (with fully-connected layers and without batchnorm), which are a family of recently proposed neural network architectures that has achieved state-of-the-art performance on image recognition tasks, beating all other approaches by a large margin. The main innovation of this family of architectures is the addition of a passthrough (identity) connection from the previous layer to the next, such that the usual nonlinearity computes the "residual" between the next-layer activation and the previous-layer activation.

In this work, we seek to characterize randomly initialized residual networks. One of our main results is that random residual networks for many nonlinearities such as $\tanh$ **live on the edge of chaos**, in that the cosine distance of two input vectors will converge to a fixed point at a polynomial rate, rather than an exponential rate, as with vanilla tanh networks. Thus a typical residual network will slowly cross the stable-chaotic boundary with depth, hovering around this boundary for many layers. In addition, for most of the nonlinearities considered here, the mean field estimate of the gradient grows subexponentially with depth. In fact, for $\alpha$-ReLU, the $\alpha$th-power of ReLU, for $\alpha < 1$, the gradient grows only polynomially. These theoretical results provide some theoretical justification for why residual networks work so well in practice. In our experiments, we are also able to predict surprisingly well the relative performances of *trained* residual networks based only on their initialization hyperparameters, in a variety of settings. In particular, we find that the quality of initialization for tanh resnets is determined by *trainability* (how much gradient explosion on average) while that for ($\alpha$-)ReLU resnets is determined by expressivity (how far can two different input vectors be pulled apart) (see Section 6). To the best of our knowledge, this is the first time that a quantity other than gradient explosion/vanishing has been found to control the quality of initialization. We establish theoretically and empirically that the best initialization variances for residual networks depend on the depth of the network (contrary to the feedforward case [11]), so that **common initialization schemes like Xavier [4] or He [5] cannot be optimal**. In fact, even the rationale of He initialization is incorrect for ReLU residual networks because it tries to control gradient dynamics rather than expressivity. However we want to emphasize that we study a simplified model of residual networks in this work, with no batchnorm or convolutional layers, so that these results are not necessarily indicative of the MSRA residual network used in practice [6].

In the body of this paper, we give account of general intuition and/or proof strategy when appropriate for our theoretical results, but we relegate all formal statements and proofs to the appendix.

## 2 Background

Consider a vanilla feedforward neural network of $L$ layers, with each layer $l$ having $N^{(l)}$ neurons; here layer 0 is the input layer. For the ease of presentation we assume all hidden layer widths are the same $N^{(l)} = N$ for all $l > 0$. Let $x^{(0)} = (x_1^{(0)}, \ldots, x_{N^{(0)}}^{(0)})$ be the input vector to the network, and let $x^{(l)}$ for $l > 0$ be the activation of layer $l$. Then a neural network is given by the equations

$$x_i^{(l)} = \phi(h_i^{(l)}), \qquad\qquad h_i^{(l)} = \sum_{j=1}^{N} w_{ij}^{(l)} x_j^{(l-1)} + b_i^{(l)}$$

where (i) $h^{(l)}$ is the pre-activation at layer $l$, (ii) $w^{(l)}$ is the weight matrix, (iii) $b^{(l)}$ is the bias vector, and (iv) $\phi$ is a nonlinearity, for example $\tanh$ or ReLU, which is applied coordinatewise to its input.

To lighten up notation, we suppress the explicit layer numbers $l$ and write

$$x_i = \phi(h_i), \qquad\qquad h_i = \sum_j w_{ij} \underline{x}_j + b_i$$

where $\bullet$ implicitly denotes $\bullet^{(l)}$, and $\underline{\bullet}$ denotes $\bullet^{(l-1)}$ (and analogously, $\overline{\bullet}$ denotes $\bullet^{(l+1)}$).

A series of papers [9, 10, 11] investigated the "average behavior" of random neural networks sampled via $w_{ij}^{(l)} \sim \mathcal{N}(0, \sigma_w^2/N), b_i^{(l)} \sim \mathcal{N}(0, \sigma_b^2)$, for fixed parameters $\sigma_w$ and $\sigma_b$, independent of $l$. Consider the expectation of $\frac{1}{N} \sum_{i=1}^{N} x_i^2$, the normalized squared length of $x$, over the sampling of $w$ and $b$. Poole et al. [9] showed that this quantity converges to a fixed point exponentially fast for sigmoid nonlinearities. Now suppose we propagate two different vectors $x^{(0)}$ and $(x^{(0)})'$ through the

network. Poole et al. [9] also showed that the expectation of the normalized dot product $\frac{1}{N}\sum_{i=1}^{N} x_i x_i'$ converges exponentially fast to a fixed point. The ratio between the normalized squared length and the normalized dot product is the cosine distance between $x$ and $x'$. Thus these two exponential convergence results show that the cosine distance converges exponentially fast to a fixed point as well. Intuitively, this means that a vanilla feedforward network "forgets" the geometry of the input space "very quickly," after only a few layers.

In addition, Schoenholz et al. [11], under certain independence assumptions, showed that the expected normalized squared norm of the gradient also vanishes or explodes in an exponential fashion with depth, with the "half-life" controlled by $\sigma_w$ and $\sigma_b$. They verified that this theoretical "half-life" correlates in practice with the maximal number of layers that are admissible to good performance.

At the same time, Daniely et al. [3] published work of similar nature, but phrased in the language of reproducing kernel Hilbert spaces, and provided high probability estimates that are meaningful for the case when the width $N$ is finite and the depth is logarithmic in $N$. However, they essentially fixed the variance parameters $\sigma_\bullet$, and furthermore, their framework (for example the notion of a "skeleton") does not immediately generalize to the residual network case.

In this work, we show that residual networks have very different dynamics from vanilla feedforward networks. In most cases, the cosine distance convergence rate and the gradient growth rate are subexponential in a residual network, and in most cases, these rates may be polynomial.

## 3 Preliminaries

Residual networks were first introduced by [6] and later refined by [7], and they are now commonplace among deployed neural systems. The key innovation there is the addition of a shortcut connection from the previous layer to the next. We define the following idealized architectures for ease of analysis. Note that we only consider fully-connected affine layers instead of convolutional layers. A **reduced residual network (RRN)** has the recurrence

$$x_i = \phi(h_i) + \underline{x}, \qquad\qquad h_i = \sum_j w_{ij}\underline{x}_j + b_i.$$

A **(full) residual network (FRN)** in addition has an affine connection given by weights $v$ and biases $a$ from the nonlinearity $\phi(h)$ to the next layer:

$$x_i = \sum_j v_{ij}\phi(h_j) + \underline{x}_i + a_i, \qquad\qquad h_i = \sum_j w_{ij}\underline{x}_j + b_i$$

We are interested in the "average behavior" of these network when the weights and biases, $w_{ij}^{(l)}, b_i^{(l)}, v_{ij}^{(l)}$, and $a_i^{(l)}$ are sampled i.i.d. from Gaussian distributions resp. with standard deviations $\sigma_w, \sigma_b, \sigma_v$, and $\sigma_a$, independent from $l$. Here we take the variance of $w_{ij}^{(l)}$ to be $\sigma_w^2/N$ so that the variance of each $h_i$ is $\sigma_w^2$, assuming each $\underline{x}_j$ is fixed (similarity for $v_{ij}^{(l)}$). Such an initialization scheme is standard in practice.

We make several key "physical assumptions" to make theoretical computations tractable:

**Axiom 3.1** (Symmetry of activations and gradients). *(a) We assume* $\langle(h_i^{(l)})^2\rangle = \langle(h_j^{(l)})^2\rangle$ *and* $\langle(x_i^{(0)})^2\rangle = \langle(x_j^{(0)})^2\rangle$ *for any* $i, j, l$. *(b) We also assume that the gradient* $\partial E/\partial x_i^{(l)}$ *with respect to the loss function* $E$ *satisfies* $\langle(\partial E/\partial x_i^{(l)})^2\rangle = \langle(\partial E/\partial x_j^{(l)})^2\rangle$ *for any* $i, j, l$.

One can see that Axiom 3.1(a) is satisfied if the input $x^{(0)} \in \{\pm 1\}^N$ and Axiom 3.1(b) is satisfied if Axiom 3.2 below is true and the gradient at the last layer $\partial E/\partial xL \in \{\pm 1\}^N$. But in general it is justified both empirically and theoretically as an approximation, because $(h_i^{(l)})^2 - (h_j^{(l)})^2$ stays about constant with $l$, but $(h_i^{(l)})^2$ and $(h_j^{(l)})^2$ grow rather quickly at the same pace with $l$ (as will be seen later in calculations), so that their additive difference becomes negligible; similarly for $(x_i^{(l)})^2$ and $(\partial E/\partial h_i^{(l)})^2$.

**Axiom 3.2** (Gradient independence). *(a) We assume the we use a different set of weights for back-propagation than those used to compute the network outputs, but sampled i.i.d. from the same distributions. (b) For any loss function $E$, we assume that the gradient at layer $l$, $\partial E / \partial x_i^{(l)}$, is independent from all activations $h_j^{(l)}$ and $x_j^{(l-1)}$ from the previous layer.*

Axiom 3.2(a) was first made in [11] for computing the mean field theory of gradients for feedforward tanh networks. This is similar to the practice of feedback alignment [8]. Even though we are the first to explicitly formulate Axiom 3.2(b), in fact it was already applied implicitly in the gradient calculations of [11]. Note that a priori Axiom 3.2(b) is not true, as $\partial E / \partial x_i^{(l)}$ depends on $\dot{\phi}(h_k^{(l+1)})$ for every $k$, which depend on $h_j^{(l)}$ for each $j$, and which depends on $x_k^{(l-1)}$ for every $k$. Nevertheless, in practice both subassumptions hold very well.

Now we define the central quantities studied in this paper. Inevitably, our paper involves a large amount of notation that may be confusing for the first-time reader. We have included a glossary of symbols (Table A.1) to ameliorate notation confusion.

**Definition 3.3.** Fix an input $x^{(0)}$. Define the **length quantities** $\mathbf{q}^{(l)} := \langle (h_1^{(l)})^2 \rangle$ and $\mathbf{p}^{(l)} := \langle (x_1^{(l)})^2 \rangle$ for $l > 0$ and $\mathbf{p}^{(0)} = \|x^{(0)}\|^2 / N$. Here the expectations $\langle \bullet \rangle$ are taken over all random initialization of weights and biases for all layers $l$, as $N \to \infty$ (large width limit).

Note that in our definition, the index 1 does not matter by Axiom 3.1.

**Definition 3.4.** Fix two inputs $x^{(0)}$ and $x^{(0)\prime}$. We write $\bullet'$ to denote a quantity $\bullet$ with respect to the input $x^{(0)\prime}$. Then define **the correlation quantities** $\boldsymbol{\gamma}^{(l)} := \langle h_1^{(l)} h_1^{(l)\prime} \rangle$ and $\boldsymbol{\lambda}^{(l)} := \langle x_1^{(l)} x_1^{(l)\prime} \rangle$ for $l > 0$ and $\boldsymbol{\gamma}^{(0)} = x^{(0)} \cdot x^{(0)\prime} / N$, where the expectations $\langle \bullet \rangle$ are taken over all random initialization of weights and biases for all layers $l$, as $N \to \infty$ (large width limit). Again, here the index 1 does not matter by Axiom 3.1. By **metric expressivity**, we mean $\mathbf{s}^{(l)} := \frac{1}{2N} \langle \|x^{(l)} - x^{(l)\prime}\|^2 \rangle = \frac{1}{2N} (\langle \|x^{(l)}\|^2 \rangle + \langle \|x^{(l)\prime}\|^2 \rangle - 2\langle x^{(l)} \cdot x^{(l)\prime} \rangle) = \frac{1}{2}(\mathbf{p}^{(l)} + \mathbf{p}^{(l)\prime}) - \boldsymbol{\gamma}^{(l)}$. Additionally, define **the cosine distance quantities** $\mathbf{e}^{(l)} := \boldsymbol{\gamma}^{(l)} / \sqrt{\mathbf{p}^{(l)} \mathbf{p}^{(l)\prime}}$ and $\mathbf{c}^{(l)} := \boldsymbol{\lambda}^{(l)} / \sqrt{\mathbf{q}^{(l)} \mathbf{q}^{(l)\prime}}$, and we will also call $\mathbf{e}^{(l)}$ **angular expressivity**.

In this paper, for the ease of presentation, we assume $\mathbf{p}^{(0)} = \mathbf{p}^{(0)\prime}$. Then, as we will see, $\mathbf{p}^{(l)} = \mathbf{p}^{(l)\prime}, \mathbf{q}^{(l)} = \mathbf{q}^{(l)\prime}$ for all $l$, and as a result, $\mathbf{e}^{(l)} = \boldsymbol{\gamma}^{(l)} / \mathbf{p}^{(l)}$ and $\mathbf{s}^{(l)} = \mathbf{p}^{(l)} - \boldsymbol{\gamma}^{(l)} = (1 - \mathbf{e}^{(l)}) \mathbf{p}^{(l)}$.

**Definition 3.5.** Fix an input $x^{(0)}$ and a gradient vector $(\partial E / \partial x_i^{(L)})_i$ of some loss function $E$ with respect to the last layer $x^{(L)}$. Then define **the gradient quantities** $\boldsymbol{\chi}^{(l)} := \langle (\partial E / \partial x_1^{(l)})^2 \rangle, \boldsymbol{\chi}_\bullet^{(l)} := \langle (\partial E / \partial \bullet_1^{(l)})^2 \rangle$ for $\bullet = a, b$, and $\boldsymbol{\chi}_\bullet^{(l)} := \langle (\partial E / \partial \bullet_{11}^{(l)})^2 \rangle$ for $\bullet = w, v$. Here the expectations are taken with Axiom 3.2 in mind, over both random initialization of forward and backward weights and biases, as $N \to \infty$ (large width limit). Again, the index 1 or 11 does not matter by Axiom 3.1.

**Asymptotic notations.** The expressions $f = O(g) \iff g = \Omega(f)$ have their typical meanings, and $f = \Theta(g)$ iff $f = O(g), g = O(f)$. We take $f(x) = \tilde{O}(g(x)) \iff g(x) = \tilde{\Omega}(f(x))$ to mean $f(x) = O(g \log^k x)$ for some $k \in \mathbb{Z}$ (this is slightly different from the standard usage of $\tilde{O}$), and $f = \tilde{\Theta}(g) \iff f = \tilde{O}(g) \ \& \ g = \tilde{O}(f)$. We introduce a new notation: $f = \check{\Theta}(g)$ if $f(x) = O(g(x) \cdot x^\epsilon)$ and $f(x) = \Omega(g(x) \cdot x^{-\epsilon})$, as $x \to \infty$, for any $\epsilon > 0$. All asymptotic notations are sign-less, i.e. can indicate either positive or negative quantities, unless stated otherwise.

# 4 Overview

The primary reason we may say anything about the average behavior of any of the above quantities is the central limit theorem: every time the activations of the previous layer pass through an affine layer whose weights are sampled i.i.d., the output is a sum of a large number of random variables, and thus follows approximately Gaussian distributions. The mean and variance of these distributions can be computed by keeping track of the mean and variances of the activations in the previous layer.

In what follows, we use this technique to derive recurrence equations governing $\mathbf{p}, \mathbf{q}, \boldsymbol{\gamma}, \boldsymbol{\lambda}, \boldsymbol{\chi}$ for different architectures and different activation functions. We use these equations to investigate the

dynamics of **e** and **s**, the key quantities in the forward pass, and the dynamics of $\boldsymbol{\chi}$, the key quantity in the backward pass.

The cosine distance **e** in some sense measures the angular geometry of two vectors. If $\mathbf{e} = 1$, then the vectors are parallel; if $\mathbf{e} = 0$, then they are orthogonal. Just as in [9] and [11], we will show that in all of the architectures and activations we consider in this paper, $\mathbf{e}^{(l)}$ converges to a fixed point $\mathbf{e}^*$ as $l \to \infty$ [1]. Thus, on the average, as vectors propagate through network, the geometry of the original input space, for example, linear separability, is "forgotten" by residual networks as well as by vanilla networks. But we will prove and verify experimentally that, while Poole et al. [9] and [11] showed that the convergence rate to $\mathbf{e}^*$ is exponential in a vanilla network, the convergence rate is rather only polynomial in residual networks, for tanh and $\alpha$-ReLU (Defn 5.2) nonlinearities; see Thm B.5, Thm B.11, Thm B.17, and Thm B.18. This slow convergence preserves geometric information in the input space, and allows a typical residual network to "hover over the edge of chaos": Even when the cosine distance $\mathbf{e}^{(l)}$ converges to 0, corresponding to "chaos", (resp. 1, corresponding to "stability"), for the number of layers usually seen in practice, $\mathbf{e}^{(l)}$ will reside well away from 0 (resp. 1).

Similarly, the quantity **s** measures the metric geometry of two vectors. The evolution of $\mathbf{s}^{(l)}$ with $l$ tells us the ability of the average network to separate two input points in terms of Euclidean distance. Again, for tanh and $\alpha$-ReLU ($\alpha < 1$) nonlinearities, **s** varies only polynomially with $l$.

On the other hand, $\boldsymbol{\chi}^{(l)}$ measures the size of gradient at layer $l$, and through it we track the dynamics of gradient backpropagation, be it explosion or vanishing. In contrast to vanilla tanh networks, which can experience both of these two phenomenon depending on the initialization variances, typical residual networks cannot have vanishing gradient, in the sense of vanishing $\boldsymbol{\chi}^{(l)}$ as $l \to 1$; see Thm B.5 and Thm B.12. Furthermore, while vanilla tanh networks exhibit exponentially vanishing or exploding gradients, all of the activation/architecture pairings considered here, except the full residual network with ReLU, have subexponential gradient dynamics. While tanh residual networks (reduced or full) has $\boldsymbol{\chi}^{(0)} \approx \exp(\Theta(\sqrt{l}))\boldsymbol{\chi}^{(l)}$ (Thm B.13), $\alpha$-ReLU residual networks for $\alpha < 1$ have $\boldsymbol{\chi}^{(0)} \approx \mathsf{poly}(l)\boldsymbol{\chi}^{(l)}$ (Thm B.20). Instead of $\partial E / \partial x_i$, we may also consider the size of gradients of actual trainable parameters. For tanh and $\alpha$-ReLU with $\alpha < 1$, they are still subexponential and polynomial (Thm B.21). On the other hand, while $\boldsymbol{\chi}^{(0)} = \exp(\Theta(l))\boldsymbol{\chi}^{(l)}$ for a ReLU resnet, its weight gradients have size independent of layer, within $O(1)$ (Thm B.21)! This is the only instance in this paper of gradient norm being completely preserved across layers.

The above overviews the theoretical portion of this paper. Through experiments, we discover that we can very accurately predict whether one random initialization leads to better performance than another on the test set, after training, by leveraging this theory we build. Residual networks of different nonlinearities have different *controlling quantities*: for resnets with tanh, the optimal initialization is obtained by controlling the gradient explosion $\boldsymbol{\chi}^{(0)}/\boldsymbol{\chi}^{(L)}$; whereas for ReLU and $\alpha$-ReLU, the optimal initialization is obtained by maximizing **s** without running into numerical issues (with floating point computation). See Section 6 for details.

Over the course of our investigation of $\alpha$-ReLU, we derived several new identities involving the associated kernel functions, first defined in [2], which relate them to the zeroth Bessel functions (Lemmas C.31 to C.34).

# 5   Theoretical Results

In what follows in the main text, we assume $\sigma_\bullet > 0$ for all $\bullet = w, v, b, a$; in the appendix, the formal statement of each main theorem will contain results for other cases. We are interested in the two major categories of nonlinearities used today: tanh-like and rectified units. We make the following formal definitions as a foundation for further consideration.

**Definition 5.1.** We say a function $\phi$ is **tanh-like** if $\phi$ is antisymmetric ($\phi(-x) = -\phi(x)$), $|\phi(x)| \leq 1$ for all $x$, $\phi(x) \geq 0, \forall x \geq 0$, and $\phi(x)$ monotonically increases to 1 as $x \to \infty$.

**Definition 5.2.** Define the $\alpha$-ReLU $\psi_\alpha(x) = x^\alpha$ if $x > 0$ and 0 otherwise. [2]

By applying the central limit theorem as described in the last section, we derive a set of recurrences for different activation/architecture pairs, shown in Table 1 (see appendix for proofs). They leverage certain integral transforms [3] as in the following

**Table 1:** Main Recurrences

| Antisymmetric/RRN | | Any/FRN | |
|---|---|---|---|
| $\mathbf{q} = \sigma_w^2 \underline{\mathbf{p}} + \sigma_b^2$ | $\mathbf{p} = \mathrm{V}\phi(\mathbf{q}) + \underline{\mathbf{p}}$ | $\mathbf{q} = \sigma_w^2 \underline{\mathbf{p}} + \sigma_b^2$ | $\mathbf{p} = \sigma_v^2 \mathrm{V}\phi(\mathbf{q}) + \sigma_a^2 + \underline{\mathbf{p}}$ |
| $\boldsymbol{\lambda} = \sigma_w^2 \underline{\boldsymbol{\gamma}} + \sigma_b^2$ | $\boldsymbol{\gamma} = \mathrm{W}\phi(\mathbf{q}, \boldsymbol{\lambda}) + \underline{\boldsymbol{\gamma}}$ | $\boldsymbol{\lambda} = \sigma_w^2 \underline{\boldsymbol{\gamma}} + \sigma_b^2$ | $\boldsymbol{\gamma} = \sigma_v^2 \mathrm{W}\phi(\mathbf{q}, \boldsymbol{\lambda}) + \sigma_a^2 + \underline{\boldsymbol{\gamma}}$ |
| | $\underline{\boldsymbol{\chi}} = (\sigma_w^2 \mathrm{V}\dot{\phi}(\mathbf{q}) + 1)\boldsymbol{\chi}$ | | $\underline{\boldsymbol{\chi}} = (\sigma_v^2 \sigma_w^2 \mathrm{V}\dot{\phi}(\mathbf{q}) + 1)\boldsymbol{\chi}$ |
| Theorems B.2, B.3, B.5 | | Theorems B.8, B.10, B.12 | |

**Table 2:** Summary of Main Dynamics Results. Note that while $\boldsymbol{\chi}^{(l)}$ is exponential for ReLU/FRN, the gradients with respect to weight parameters have norms ($\boldsymbol{\chi}_w$ and $\boldsymbol{\chi}_v$) constant in $l$ (Thm B.21). Also, the $\boldsymbol{\chi}^{(l)}$ entry for $\alpha$-ReLU is for $\alpha \in (3/4, 1)$ only

| | Tanh/RRN | | Tanh/FRN | | ReLU/FRN | $\alpha$-ReLU/FRN, $\alpha < 1$ | |
|---|---|---|---|---|---|---|---|
| $\mathbf{p}^{(l)}$ | $\Theta(l)$, | B.2 | $\Theta(l)$, | B.9 | $\exp(\Theta(l))$, B.16 | $\Theta(l^{1/(1-\alpha)})$, | B.16 |
| $\mathbf{s}^{(l)}$ | $\Theta(l)$, | B.4 | $\Theta(l)$, | B.11 | $\exp(\Theta(l))$, B.17 | $\Theta(l^{1/(1-\alpha)})$, | B.18 |
| $\mathbf{e}^{(l)} - \mathbf{e}^*$ | $\check{\Theta}(l^{\frac{2}{\pi}-1})$, | B.4 | $\mathrm{poly}(l)$, | B.11 | $\Theta(l^{-2})$, B.17 | $\mathrm{poly}(l)$, | B.18 |
| $\boldsymbol{\chi}^{(l)}$ | $\exp(\Theta(\sqrt{l}))$, B.6 | | $\exp(\Theta(\sqrt{l}))$, B.12 | | $\exp(\Theta(l))$, B.20 | $\Theta(l^{\frac{\alpha^2}{(1-\alpha)(2\alpha-1)}})$, B.20 | |

**Definition 5.3.** Define the transforms V and W by $\mathrm{V}\phi(q) := \mathrm{E}[\phi(z)^2 : z \sim \mathcal{N}(0, q)]$ and $\mathrm{W}\phi(\rho, \nu) := \mathrm{E}[\phi(z)\phi(z') : (z, z') \sim \mathcal{N}(0, \begin{pmatrix} \rho & \nu \\ \nu & \rho \end{pmatrix})]$.

These recurrences are able to track the corresponding quantities in practice very well. For example, Fig. 1 compares theory vs experiments for the tanh/FRN pair. The agreement is very good for tanh/RRN (not shown, but similar to the case of tanh/FRN with $\sigma_v = 1$ and $\sigma_a = 0$) and $\alpha$-ReLU/FRN as well (see Fig. A.1).

As mentioned in previous sections, we seek to characterize the long term/high depth behavior of all of the quantities defined in Section 2. To do so, we solve for the asymptotics of the recurrences in Table 1, where $\phi$ is instantiated with tanh or $\alpha$-ReLU. Our main dynamics results are summarized in Table 2.

## 5.1 Tanh

**Forward dynamics.** When $\phi = \tanh$, $\mathbf{p}^{(l)}$ and $\mathbf{q}^{(l)}$ increase as $\Theta(l)$ in either RRN or FRN (Thm B.2), as one might expect by observing that $\mathrm{V}\tanh(\mathbf{q}) \to 1$ as $\mathbf{q} \to \infty$ so that, for example in the RRN case, the recurrence $\mathbf{p} = \mathrm{V}\tanh(\mathbf{q}) + \underline{\mathbf{p}}$ becomes $\mathbf{p} = 1 + \underline{\mathbf{p}}$. This is confirmed graphically by the black lines of the leftmost chart of Fig. 1. We carefully verify that this intuition is correct in its proof in the appendix, and find that in fact $\mathbf{p}^{(l)} \sim l$ in the RRN case and $\mathbf{p}^{(l)} \sim (\sigma_v^2 + \sigma_a^2)l$ in the FRN case.

What about $\boldsymbol{\gamma}^{(l)}$? The middle chart of Fig. 1 shows that over time, $\mathbf{e}^{(l)} = \boldsymbol{\gamma}^{(l)}/\mathbf{p}^{(l)}$ contracts toward the center of the interval $[0, 1]$, but from the looks of it, it is not clear whether there is a stable fixed point $\mathbf{e}^*$ of $\mathbf{e}$ or not. We prove that, in fact **all trajectories of $\mathbf{e}$ not starting at 1 do converge to a single fixed point, but only at a polynomial rate**, in both the RRN and FRN cases (Thm B.2 and Thm B.10); we can even explicitly compute the fixed point and the rate of convergence: For FRN, there is a **unique stable fixed point $\mathbf{e}^* < 1$** determined by the equation

$$\mathbf{e}^* = \frac{1}{\sigma_v^2 + \sigma_a^2}[\sigma_v^2 \frac{2}{\pi} \arcsin(\mathbf{e}^*) + \sigma_a^2],$$

and $|\mathbf{e}^* - \mathbf{e}^{(l)}|$ decreases like $l^{-\delta^*}$, where

$$\delta^* := 1 - \frac{2}{\pi} \frac{1}{\sqrt{1 - (\mathbf{e}^*)^2}} \frac{\sigma_v^2}{\sigma_v^2 + \sigma_a^2}.$$

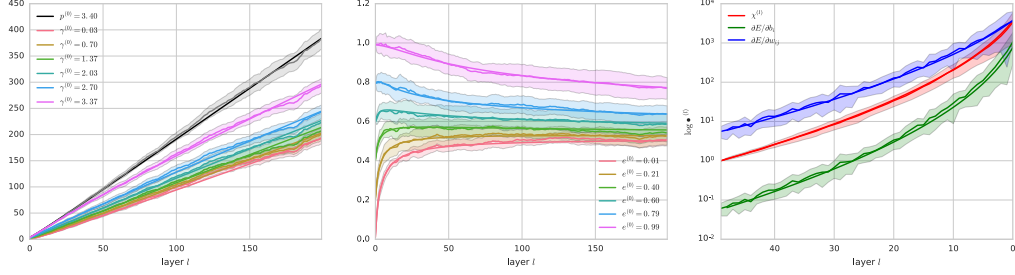

**Figure 1:** Our equations predict the relevant quantities very well in practice. These plots make the comparison between prediction and measurements for the full resnet with tanh activation, with $\sigma_v^2 = 1.5$, $\sigma_a^2 = .5$, $\sigma_w^2 = 1.69$, $\sigma_b^2 = .49$. Left-to-right: **(a)** $\mathbf{p}^{(l)}$ and $\boldsymbol{\gamma}^{(l)}$ against layer $l$ for 200 layers. **(b)** $\mathbf{e}^{(l)} = \boldsymbol{\gamma}^{(l)}/\mathbf{p}^{(l)}$ against $l$ for 200 layers. Both (a) and (b) trace out curves for different initial conditions. **(c)** Different gradient quantities against $l$ for 50 layers. From left to right the layer number $l$ decreases, following the direction of backpropagation. Notice that the gradient increases in norm as $l \to 1$. All three figures exhibit smooth curves, which are theoretical estimates, and irregular curves with shades around them, which indicate empirical means and standard deviations (both of which taken in regular scale, not log scale). (a) and (b) are made with 20 runs of resnets of width 1000. (c) is made with 25 runs of resnets of width 250.

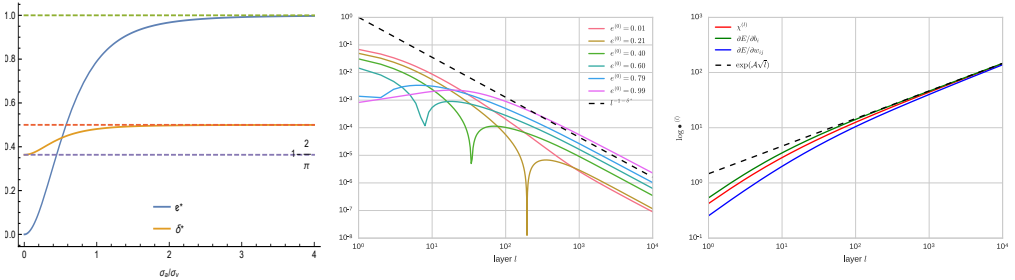

**Figure 2:** Left-to-right: **(a)** Plots of $\mathbf{e}^*$ and $\delta^*$ against $\sigma_a/\sigma_v$. **(b)** In log-log scale: the dashed line is $l^{-\delta^*-1}$, and the colored lines are $\mathbf{e}^{(l)} - \mathbf{e}^{(l-1)}$ for different initial conditions $\mathbf{e}^{(0)}$. That they become parallel at about $l = 400$ on verifies that $\mathbf{e}^{(l)} = \Theta(l^{-\delta^*})$. [4] **(c)** In log-log scale: The dashed line is $\mathcal{A}\sqrt{l}$ ($\mathcal{A}$ given in Thm B.13), and the colored lines are $\log(\bullet^{(1)}/\bullet^{(l)})$ for $\bullet = \chi, \chi_b, \chi_w$. That they all converge together starting around $l = 1000$ indicates that the approximation in Thm B.13 is very good for large $l$.

Since $\mathbf{e}^* < 1$, $\mathbf{s} = (1 - \mathbf{e})\mathbf{p} = \Theta(\mathbf{p}) = \Theta(l)$. The case of RRN can be viewed as a special case of the above, setting $\sigma_v^2 = 1$ and $\sigma_a^2 = 0$, which yields $\mathbf{e}^* = 0$ and $\delta^* = 1 - \frac{2}{\pi}$. We observe that both $\mathbf{e}^*$ and $\delta^*$ only depend on the ratio $\rho := \sigma_a/\sigma_v$, so in Fig. 2 we graph these two quantities as a function of $\rho$. $\mathbf{e}^*$ and $\delta^*$ both increase with $\rho$ and asymptotically approach 1 and $1/2$ respectively from below. When $\rho = \sigma_a = 0$, $\mathbf{e}^* = 0$ and $\delta^* = 1 - \frac{2}{\pi}$. Thus the rate of convergence at its **slowest** for tanh/FRN is $\delta^* = 1 - \frac{2}{\pi} \approx 0.36338$, where asymptotically the network tends toward a **chaotic regime** $\mathbf{e}^* = 0$, corresponding to a large weight variance and a small bias variance; it at its **fastest** is $\delta^* = 1/2$, where asymptotically the network tends toward a **stable regime** $\mathbf{e}^* = 1$, corresponding to a large bias variance and small weight variance. We verify $\delta^*$ by comparing $\mathbf{e}^{(l)} - \mathbf{e}^{(l-1)}$ to $l^{-\delta^*-1}$ in log-log scale. If $\mathbf{e}^{(l)} = \Theta(l^{-\delta^*})$, then $\mathbf{e}^{(l)} - \mathbf{e}^{(l-1)} = \Theta(l^{-\delta^*-1})$ and should obtain the same slope as $l^{-\delta^*-1}$ as $l \to \infty$. The middle figure of Fig. 2 ascertains that this is indeed the case, starting around layer number 400.

**Backward dynamics.** Finally, we show that the gradient is approximated by

$$\boldsymbol{\chi}^{(m)} = \exp(\mathcal{A}(\sqrt{l} - \sqrt{m}) + O(\log l - \log m))\boldsymbol{\chi}^{(l)} \qquad (\star)$$

where $\mathcal{A} = \frac{4}{3}\sqrt{\frac{2}{\pi}}\sigma_w$ in the RRN case and $\mathcal{A} = \frac{4}{3}\sqrt{\frac{2}{\pi}}\frac{\sigma_v^2\sigma_w}{\sqrt{\sigma_v^2+\sigma_a^2}}$ in the FRN case (Thm B.6 and Thm B.13). The rightmost plot of Fig. 2 verifies that indeed, for large $l \geq 1000$, this is a very good approximation. This demonstrates that the mean field assumption of independent backpropagation weights is very practical and convenient even for residual networks.

Note that in the FRN case, the constant $\mathcal{A}$ can be decomposed into $\mathcal{A} = \frac{4}{3}\sqrt{\frac{2}{\pi}} \cdot \sigma_v \cdot \sigma_w \cdot (1 + \sigma_a^2/\sigma_v^2)^{-1/2}$. Consider the ratio $\rho := \sigma_a/\sigma_v$. If $\rho \gg 1$, then $\mathbf{e}^* \approx 1$ (Fig. C.17), meaning that the typical network essentially computes a constant function, and thus unexpressive; at the same time, large $\rho$ makes $\mathcal{A}$ small, and thus ameliorating the gradient explosion problem, making the network more trainable. On the other hand, if $\rho \ll 1$, then $\mathbf{e}^* \approx 0$ (Fig. C.17), the typical network can tease out the finest differences between any two input vectors, and a final linear layer on top of such a network should be able to express a wide variety of functions [9]; at the same time, small $\rho$ increases $\mathcal{A}$, worsening the gradient explosion problem, making the network less trainable. This is the same expressivity-trainability tradeoff discussed in [11].

## 5.2 $\alpha$-ReLU

**Forward dynamics.** As with the tanh case, to deduce the asymptotic behavior of random $\alpha$-ReLU resnets, we need to understand the transforms $\mathrm{V}\psi_\alpha$ and $\mathrm{W}\psi_\alpha$. Fortunately, $\mathrm{V}\psi_\alpha$ has a closed form, and $\mathrm{W}\psi_\alpha$ has been studied before [2]. In particular, if $\alpha > -\frac{1}{2}$, then $\mathrm{V}\psi_\alpha(\mathbf{q}) = \mathsf{c}_\alpha \mathbf{q}^\alpha$, where $\mathsf{c}_\alpha$ is a constant with a closed form given by Lemma B.15. In addition, by [2], we know that $\mathrm{W}\psi_\alpha(\mathbf{q}, \mathbf{cq}) = \mathrm{V}\psi_\alpha(\mathbf{q})\mathbb{J}_\alpha(\mathbf{c})$ for $\mathbb{J}_\alpha$ given in Appendix C.7.1. Fig. C.17 shows a comparison of $\mathbb{J}_\alpha$ for different $\alpha$s along with the identity function.

Substituting in $\mathsf{c}_\alpha \mathbf{q}^\alpha$ for $\mathrm{V}\psi_\alpha$, we get a difference equation $\mathbf{p} - \underline{\mathbf{p}} = \sigma_v^2 \mathsf{c}_\alpha (\sigma_w^2 \underline{\mathbf{p}} + \sigma_b^2)^\alpha + \sigma_a^2$ governing the evolution of $\mathbf{p}$. This should be reminiscent of the differential equation $\dot{P}(l) = CP(l)^\alpha$, which has solution $\propto l^{1/(1-\alpha)}$ for $\alpha < 1$, and $\propto \exp(Cl)$ when $\alpha = 1$. And indeed, the solutions $\mathbf{p}^{(l)}$ to these difference equations behave asymptotically exactly like so (Thm B.16). Thus **ReLU behaves very explosively compared to $\alpha$-ReLU with $\alpha < 1$.** In fact, in simulations, for $\sigma_w^2 = 1.69$ and $\sigma_v^2 = 1.5$, the ReLU resnets overflows into `inf`s after around 100 layers, while there's no problem from any other kind of networks we consider.

Regardless, $\alpha$**-ReLU for all $\alpha$ massages $\mathbf{e}^{(l)}$ toward a fixed point $\mathbf{e}^*$ that depends on $\alpha$.** When $\phi = \psi_1$, the standard ReLU, $\mathbf{e}^{(l)}$ converges to 1 asymptotically as $Cl^{-2}$ for an explicit constant $C$ depending on $\sigma_v$ and $\sigma_w$ only (Thm B.17), so that $\mathbf{s} = (1 - \mathbf{e})\mathbf{p} = \Theta(l^{-2}\exp(\Theta(l))) = \exp(\Theta(l))$. When $\phi = \psi_\alpha$ for $\alpha < 1$, then $\mathbf{e}^{(l)}$ converges to the nonunit fixed point $\mathbf{e}^*$ of $\mathbb{J}_\alpha$ at a rate of $\tilde{\Theta}(l^{-\mu})$, where $\mu = (1 - \dot{\mathbb{J}}_\alpha(\mathbf{e}^*))/(1-\alpha)$ is independent of the variances (Thm B.18), so that $\mathbf{s} = \Theta(\mathbf{p})$. These rates are verified in Fig. A.2.

**Backward dynamics.** Finally, we have also characterized the rate of gradient growth for any $\alpha \in (\frac{3}{4}, 1]$.[5] **In the case of $\alpha = 1$, the dynamics of $\boldsymbol{\chi}$ is exponential**, the same as that of $\mathbf{p}$, $\boldsymbol{\chi}^{(l-m)} = \boldsymbol{\chi}^{(l)}B^m$ where $B = \frac{1}{2}\sigma_v^2\sigma_w^2 + 1$. **For $\alpha \in (\frac{3}{4}, 1)$, the dynamics is polynomial**, but with different exponent in general from that of the forward pass: $\boldsymbol{\chi}^{(l-m)} = \Theta(1)\boldsymbol{\chi}^{(l)}(l/(l-m))^R$ for $R = \frac{\alpha^2}{(1-\alpha)(2\alpha-1)}$, where the constants in $\Theta(1)$ do not depend on $l$ or $m$. This exponent $R$ is minimized on $\alpha \in [\frac{3}{4}, 1)$ at $\alpha = {}^3\!/_4$, where $R = {}^9\!/_2$ (but on $\alpha \in (\frac{1}{2}, 1)$ it is minimized at $\alpha = {}^2\!/_3$, where $R = 4$); see Fig. B.8. These exponents are verified empirically in Fig. A.2.

Looking only at $\boldsymbol{\chi}$ and the gradients against the biases, it seems that ReLU suffers from a dramatic case of exploding gradients. But in fact, because $\boldsymbol{\chi}$ gains a factor of $B$ moving backwards while $\mathbf{p}$ loses a factor of $B$, the gradient norm $\chi_w^{(l-m)}$ (and similarly for $\chi_v^{(l-m)}$) is independent of how far, $m$, the gradient has been propagated (Thm B.21) — this is certainly the best gradient preservation among all of the models considered in this paper. Thus strangely, random ReLU FRN exhibits both the best (constant for $v$ and $w$) and the worse (exponential for $a$ and $b$) gradient dynamics. This begs the question, then, is this a better deal than other $\alpha$-ReLU for which for any learnable parameter we have at most a polynomial blowup with depth in its gradient? Our experiments (discussed below) show that $\alpha$-ReLU is useful to the extent that smaller $\alpha$ avoids numerical issues with exponentiating forward and backward dynamics, but the best performance is given by the largest $\alpha$ that avoids them (Fig. 3(c, d)); in fact, the metric expressivity $\mathbf{s}$, determines performance, not gradient explosion (see $\alpha$-ReLU experiments).

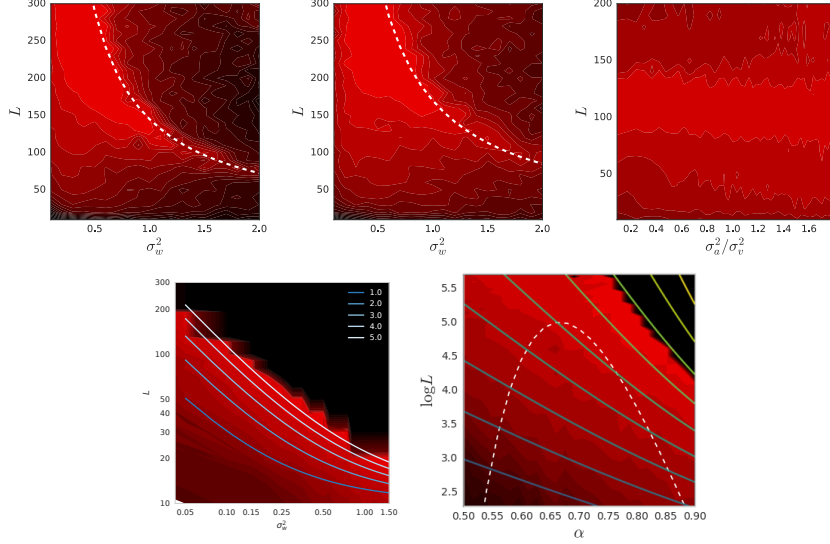

**Figure 3:** From left to right, top to bottom: **(a)** and **(b)**: $\sigma_w^2$, $L$, and test set accuracy of a grid of tanh reduced (left) and full (right) resnets trained on MNIST. Color indicates performance, with ligher colors indicating higher accuracy on test set. Other than the values on the axes, we have fixed $\sigma_b^2 = \sigma_a^2 = \frac{1}{2}$ and $\sigma_v^2 = 1$. The white dotted lines are given by $\sigma_w^2 L = C$, where $C = 170$ on the left and $C = 145$ on the right. We see that both dotted lines accurately predict the largest optimal $\sigma_w$ for each depth $L$. **(c)** Varying the ratio $\sigma_a^2/\sigma_v^2$ while fixing $\sigma_v/\sqrt{1 + \sigma_a^2/\sigma_v^2}$, and thus fixing $\mathcal{A}$, the leading constant of $\log \chi^{(0)}/\chi^{(L)}$. **(d)** in log-log scale: Heatmap gives the test accuracies of ReLU FRN for varying $\sigma_w^2$ and $L$. Curves give level sets for the log ratios $\log \mathbf{s}^{(L)}/\mathbf{s}^{(0)} \approx \log \mathbf{p}^{(L)}/\mathbf{p}^{(0)} \approx \log \boldsymbol{\chi}^{(0)}/\boldsymbol{\chi}^{(L)} = L \log(1 + \sigma_v^2 \sigma_w^2/2)$. **(e)** Red heatmap shows the test accuracies of a grid of $\alpha$-ReLU FRN with varying $\alpha$ and $L$ as shown, but with all $\sigma_\bullet$s fixed. The white dashed curve gives a typical contour line of $L^R = \text{const}$, where $R = \frac{\alpha^2}{(1-\alpha)(2\alpha-1)}$. The yellow-to-blue curves form a set of level curves for $\mathbf{s}^{(l)} = \mathbf{p}^{(l)} - \boldsymbol{\gamma}^{(l)} = \text{const}$, with yellow curves corresponding to higher levels.

## 6  Experimental Results

Our experiments show a dichotomy of what matters in initialization: for tanh resnets, quality of an initialization is determined by how much gradient explosion there is (measured by $\boldsymbol{\chi}^{(0)}/\boldsymbol{\chi}^{(L)}$); for ($\alpha$-)ReLU resnets, it is determined by how expressive the random network is (measured by the metric expressivity $\mathbf{s}^{(L)}$). We hypothesize this is because in tanh resnets, the gradient dynamics is much more explosive than the expressivity dynamics ($\exp(\Theta(\sqrt{l}))$ vs $\Theta(l)$), whereas for ReLU it's somewhat the opposite ($\boldsymbol{\chi}_w, \boldsymbol{\chi}_v = \Theta(1)$ vs $\mathbf{s} = \exp(\Theta(l))$).

**Tanh, vary $\sigma_w$.**    We train a grid of reduced and full tanh resnets on MNIST, varying the variance $\sigma_w^2$ and the number of layers (for FRN we fix $\sigma_v = 1$). The results are indicated in Fig. 3(a, b). We see that in either model, deeper resnets favor much smaller $\sigma_w$ than shallower ones. The white dotted lines in Fig. 3(a, b) confirm our theory: according to Eq. ($\star$), for the same gradient ratio $R = \boldsymbol{\chi}^{(0)}/\boldsymbol{\chi}^{(L)}$, we want $\log R \approx \sigma_w \sqrt{L}$. Indeed, the white dotted lines in Fig. 3(a, b) trace out such a level curve and it remarkably pinpoints the largest $\sigma_w$ that gives the optimal test set accuracy for each depth $L$. Why isn't the best initialization given by $R = 1 \iff \sigma_w = 0$? We believe that when $L$ and/or $\sigma_w$ is small, gradient dynamics no longer dominates the initialization quality because it has "less room to explode," and expressivity issues start to dampen the test time performance.

**Tanh, vary $\sigma_a^2/\sigma_v^2$.**    As suggested in the analysis of Eq. ($\star$), the ratio $\rho^2 = \sigma_a^2/\sigma_v^2$ determines the fixed point $\mathbf{e}^*$ and its convergence rate by itself while also contributes to the rate of gradient explosion in tanh FRN. We seek to isolate its effect on forward dynamics by varying $\sigma_v$ with $\rho$ such that $\sigma_v/\sqrt{1 + \rho^2}$ is kept constant, so that the leading term of the log gradient ratio is kept approximately equal for each $L$ and $\rho$. Fig. 3(c) shows the test accuracies of a grid of tanh FRN initialized with such an ensemble of $\sigma_\bullet$s. What stands out the most is that performance is maximized essentially

around a fixed value of $L$ regardless of $\rho$, which shows that indeed gradient dynamics determines the initialization quality in tanh resnets. There is also a minor increase in performance with increasing $\rho$ regardless of $L$; this is counterintuitive as increasing $\rho$ means "decreasing expressivity." It is currently not clear what accounts for this effect.

**ReLU, vary $\sigma_w$**    We train a grid of ReLU FRN on MNIST, varying $\sigma_w^2 \in [0, 1.5]$ while fixing $\sigma_v^2 = 1, \sigma_a^2 = \sigma_b^2 = \frac{1}{2}$. The resulting test set accuracies are shown in Fig. 3(d). The dark upper region signifies failure of training caused by numerical issues with exploding activation and gradient norms: This corresponds to the region where $\mathbf{p}^{(L)}$, which is a measure of the mean magnitude of an neuronal activation in layer $L$, becomes too big. We see that the best test accuracies are given by depths just below where these numerical issues occur. However, if we were to predict that the optimal init is the one minimizing $\boldsymbol{\chi}^{(0)}/\boldsymbol{\chi}^{(L)} \geq 1$, then we would be wrong — in fact it is exactly the opposite. In this case, the dynamics of $\mathbf{s}^{(l)}, \mathbf{p}^{(l)}$, and $\boldsymbol{\chi}^{(0)}/\boldsymbol{\chi}^{(l)}$ are approximately the same (all $\exp(\Theta(l))$ with the same hidden constants), and optimal performance corresponds to the highest $\mathbf{s}^{(L)}$, $\mathbf{p}^{(L)}$, and $\boldsymbol{\chi}^{(0)}/\boldsymbol{\chi}^{(L)}$ without running into infs.

**$\alpha$-ReLU, vary $\alpha$.**    We similarly trained a grid of $\alpha$-ReLU FRN on MNIST, varying only $\alpha$ and the depth, fixing all $\sigma_\bullet$. Fig. 3(e) shows their test accuracies. We see similar behavior to ReLU, where when the net is too deep, numerical issues doom the training (black upper right corner), but the best performance is given by $L$ just below where this problem occurs. In this case, if we were to predict optimality based on minimizing gradient explosion, we would be again wrong, and furthermore, the contour plot of $\boldsymbol{\chi}^{(0)}/\boldsymbol{\chi}^{(L)}$ (white dashed line) now gives no information at all on the test set accuracy. In contrast, the contours for $\mathbf{s}^{(l)}$ succeeds remarkably well at this prediction (yellow/green lines).[6] By interpolation, this suggests that indeed in the ReLU case, it is expressivity, not trainability, which determines performance at test time.

In all of our experiments, we did not find $\mathbf{e}$ dynamics to be predictive of neural network performance.

# 7    Conclusion

In this paper, we have extended the mean field formalism developed by [9, 10, 11] to residual networks, a class of models closer to practice than classical feedforward neural networks as were investigated earlier. We proved and verified that in both the forward and backward passes, most of the residual networks discussed here do not collapse their input space geometry or the gradient information exponentially. We found our theory incredibly predictive of test time performance despite saying nothing about the dynamics of training. In addition, we overwhelmingly find, through theory and experiments, that an optimal initialization scheme must take into account the depth of the residual network. The reason that Xavier [4] or He [5] scheme are not the best for residual networks is in fact not that their statistical assumptions are fragile — theirs are similar to our mean field theoretic assumptions, and they hold up in experiments for large width — but rather that their structural assumptions on the network break very badly on residual nets.

**Open Problems.**    Our work thus have shown that optimality of initialization schemes can be very unstable with respect to architecture. We hope this work will form a foundation toward a mathematically grounded initialization scheme for state-of-the-art architectures like the original He et al. residual network. To do so, there are still two major components left to study out of the following three: 1. Residual/skip connection 2. Batchnorm 3. Convolutional layers. Recurrent architectures and attention mechanisms are also still mostly unexplored in terms of mean field theory. Furthermore, many theoretical questions still yet to be resolved; the most important with regard to mean field theory is: why can we make Axioms 3.1 and 3.2 and still be able to make accurate predictions? We hope to make progress on these problems in the future and encourage readers to take part in this effort.

## Acknowledgments

Thanks to Jeffrey Ling for early exploration experiments and help with the initial draft. Thanks to Felix Wong for offering his wisdom and experience working in statistical physics.

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

# Notes

[1]Under simplified conditions, Daniely et al. [3] showed that there exists a fixed point for any "well-behaved" activation function in a feedforward net. However, this result does not apply to architectures with residual connections.

[2] Note that in practice, to avoid the diverging gradient $\dot{\psi}_\alpha(x) \to \infty$ as $x \to 0$, we can use a tempered version $\Psi_\alpha(x)$ of $\alpha$-ReLU, defined by $\Psi_\alpha(x) = (x + \epsilon)^\alpha - \epsilon^\alpha$ on $x > 0$ and 0 otherwise, for some small $\epsilon > 0$. The conclusions of this paper on $\psi_\alpha$ should hold similarly for $\Psi_\alpha$ as well.

[3]Daniely et al. [3] called the version of $W\phi$ with fixed $\rho = 1$ the "dual function" of $\phi$.

[4]A more natural visualization is to graph $\mathbf{e}^{(l)} - \mathbf{e}^*$ versus $l^{-\delta^*}$, but because of floating point precision, $\mathbf{e}^{(l)} - \mathbf{e}^*$ doesn't converge to 0, but a small number close to 0, so that the log-log plot wouldn't look like what is expected.

[5]Our derivations actually apply to all $\alpha \in (\frac{1}{2}, 1]$, where at $\alpha = \frac{1}{2}$, the expected norm of the gradient diverges within our mean field formalism. However, at $\alpha \leq \frac{3}{4}$, the variance of the gradient already diverges (Thm B.19), so we cannot expect the empirical values to agree with our theoretical predictions. But in fact, empirically our theoretical predictions seem to form an upper bound on the gradient norms (see Fig. A.1).

[6]the contour for $\mathbf{p}^{(l)}$ is similar, but its slopes are slightly off from the heatmap contours.

