[Supplementary Material]

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

# Appendices

## A   Additional Figures

In figures appearing in the appendix, ⅂ means $\chi$ (due to legacy reasons).

$\alpha = 1$

$\alpha = .9$

$\alpha = .8$

$\alpha = .7$

$\alpha = .6$

$\alpha = .55$

$\alpha = .51$

**Figure A.1:** Empirical vs theoretical dynamics for $\mathbf{p}^{(l)}, \mathbf{e}^{(l)}$, and different gradient quantities for $\alpha$-ReLU, with format similar to Fig. 1. We refer to each figure on each row from left to right as (a), (b), and (c). Note that in the $\alpha = 1$ case, figure (a) ($\mathbf{p}^{(l)}$ and $\boldsymbol{\gamma}^{(l)}$ for different initial values) has log scale y-axis and (a) and (b) have x-axis ranging from 1 to 50, while for other $\alpha$, (a) has normal y-axis and (a) and (b) have x-axis ranging from 1 to 200. We do so because the norm of the activation vector in a typical ReLU resnet blows up into NaN at around layer 90, while this is not a problem for $\alpha < 1$. Our theoretical predictions track the average of empirical values closely for forward quantities $\mathbf{p}^{(l)}, \boldsymbol{\gamma}^{(l)}$, and $\mathbf{e}^{(l)}$ for all $\alpha$, but variance is extremely large for $\mathbf{e}^{(l)}$ at $\alpha = 1$; it also predicts the average gradient norm accurately for $\alpha = 1$ to $\alpha = .7$ (despite the fact that we should not expect so for $\alpha \leq .75$ due to exploding variance (Thm B.19)), although variance is large for $\alpha = 1$ at earlier layers (i.e. later layers w.r.t backpropagation). However it *consistently and significantly overestimates* the average gradient norm for $\alpha = .6$ to $\alpha = .5$, where the variance is so large that one standard deviation below the mean results in negative values. All plots are made with parameters $\sigma_v^2 = 1.5, \sigma_a^2 = .5, \sigma_w^2 = 1.69, \sigma_b^2 = .49$; only $\alpha$ is varied. All figures exhibit smooth curves, which are theoretical estimates, and irregular curves with shades around them, which indicate empirical means and standard deviations (both of which taken in regular scale, not log scale). For each $\alpha$, figures (a) and (b) are made with 20 runs of resnets of width 1000. (c) is made with 25 runs of resnets of width 250.

**Figure A.2:** We verify the exponents of the forward and backward dynamics for $\alpha$-ReLU FRN. For each row, the figures are labeled (a) and (b) from left to right. The format is the same as in Fig. C.17. All figures are in log-log scale. **(a)** We exhibit our theoretical dynamics of the cosine distance $\mathbf{e}^{(l)}$ based on the recurrences Thm B.8 and Thm B.10 for different initial conditions $\mathbf{e}^{(0)}$. We draw $|\mathbf{e}^{(l)} - \mathbf{e}^{(l-1)}|$ for each of these dynamics in colored solid lines. We predict that each dynamic is $\check{\Theta}(l^{-\mu})$, where $\mu = (1 - \mathring{\mathbb{J}}_\alpha(\mathbf{e}^*))/(1 - \alpha)$, and the dashed line gives $l^{-\mu-1}$ (Thm B.18), shifted vertically to better compare the slope in log scale (i.e. the exponent of the polynomial dynamics). (See footnote 4 for why we plot the dynamics this way). We see that the our asymptotic prediction is very accurate for the sequence of $\mathbf{e}^{(l)}$ that starts with $\mathbf{e}^{(0)} = 0.99$, the closest to $\mathbf{e}^*$ for each $\alpha$, while other lines only slowly converge to the same exponent (which is the slope in the log-log plot). This is to be expected based on the proof of Thm B.18. For $\alpha = .9$, the $\mathbf{e}^{(0)} = .99$ line upticks at around $10^3$ and then turn into NaNs due to numerical instability. **(b)** Colored lines are $\bullet^{(0)}/\bullet^{(l)}$ for $\bullet = \chi, \chi_b, \chi_w$ (we are not taking logs in addition to plotting in log-log scale like in Fig. C.15). The dashed lines are our asymptotic predictions for the dynamics with corresponding colors, based on Thm B.21, again shifted appropriately to easily compare slope visually. We see that for every alpha our asymptotic predictions are highly accurate. For both (a) and (b), we did not show $\alpha = 1$ case as ReLU FRN runs into numerical issues quickly (i.e. with even for 100 layers) because of exponential explosions in $\mathbf{p}^{(l)}$ and $\chi^{(l)}$ as predicted by Thms B.16 and B.20, so we cannot expect to empirically verify the precise predicted asymptotics. All plots are made with parameters $\sigma_v^2 = 1.5, \sigma_a^2 = .5, \sigma_w^2 = 1.69, \sigma_b^2 = .49$; only $\alpha$ is varied.

**Table A.1:** Glossary of Symbols. "Mean normalized" is abbreviated "m.n."

| Symbol | Meaning | Ref |
|---|---|---|
| $\sigma_\bullet$ | standard deviation of trainable parameter $\bullet$ | |
| $x^{(l)}$ | activation vector/input vector | |
| $h^{(l)}$ | hidden vector | |
| $N$ | width (same across all layers) | |
| $\mathbf{p}^{(l)}$ | m.n. squared length of activation vector $x^{(l)}$ | 3.3 |
| $\mathbf{q}^{(l)}$ | m.n. squared length of hidden vector $h^{(l)}$ | 3.3 |
| $\boldsymbol{\gamma}^{(l)}$ | m.n. dot product $x^{(l)} \cdot x^{(l)\prime}$ | 3.4 |
| $\boldsymbol{\lambda}^{(l)}$ | m.n. dot product $h^{(l)} \cdot h^{(l)\prime}$ | 3.4 |
| $\mathbf{s}^{(l)}$ | m.n. squared distance $\|x^{(l)} - x^{(l)\prime}\|^2$ | 3.4 |
| $\mathbf{e}^{(l)}$ | cosine distance $\boldsymbol{\gamma}^{(l)}/\sqrt{\mathbf{p}^{(l)}\mathbf{p}^{(l)\prime}}$ | 3.4 |
| $\mathbf{e}^*$ | limit value of $\mathbf{e}^{(l)}$ as $l \to \infty$ | |
| $\mathbf{c}^{(l)}$ | cosine distance $\boldsymbol{\lambda}^{(l)}/\sqrt{\mathbf{q}^{(l)}\mathbf{q}^{(l)\prime}}$ | 3.4 |
| $\boldsymbol{\chi}^{(l)}$ | m.n. gradient squared norm w.r.t. $x^{(l)}$ | 3.5 |
| $\boldsymbol{\chi}_\bullet^{(l)}$ | m.n. gradient squared norm w.r.t. trainable parameter $\bullet$ | 3.5 |
| $\phi$ | variable nonlinearity $\mathbb{R} \to \mathbb{R}$ | |
| $\psi_\alpha$ | $\alpha$-ReLU | 5.2 |
| V | variance integral transform | 5.3 |
| W | covariance integral transform | 5.3 |
| $\delta^*$ | $\mathbf{e}^{(l)}$ converges like $\Theta(l^{-\delta^*})$ in tanh FRN | B.11 |
| $\mathcal{A}$ | leading coeff of $\log \boldsymbol{\chi}^{(0)}/\boldsymbol{\chi}^{(L)}$ in tanh FRN | B.13 |
| $R$ | $\log \boldsymbol{\chi}^{(0)}/\boldsymbol{\chi}^{(L)} \sim R \log L$ for $(\alpha < 1)$-ReLU | B.20 |
| $\mathbb{J}_\alpha$ | kernel function of $\alpha$-ReLU | C.30 |

# B A Listing of Main Theorems

## B.1 Tanh

### B.1.1 Reduced Residual Network

**Lemma B.1.** *Suppose $\phi$ is antisymmetric. Then in an RRN, $\mathbf{p}$ and $\mathbf{q}$ satisfy the recurrence*

$$\mathbf{q} = \sigma_w^2 \underline{\mathbf{p}} + \sigma_b^2$$
$$\mathbf{p} = \mathrm{V}\phi(\mathbf{q}) + \underline{\mathbf{p}}.$$

**Theorem B.2.** *Suppose $\phi$ is tanh-like. Assume RRN architecture.*

- *If $\sigma_w = 0$, then $\mathbf{p}^{(l)} = l\mathrm{V}\phi(\sigma_b^2) + \mathbf{p}^{(0)}$ and $\mathbf{q}^{(l)} = \sigma_b^2$.*

- *If $\sigma_w > 0$, $\lim_{l\to\infty} \mathbf{p}^{(l)}/l = 1$ and $\lim_{l\to\infty} \mathbf{q}^{(l)}/(\sigma_w^2 l) = 1$. If $\phi = \tanh$, then we can obtain more terms of the asymptotic expansions:*

$$\mathbf{p}^{(l)} = l - 2C\sigma_w^{-1}l^{1/2} - C^2\sigma_w^{-2}\log l + O(1)$$
$$\mathbf{q}^{(l)} = \sigma_w^2 l - 2C\sigma_w l^{1/2} - C^2 \log l + O(1)$$

*as $l \to \infty$, where $C = \sqrt{2/\pi}$.*

**Theorem B.3.** *Suppose $\phi$ is antisymmetric. Then in an RRN, $\boldsymbol{\lambda}$ and $\boldsymbol{\gamma}$ satisfy the recurrence*

$$\boldsymbol{\lambda} = \sigma_w^2 \underline{\boldsymbol{\gamma}} + \sigma_b^2$$
$$\boldsymbol{\gamma} = \mathrm{W}\phi(\mathbf{q}, \boldsymbol{\lambda}) + \underline{\boldsymbol{\gamma}}.$$

**Theorem B.4.** *Suppose $\phi$ is a tanh-like nonlinearity in an RRN. Assume $\mathbf{e}^{(0)} < 1$.*

- *If $\sigma_w = 0$, then $\boldsymbol{\gamma}^{(l)} = l\mathrm{W}\phi(\sigma_b^2, \sigma_b^2) + \boldsymbol{\gamma}^{(0)} = l\mathrm{V}\phi(\sigma_b^2) + \boldsymbol{\gamma}^{(0)}$ and $\boldsymbol{\lambda}^{(l)} = \sigma_b^2$, so that $\mathbf{e}^{(l)} \to 1$ and $1 - \mathbf{e}^{(l)} = \Theta(l^{-1})$. As a result, $\mathbf{s}^{(l)} = \mathbf{p}^{(l)}(1 - \mathbf{e}^{(l)}) = \Theta(1)$.*

- If $\sigma_w > 0$, then $\boldsymbol{\gamma}^{(l)} = \check{\Theta}(l^{\frac{2}{\pi}})$, and $\mathbf{e}^{(l)} \to 0$ like $\check{\Theta}(l^{\frac{2}{\pi}-1})$. Thus $\mathbf{s}^{(l)} = \Theta(\mathbf{p}^{(l)}) = \Theta(l)$.

**Theorem B.5.** *For any nonlinearity $\phi$ in an RRN, under assumptions Axiom 3.1 and Axiom 3.2, whenever $\dot{\phi}^2(\zeta)$ has finite variance for Gaussian variable $\zeta$,*

$$\underline{\boldsymbol{\chi}} = (\sigma_w^2 \mathrm{V}\dot{\phi}(\mathbf{q}) + 1)\boldsymbol{\chi}, \qquad \boldsymbol{\chi}_b = \boldsymbol{\chi}\mathrm{V}\dot{\phi}(\mathbf{q}), \qquad \boldsymbol{\chi}_w = \boldsymbol{\chi}\mathrm{V}\dot{\phi}(\mathbf{q})\underline{\mathbf{p}}.$$

**Theorem B.6.** *For $\phi = \tanh$ in an RRN,*

- If $\sigma_w = 0$, $\boldsymbol{\chi}^{(m)} = \boldsymbol{\chi}^{(l)}$ for all $l, m$.

- If $\sigma_w > 0$,

$$\log(\boldsymbol{\chi}^{(m)}/\boldsymbol{\chi}^{(l)}) = \mathcal{A}(\sqrt{l} - \sqrt{m}) + \mathcal{B}(\log l - \log m) + O(1)$$

where $\mathcal{A} = \frac{4}{3}\sqrt{\frac{2}{\pi}}\sigma_w$ and $\mathcal{B} = \frac{4}{3\pi} - \sigma_w^2 \frac{4}{9\pi}$.

**Theorem B.7.** *Suppose $\phi = \tanh$. Then in an RRN*

- If $\sigma_w = 0$, $\boldsymbol{\chi}_b^{(l)} = \boldsymbol{\chi}^{(L)}\mathrm{V}\dot{\phi}(\sigma_b^2)$ and $\boldsymbol{\chi}_w^{(l)} = \boldsymbol{\chi}^{(L)}\mathrm{V}\dot{\phi}(\sigma_b^2)((l-1)\mathrm{V}\phi(\sigma_b^2) + \mathbf{p}^{(0)})$, where $L$ is the last layer.

- If $\sigma_w > 0$,

$$\log(\boldsymbol{\chi}_b^{(m)}/\boldsymbol{\chi}_b^{(l)}) = \mathcal{A}(\sqrt{l} - \sqrt{m}) + \mathcal{B}_b(\log l - \log m) + O(1)$$
$$\log(\boldsymbol{\chi}_w^{(m)}/\boldsymbol{\chi}_w^{(l)}) = \mathcal{A}(\sqrt{l} - \sqrt{m}) + \mathcal{B}_w(\log l - \log m) + O(1)$$

where $\mathcal{A} = \frac{4}{3}\sqrt{\frac{2}{\pi}}\sigma_w$ (same as $\mathcal{A}$ in Thm B.6) and $\mathcal{B}_b = \mathcal{B} + \frac{1}{2}, \mathcal{B}_w = \mathcal{B} - \frac{1}{2}$, with $\mathcal{B} = \frac{4}{3\pi} - \sigma_w^2 \frac{4}{9\pi}$ (same as $\mathcal{B}$ in Thm B.6).

### B.1.2 Full Residual Network

**Theorem B.8.** *For any nonlinearity $\phi$ in an FRN,*

$$\mathbf{q} = \sigma_w^2\underline{\mathbf{p}} + \sigma_b^2$$
$$\mathbf{p} = \sigma_v^2\mathrm{V}\phi(\mathbf{q}) + \sigma_a^2 + \underline{\mathbf{p}}$$

**Theorem B.9.** *Suppose $\phi$ is tanh-like. Assume the FRN architecture.*

- If $\sigma_w = 0$, then $\mathbf{p}^{(l)} = (\sigma_v^2\mathrm{V}\phi(\sigma_b^2) + \sigma_a^2)l + \mathbf{p}^{(0)}$, and $\mathbf{q}^{(l)} = \sigma_b^2$.

- If $\sigma_w > 0$, then $\mathbf{p}^{(l)} = b_0 l + b_1 l^{1/2} + b_2 \log l + O(1)$, where

$$b_0 = \sigma_v^2 + \sigma_a^2$$
$$b_1 = \frac{-2C\sigma_v^2\sigma_w^{-1}}{\sqrt{\sigma_v^2 + \sigma_a^2}}$$
$$b_2 = \frac{-C^2\sigma_v^4\sigma_w^{-2}}{(\sigma_v^2 + \sigma_a^2)^2}$$

and $C = \sqrt{\frac{2}{\pi}}$. Additionally, $\mathbf{q}^{(l)} = \sigma_w^2 b_0 l + \sigma_w^2 b_1 l^{1/2} + \sigma_w^2 b_2 \log l + O(1)$.

**Theorem B.10.** *For any nonlinearity $\phi$, in an FRN*

$$\boldsymbol{\lambda} = \sigma_w^2\underline{\boldsymbol{\gamma}} + \sigma_b^2$$
$$\boldsymbol{\gamma} = \sigma_v^2\mathrm{W}\phi(\mathbf{q}, \boldsymbol{\lambda}) + \sigma_a^2 + \underline{\boldsymbol{\gamma}}$$

**Theorem B.11.** *Assume $\phi = \tanh$ in an FRN. Suppose $\mathbf{e}^{(0)} < 1$.*

- If $\sigma_w = 0$, then $\boldsymbol{\lambda}^{(l)} = \sigma_b^2$ and $\boldsymbol{\gamma}^{(l)} = l(\sigma_v^2\mathrm{W}\phi(\sigma_b^2, \sigma_b^2) + \sigma_a^2) + \boldsymbol{\gamma}^{(0)} = l(\sigma_v^2\mathrm{V}\phi(\sigma_b^2) + \sigma_a^2) + \boldsymbol{\gamma}^{(0)}$. Thus $\mathbf{e}^{(l)} \to 1$ and $1 - \mathbf{e}^{(l)} = \Theta(l^{-1})$. As a result, $\mathbf{s}^{(l)} = \mathbf{p}^{(l)}(1 - \mathbf{e}^{(l)}) = \Theta(1)$.

**Figure B.3:** Empirical verification of Thm B.9.

- *If $\sigma_w > 0$, then $\mathbf{e}^{(l)}$ converges to the unique fixed point $\mathbf{e}^* \neq 1$ determined by the equation*

$$\mathbf{e}^* = \frac{1}{\sigma_v^2 + \sigma_a^2}[\sigma_v^2 \frac{2}{\pi} \arcsin{(\mathbf{e}^*)} + \sigma_a^2].$$

*Furthermore, $\mathbf{e}^{(l)}$ converges to $\mathbf{e}^*$ polynomially: $|\mathbf{e}^{(l)} - \mathbf{e}^*|$ is $\check{\Theta}(l^{-\delta^*})$, where*

$$\delta^* := 1 - \frac{2}{\pi} \frac{1}{\sqrt{1 - (\mathbf{e}^*)^2}} \frac{\sigma_v^2}{\sigma_v^2 + \sigma_a^2} \in [\frac{2}{\pi} - 1, \frac{1}{2})$$

*Since $\mathbf{e}^* < 1$, $\mathbf{s}^{(l)} = \Theta(\mathbf{p}^{(l)}) = \Theta(l)$.*

**Theorem B.12.** *For any nonlinearity $\phi$ in an FRN, under assumptions Axiom 3.1 and Axiom 3.2, whenever $\dot{\phi}(\zeta)^2$ has finite variance for Gaussian variable $\zeta$,*

$$\underline{\chi} = (\sigma_v^2 \sigma_w^2 \mathrm{V}\dot{\phi}(\mathbf{q}) + 1)\chi, \qquad \chi_b = \sigma_v^2 \chi \mathrm{V}\dot{\phi}(\mathbf{q}),$$
$$\chi_w = \sigma_v^2 \chi \mathrm{V}\dot{\phi}(\mathbf{q})\underline{\mathbf{p}}, \qquad \chi_v = \chi \mathrm{V}\phi(\mathbf{q}), \qquad \chi_a = \chi$$

**Theorem B.13.** *Assume $\phi = \tanh$ in an FRN.*

- *If $\sigma_w = 0$, $\chi^{(m)} = \chi^{(l)}$ for all $l, m$.*

- *If $\sigma_w > 0$, then for $l \geq m \geq 0$,*

$$\log(\chi^{(m)}/\chi^{(l)}) = \mathcal{A}(\sqrt{l} - \sqrt{m}) + \mathcal{B}(\log l - \log m) + O(1)$$

*where*

$$\mathcal{A} = \frac{4}{3}\sqrt{\frac{2}{\pi}} \frac{\sigma_v^2 \sigma_w}{\sqrt{\sigma_v^2 + \sigma_a^2}}$$
$$\mathcal{B} = \frac{4}{9\pi} \frac{\sigma_v^4}{\sigma_v^2 + \sigma_a^2} \left(\frac{3}{\sigma_v^2 + \sigma_a^2} - \sigma_w^2\right)$$

Fig. B.4 shows empirical verification of the asymptotic expansion of $\chi$ for various values of $\sigma_\bullet$s.

**Theorem B.14.** *Suppose $\phi = \tanh$ in an FRN.*

- *If $\sigma_w = 0$, then*

$$\chi_b^{(l)} = \sigma_v^2 \chi^{(L)} \mathrm{V}\dot{\phi}(\sigma_b^2)$$
$$\chi_w^{(l)} = \sigma_v^2 \chi^{(L)} \mathrm{V}\dot{\phi}(\sigma_b^2)((\sigma_v^2 \mathrm{V}\phi(\sigma_b^2) + \sigma_a^2)(l-1) + \mathbf{p}^{(0)})$$
$$\chi_v^{(l)} = \chi^{(L)} \mathrm{V}\phi(\sigma_b^2)$$
$$\chi_a^{(l)} = \chi^{(L)}.$$

- *If $\sigma_w > 0$, then for $l \geq m \geq 0$,*

**Figure B.4:** Empirical verification of the asymptotic expansion of $\boldsymbol{\chi}$ for various values of $\sigma_\bullet$s. Note that we have chosen all small values for $\sigma_\bullet$s. For larger values, the constant term in Thm B.13 begins to dominate (primarily because of the expansion $\log(1+x) = x + \Theta(x^2)$ has large $\Theta$ term when $x$ is large), and $\boldsymbol{\chi}$ behaves more like $\exp(\Theta(l))$ up to depth 1000.

$$\log(\boldsymbol{\chi}_b^{(m)}/\boldsymbol{\chi}_b^{(l)}) = \mathcal{A}(\sqrt{l} - \sqrt{m}) + \mathcal{B}_b(\log l - \log m) + O(1)$$
$$\log(\boldsymbol{\chi}_w^{(m)}/\boldsymbol{\chi}_w^{(l)}) = \mathcal{A}(\sqrt{l} - \sqrt{m}) + \mathcal{B}_w(\log l - \log m) + O(1)$$
$$\log(\boldsymbol{\chi}_a^{(m)}/\boldsymbol{\chi}_a^{(l)}) = \mathcal{A}(\sqrt{l} - \sqrt{m}) + \mathcal{B}(\log l - \log m) + O(1)$$
$$\log(\boldsymbol{\chi}_v^{(m)}/\boldsymbol{\chi}_v^{(l)}) = \mathcal{A}(\sqrt{l} - \sqrt{m}) + \mathcal{B}(\log l - \log m) + O(1)$$

*where $\mathcal{A} = \frac{4}{3}\sqrt{\frac{2}{\pi}}\frac{\sigma_v^2\sigma_w}{\sqrt{\sigma_v^2+\sigma_a^2}}$ and $\mathcal{B} = \frac{4}{9\pi}\frac{\sigma_v^4}{\sigma_v^2+\sigma_a^2}\left(\frac{3}{\sigma_v^2+\sigma_a^2} - \sigma_w^2\right)$ are as in Thm B.13 and $\mathcal{B}_b = \mathcal{B} + \frac{1}{2}$ and $\mathcal{B}_w = \mathcal{B} - \frac{1}{2}$.*

## B.2 $\alpha$-ReLU

**Lemma B.15.** *If $\alpha > -\frac{1}{2}$, then $V\psi_\alpha(q) = \mathsf{c}_\alpha q^\alpha$, where $\mathsf{c}_\alpha = \frac{1}{\sqrt{\pi}}2^{\alpha-1}\Gamma\left(\alpha + \frac{1}{2}\right)$.*

Note that if $\alpha \leq -\frac{1}{2}$, then $V\psi_\alpha(q)$ is not defined (its defining integral does not converge).

### B.2.1 Full Residual Network

By Thm B.8 and Lemma B.15, we have the length recurrences

$$\mathbf{q} = \sigma_w^2\underline{\mathbf{p}} + \sigma_b^2$$
$$\mathbf{p} = \sigma_v^2\mathsf{c}_\alpha\mathbf{q}^\alpha + \sigma_a^2 + \underline{\mathbf{p}}$$

**Theorem B.16.** *Suppose we have the nonlinearity $\phi = \psi_\alpha$. The in an FRN: If $\alpha = 1$, then $\mathbf{p}^{(l)} = \Theta((1 + \sigma_v^2\sigma_w^2/2)^l)$, with the hidden constant depending on the initial condition. If $0 < \alpha < 1$, then $\mathbf{p}^{(l)} = \Theta(l^{\frac{1}{1-\alpha}})$. More precisely, $\lim_{l\to\infty}\mathbf{p}/l^{\frac{1}{1-\alpha}} = [\sigma_v^2\sigma_w^{2\alpha}\mathsf{c}_\alpha(1-\alpha)]^{\frac{1}{1-\alpha}}$.*

Fig. B.5 empirically verifies the asymptotics for $\alpha = 1$ for various $\sigma_v$ and $\sigma_w$.

Similarly, by Thm B.10, if $\mathbf{q} = \mathbf{q}'$, then

$$\boldsymbol{\lambda} = \sigma_w^2\underline{\boldsymbol{\gamma}} + \sigma_b^2$$
$$\boldsymbol{\gamma} = \sigma_v^2\mathbf{q}^\alpha W\psi_\alpha(1, \mathbf{c}) + \sigma_a^2 + \underline{\boldsymbol{\gamma}}$$

**Figure B.5:** Verification of the exponential asymptotics of $\mathbf{p}^{(l)}$ when $\alpha = 1$. The lines of each color correspond to different $(\sigma_w, \sigma_v)$ pairs, which are given in the legend. The solid lines are given by the recurrences Thm B.8, and the dashed lines are given by our asymptotics $(1 + \sigma_v^2 \sigma_w^2/2)^l$ (Thm B.16). Note that the y-axis is in log-scale.

**Figure B.6:** Verification of fixed point $\mathbf{e}^*$ in Thm B.18 for $\alpha = .6$. Different colors correspond to different initial conditions $\mathbf{e}^{(0)}$, and the dashed line gives the fixed point.

**Theorem B.17.** *Suppose $\phi = \psi_1$. Then in an FRN, $\mathbf{e}^{(l)} \to 1$ and $1 - \mathbf{e}^{(l)} \sim [\frac{1}{4}\sigma_v^2 \sigma_w^2 B^{-1} U l]^{-2}$ for $B = 1 + \sigma_v^2 \sigma_w^2/2$ and $U = \frac{2\sqrt{2}}{3\pi}$. As a result, $\mathbf{s}^{(l)} = (1 - \mathbf{e}^{(l)})\mathbf{p}^{(l)} = \Theta(l^{-2}\exp(\Theta(l))) = \exp(\Theta(l))$.*

**Theorem B.18.** *Suppose $\phi = \psi_\alpha$ for $0 < \alpha < 1$ in an FRN. Then $\mathbf{e}$ converges to the unique nonunit fixed point $\mathbf{e}^*$ of $\mathbb{J}_\alpha$, and $|\mathbf{e}^* - \mathbf{e}^{(l)}|$ is $\check{\Theta}(l^{-\mu})$, where $\mu = (1 - \dot{\mathbb{J}}_\alpha(\mathbf{e}^*))/(1 - \alpha)$. Additionally, $\mathbf{s}^{(l)} = \Theta(\mathbf{p}^{(l)}) = \Theta(l^{1/(1-\alpha)})$.*

Fig. B.6 verifies empirically that $\mathbf{e}^*$ is indeed the fixed point of $\mathbf{e}^{(l)}$. Fig. A.2 verifies empirically the convergence rate $l^{-\mu}$. Fig. B.7 plots $\dot{\mathbb{J}}_\alpha(\mathbf{e}^*)$ and $\mu$ versus $\alpha$. It certainly looks like $\mu = \frac{1}{2}(1 - \alpha)$, but we have no proof for it. Based on this conjecture, we see there is a "discontinuity" of $\mu$ at $\alpha = 1$: $\mu \to 0$ as $\alpha \to 1$, but for $\alpha = 1$, the actual convergence dynamics has exponent $-2$ by Thm B.17.

Because of the following theorem, we cannot expect the equations of Thm B.12 to hold for $\alpha \leq \frac{3}{4}$.

**Theorem B.19.** *Suppose we have the nonlinearity $\psi_\alpha$ in an FRN. $\mathrm{Var}(\dot{\psi}_\alpha(\zeta)^2)$ diverges for any Gaussian variable $\zeta$ with mean 0 if $\alpha \leq \frac{3}{4}$ but is finite if $\alpha > \frac{3}{4}$.*

**Theorem B.20.** *Suppose we have the nonlinearity $\psi_\alpha$ in an FRN. If $\alpha = 1$, then $\chi^{(l-m)} = \chi^{(l)}\left(\frac{1}{2}\sigma_v^2\sigma_w^2 + 1\right)^m$. If $\alpha \in (\frac{3}{4}, 1)$, then $\chi^{(l-m)} = \Theta(1)\chi^{(l)}(l/(l-m))^R$ for $R = \frac{\alpha^2}{(1-\alpha)(2\alpha-1)}$, where the constants in $\Theta(1)$ do not depend on $l$ or $m$.*

This exponent $\frac{\alpha^2}{(1-\alpha)(2\alpha-1)}$ is minimized at $\alpha = \frac{3}{4}$ on $\alpha \in (3/4, 1)$, where the value is $\frac{9}{2}$ (and at $\alpha = \frac{2}{3}$ on $\alpha \in (1/2, 1)$, where the value achieved is 4) (Fig. B.8(a)).

As a corollary,

**Figure B.7: (a)** A plot of $\dot{\mathbb{J}}_\alpha(\mathbf{e}^*)$ versus $\alpha$. **(b)** A plot of the exponent $\mu$ of the dynamics of $|\mathbf{e}^* - \mathbf{e}^{(l)}|$ (see Thm B.18)

**Figure B.8: (a)** The exponent of the polynomial gradient dynamics with respect to $\alpha$-ReLU versus $\alpha$. **(b)** The exponent of the dynamics of $\boldsymbol{\chi}_v$ and $\boldsymbol{\chi}_w$.

**Theorem B.21.** *If $\phi = \psi_1$ in an FRN, then for $l \geq m \geq 0$,*

$$
\begin{aligned}
\boldsymbol{\chi}_b^{(l-m)} &= \Theta(1)\boldsymbol{\chi}^{(l)}B^m, & \boldsymbol{\chi}_w^{(l-m)} &= \Theta(1)\boldsymbol{\chi}^{(l)}B^l, \\
\boldsymbol{\chi}_v^{(l-m)} &= \Theta(1)\boldsymbol{\chi}^{(l)}B^l, & \boldsymbol{\chi}_a^{(l-m)} &= \Theta(1)\boldsymbol{\chi}^{(l)}B^m.
\end{aligned}
$$

*where $B = 1 + \sigma_v^2\sigma_w^2/2$.*

*If $\phi = \psi_\alpha$ in an FRN, for $\alpha < 1$, then for $l \geq m \geq 0$,*

$$
\begin{aligned}
\boldsymbol{\chi}_b^{(l-m)} &= \Theta(1)\boldsymbol{\chi}^{(l)}l^R(l-m)^{-R-1}, & \boldsymbol{\chi}_w^{(l-m)} &= \Theta(1)\boldsymbol{\chi}^{(l)}l^R(l-m)^{\frac{\alpha}{1-\alpha}-R}, \\
\boldsymbol{\chi}_v^{(l-m)} &= \Theta(1)\boldsymbol{\chi}^{(l)}l^R(l-m)^{\frac{\alpha}{1-\alpha}-R}, & \boldsymbol{\chi}_a^{(l-m)} &= \Theta(1)\boldsymbol{\chi}^{(l)}(l/(l-m))^R.
\end{aligned}
$$

Fig. A.2 verifies the backward asymptotic dynamics empirically for different $\alpha < 1$. Fig. B.8(b) graphs the exponent $\frac{\alpha}{1-\alpha} - R$ in terms of $\alpha$. We see that on $[0.5, 1]$, the maximum of this exponent is at $\alpha = 1$.

## C  Proofs

A brief note about notation: We use $\sim$ to denote both how a random variable is sampled (ex: $x \sim \mathcal{N}(0,1)$ for a Gaussian $x$) and how a function behaves asymptotically, i.e. $f(x) \sim g(x)$ as $x \to a$ iff $\lim_{x\to a} f(x)/g(x) = 1$. Context should be enough to differentiate between these two cases. We in addition use $\simeq$ to denote asymptotic expansion. For example, if $\{\alpha_i\}_{i\geq0}$ is a sequence of strictly decreasing reals and $\{\beta_i\}_{i\geq0}$ is a sequence of nonzero reals, then

$$
f(x) \simeq \sum_{i\geq0} \beta_i(x-\xi)^{\alpha_i}
$$

means that as $x \to \xi$, $f(x) - \sum_{i=0}^N \beta_i(x-\xi)^{\alpha_i} = \Theta((x-\xi)^{\alpha_{N+1}})$.

## C.1 Preliminary Lemmas

**Lemma C.1.** *We have*

$$\frac{\sigma_w^2 \boldsymbol{\gamma} + \sigma_b^2}{\sigma_w^2 \mathbf{p} + \sigma_b^2} = \mathbf{e}(1 + O(\boldsymbol{\gamma}^{-1})).$$

*regardless of whether $\mathbf{e}^{(l)} = \boldsymbol{\gamma}^{(l)}/\mathbf{p}^{(l)}$ converges.*

*But suppose $\mathbf{e}^{(l)} = \boldsymbol{\gamma}^{(l)}/\mathbf{p}^{(l)} \to \mathbf{e}^*$. If $\mathbf{e}^* < 1$, then*

$$\frac{\sigma_w^2 \boldsymbol{\gamma} + \sigma_b^2}{\sigma_w^2 \mathbf{p} + \sigma_b^2} = \mathbf{e}(1 + \Theta(\boldsymbol{\gamma}^{-1})).$$

*If $\mathbf{e}^* = 1$, then*

$$\frac{\sigma_w^2 \boldsymbol{\gamma} + \sigma_b^2}{\sigma_w^2 \mathbf{p} + \sigma_b^2} = \mathbf{e}(1 + \Theta(\epsilon \mathbf{p}^{-1})),$$

*where $\epsilon = 1 - \mathbf{e}$.*

*Proof.* Write $M = \sigma_b^2/\sigma_w^2$.

$$\frac{\sigma_w^2 \boldsymbol{\gamma} + \sigma_b^2}{\sigma_w^2 \mathbf{p} + \sigma_b^2} = \mathbf{e}(1 + \frac{1 + M\boldsymbol{\gamma}^{-1}}{1 + M\mathbf{p}^{-1}})$$
$$= \mathbf{e}(1 + M(\boldsymbol{\gamma}^{-1} - \mathbf{p}^{-1}) + O(\mathbf{p}^{-1}(\boldsymbol{\gamma}^{-1} - \mathbf{p}^{-1}))).$$

In any situation, $\boldsymbol{\gamma}^{-1} - \mathbf{p}^{-1} = O(\boldsymbol{\gamma}^{-1})$ because $\boldsymbol{\gamma} \leq \mathbf{p}$, so this gives the first statement. If $\mathbf{e}^*$ exists and $\mathbf{e}^* < 1$, then $\boldsymbol{\gamma}^{-1} - \mathbf{p}^{-1} = \Theta(\boldsymbol{\gamma}^{-1})$, which yields the second statement. If $\mathbf{e}^*$ exists and $\mathbf{e}^* = 1$, then $\boldsymbol{\gamma}^{-1} - \mathbf{p}^{-1} = \mathbf{p}^{-1}((1-\epsilon)^{-1} - 1) = \mathbf{p}^{-1}(\epsilon + O(\epsilon^2)) = \Theta(\epsilon \mathbf{p}^{-1})$. $\square$

For any function $f$ that is $(k+1)$-times differentiable in a neighborhood of $0$, we have the asymptotic expansion

$$f(z) = \sum_{n=0}^{k} \frac{d^n f}{dz^n}(0)\frac{z^n}{n!} + O(z^{k+1}), \text{ as } z \to 0.$$

Since

$$\frac{d^n}{d(1/q)^n} q^{1/2} \mathrm{V}\phi(q)\Big|_{q\to\infty} = \frac{(-1)^n}{2^n\sqrt{2\pi}} \int_{-\infty}^{\infty} \phi^2(z)z^{2n}\,\mathrm{d}z$$

whenever the RHS is integrable, we have

**Lemma C.2.** *Suppose $\phi^2(z)z^{2n}$ is integrable over $z \in \mathbb{R}$ for all $0 \leq n \leq N+1$. Then $\mathrm{V}\phi(q) = q^{-1/2}(\sum_{n=0}^{N} C_n q^{-n} + O(q^{-N-1}))$ as $q \to \infty$, where*

$$C_n := \frac{(-1)^n}{2^n n!\sqrt{2\pi}} \int_{-\infty}^{\infty} \phi^2(z)z^{2n}\,\mathrm{d}z.$$

Note that $\mathrm{sech}^d(z) = \Theta(e^{-d|z|})$ for $z \to \infty$ as long as $d > 0$, so that $C_n$ from the above result converges when $\phi = \mathrm{sech}^d$. Therefore

**Lemma C.3.** *Let $d > 0$. We have $\mathrm{V}\,\mathrm{sech}^d(q) \simeq q^{-1/2}\sum_{n\geq 0} C_n q^{-n}$, where*

$$C_n := \frac{(-1)^n}{2^n n!\sqrt{2\pi}} \int_{-\infty}^{\infty} \mathrm{sech}^{2d}(z)z^{2n}\,\mathrm{d}z.$$

As corollaries, we obtain the following asymptotics.

**Lemma C.4.** $\mathrm{V}\dot{\tanh}(q) = \frac{2}{3}\sqrt{\frac{2}{\pi}}q^{-1/2} + \Theta(q^{-3/2})$ *as $q \to \infty$.*

*Proof.* Use [Lemma C.3](#) along with the fact that $\dot{\tanh}(z) = \mathrm{sech}^2(z)$ and $\int \mathrm{sech}^4 z \,\mathrm{d}z = \frac{2}{3}\tanh z + \frac{1}{2}\mathrm{sech}^2 z \tanh z$. $\square$

**Lemma C.5.** $1 - \mathrm{V}\tanh(q) = \sqrt{\frac{2}{\pi}}q^{-1/2} + \Theta(q^{-3/2})$ *as* $q \to \infty$.

*Proof.* Use Lemma C.3 along with the fact that $1 - \tanh^2(z) = \operatorname{sech}^2(z)$ and $\int \operatorname{sech}^2 z \, dz = \tanh z$. $\qquad \square$

**Lemma C.6.** $\operatorname{sech}^2(t) \geq \exp(-t^2)$ *for all t, with equality iff* $t = 0$.

*Proof.* The lower bound is equivalent to

$$2 \geq e^{t-t^2/2} + e^{-t-t^2/2}$$

The RHS has derivative $(1-t)e^{t-t^2/2} - (1+t)e^{-t-t^2/2}$. This is 0 iff

$$\frac{1-t}{1+t} = e^{-2t}$$

which has a solution 0 and in general can only have solution $t \in (-1, 1)$ (by considering the sign of the LHS). Since each side is analytic in $t \in (-1, 1)$, we expand

$$\log \frac{1-t}{1+t} = \log e^{-2t}$$
$$\log(1-t) - \log(1+t) = -2t$$
$$(-t - t^2 - \cdots) - (t - t^2 + \cdots) = -2t$$
$$-2t - 2t^3 - \cdots = -2t$$

which shows that the only solution is $t = 0$. A simple plot shows that $t = 0$ is a maximum, where the bound in question achieves equality.

$\qquad \square$

**Lemma C.7.** *Suppose* $\phi = \tanh$. *Then* $\mathrm{V}\dot{\phi}(q) \geq \frac{1}{\sqrt{4q+1}}$.

As a sanity check, Lemma C.4 shows that $\mathrm{V}\dot{\phi}(q) \sim C_0 q^{1/2}$ where $C_0 \approx .5319$, which is above the .5 in this lemma.

*Proof.* By Lemma C.6,

$$\mathrm{V}\dot{\phi}(q) = \int \mathrm{d}\mu(z)\dot{\phi}^2(\sqrt{q}z)$$

$$\geq \frac{1}{\sqrt{2\pi}} \int \mathrm{d}z \exp(-z^2/2 - 2qz^2)$$

$$= \frac{1}{\sqrt{2\pi}} \int \mathrm{d}z \exp(-(4q+1)z^2/2)$$

$$= \frac{1}{\sqrt{4q+1}}.$$

$\qquad \square$

Fig. C.9 demonstrates Lemma C.7.

**Lemma C.8.** *Let* $d \in \mathbb{R}$ *and* $1 < M < N$ *with* $N - M \in \mathbb{Z}^{\geq 0}$. *Set* $\Sigma(M, N, d) := \sum_{a=M}^{N} a^d$. *If we fix M and let* $N \to \infty$,

$$\Sigma(M, N, d) = \begin{cases} \Theta(1) & \text{if } d < -1 \\ \log N + O(1) & \text{if } d = -1 \\ \frac{N^{d+1}}{d+1} + O(1) & \text{if } -1 < d < 0 \\ N - M + 1 & \text{if } d = 0 \\ \frac{1}{d+1}N^{d+1} + \frac{1}{2}N^d + O(N^{\max(0,d-1)}) & \text{if } d > 0 \end{cases}$$

**Figure C.9:** Illustration of Lemma C.7: $V\dot{\phi}(q)$ vs $\frac{1}{\sqrt{4q+1}}$ for $\phi = \tanh$. This bound is very tight, and for most purposes, $\frac{1}{\sqrt{4q+1}}$ can be taken as a good approximation of $V\dot{\phi}(q)$.

*Proof.* Consider the integrals $A = \int_M^{N+1} a^d \, \mathrm{d}a$ and $B = \int_{M-1}^N a^d \, \mathrm{d}a$. They evaluate to $A = \frac{1}{d+1}((N+1)^{d+1} - M^{d+1})$ and $B = \frac{1}{d+1}(N^{d+1} - (M-1)^{d+1})$ when $d \neq -1$ and to $A = \log(N+1) - \log M$ and $B = \log N - \log(M-1)$ when $d = -1$. When $d \leq 0$, we have $A \leq B$ and $\Sigma(M, N, d) \in [A, B]$; when $d > 0$, $B \leq A$ and $\Sigma(M, N, d) \in [B, A]$. Thus, as $N \to \infty$ with $M$ fixed, when $d < -1$, $\Sigma(M, N, d) = \Theta(1)$; when $d = -1$, $\Sigma(M, N, -1) = \log N + O(1)$; and when $d > -1$, we have $\Sigma(M, N, d) = \frac{N^{d+1}}{d+1} + O(N^d)$.

Now for $a > 0$ and $d > -1$ and $d \neq 0, 1$,

$$\int_a^{a+1} z^d - a^d \, \mathrm{d}z = \frac{1}{d+1}((a+1)^{d+1} - a^d)$$

$$= (a^d + \frac{d}{2}a^{d-1} + \cdots) - a^d$$

$$= \frac{d}{2}a^{d-1} + \Theta(a^{d-2}).$$

where the hidden constants in $\Theta$ depend only on $d$ (and in fact this term vanishes if $d = 1$). Thus

$$\Sigma(M, N, d) = \int_M^{N+1} z^d \, \mathrm{d}z - \sum_{a=M}^N [\frac{d}{2}a^{d-1} + \Theta(a^{d-2})]$$

$$= \frac{1}{d+1}((N+1)^{d+1} - M^{d+1}) - \frac{d}{2}\Sigma(M, N, d-1) + \Theta(\Sigma(M, N, d-2))$$

If $-1 < d < 0$, then $\Sigma(M, N, d-1) = \Theta(1)$, so that $\Sigma(M, N, d) = \frac{(N+1)^{d+1}}{d+1} + O(1) = \frac{N^{d+1}}{d+1} + O(1)$. If $d > 0$ and $d \neq 1$, then $\Sigma(M, N, d-1) = \frac{N^d}{d}$, so that

$$\Sigma(M, N, d) = \frac{1}{d+1}N^{d+1} + N^d + \Theta(N^{\max(0,d-1)}) - \frac{1}{2}N^d + \Theta(\Sigma(M, N, d-2))$$

$$= \frac{1}{d+1}N^{d+1} + \frac{1}{2}N^d + O(N^{\max(0,d-1)}).$$

□

We can obtain more terms in the expansion for higher $d$ via the Euler-Maclaurin formula, but this suffices for our purposes.

## C.2   Dynamics Zoo

This section deduces the asymptotic behaviors of some sequences governed by recurrence equations. For the most part, the leading term of their asymptotic expansions is as one would expect from the corresponding differential equation. However, in some cases we need subleading terms for later results. They require slightly more nuanced reasoning. First we present a technical lemma.

**Lemma C.9.** *Let $F : \mathbb{R} \times \mathbb{N} \to \mathbb{R}$ be a function such that for a subset $U \subseteq \mathbb{R}$, and for all $z, z' \in U, z \geq z' \implies F(z, n) \geq F(z', n)$ for every $n$. Suppose sequences $a^{(l)}, b^{(l)}, c^{(l)}$ satisfy*

- $a^{(l+1)} = F(a^{(l)}, l)$ *for all l;*

- $b^{(l+1)} \leq F(b^{(l)}, l)$ *for all l above a constant $K_b$.*

- $c^{(l+1)} \geq F(c^{(l)}, l)$ *for all l above a constant $K_c$.*

*and furthermore, $a^{(l)}, b^{(l)}, c^{(l)}$ all fall into U for l above a constant $K_U$.*

*If for some $m \geq \max(K_b, K_U)$, $b^{(m)} \leq a^{(m)}$, then $b^{(l)} \leq a^{(l)}, \forall l \geq m$. Similarly, if for some $n \geq \max(K_c, K_U)$, $c^{(n)} \geq a^{(n)}$, then $c^{(l)} \geq a^{(l)}, \forall l \geq n$.*

*Proof.* For the first claim: $b^{(m)} \leq a^{(m)} \implies b^{(m+1)} \leq F(b^{(m)}, m) \leq F(a^{(m)}, m) = a^{(m+1)}$. Here the last inequality used the monotonicity of $F$. Induction gives the desired result.

It's similar for the second claim, where the inductive step is $c^{(m)} \geq a^{(m)} \implies c^{(m+1)} \geq F(c^{(m)}, m) \geq F(a^{(m)}, m) = a^{(m+1)}$. $\square$

**Lemma C.10.** *Suppose $\epsilon^{(l)}$ satisfies the recurrence*

$$\epsilon^{(l)} = \epsilon^{(l-1)}(1 + \frac{\delta}{l^\beta}).$$

*for some nonzero constant $\delta \in \mathbb{R}$ independent of l.*

- *If $\beta > 1$, then $\epsilon^{(l)} = \Theta(1)$.*

- *If $\beta = 1$, then $\epsilon^{(l)} = \Theta(l^\delta)$.*

- *If $0 < \beta < 1$, then $\epsilon^{(l)} = \exp(\frac{\delta}{1-\beta}l^{1-\beta} + \tilde{\Theta}(l^{\psi_1(1-2\beta)}))$, where $\psi_1(x) = \max(0, x)$ is the ReLU function.*

*Proof.* We have

$$\log \epsilon^{(l)} = \log \epsilon^{(l-1)} + \log(1 + \delta/l^\beta)$$
$$= \log \epsilon^{(l-1)} + \delta/l^\beta + \Theta(\delta^2/l^{2\beta})$$

for large l. If $\beta > 1$, then $\sum_l l^{-\beta}$ converges, and

$$\log \epsilon^{(l)} = \log \epsilon^{(0)} - \Theta(1)$$
$$\epsilon^{(l)} = \Theta(1).$$

If $\beta = 1$, then

$$\log \epsilon^{(l)} = \log \epsilon^{(0)} + \delta \log l + \Theta(1)$$
$$\epsilon^{(l)} = \Theta(l^\delta).$$

If $\beta < 1$, then

$$\log \epsilon^{(l)} = \log \epsilon^{(0)} + \frac{\delta}{1-\beta}l^{1-\beta} + \tilde{\Theta}(l^{1-2\beta})$$

$$\epsilon^{(l)} = \exp(\frac{\delta}{1-\beta}l^{1-\beta} + \tilde{\Theta}(l^{\psi_1(1-2\beta)})).$$

$\square$

**Lemma C.11.** *Suppose $\epsilon^{(l)} = Cl^{-\alpha} + \epsilon^{(l-1)}(1 + \delta/l^\beta)$ for $\alpha \in \mathbb{R}$, $C \neq 0$, and $\delta \neq 0$. Then*

- *If $\beta > 1$, then*

  - $\epsilon^{(l)} = \Theta(l^{1-\alpha})$ *if $\alpha \in (0, 1)$;*
  - $\epsilon^{(l)} = \Theta(\log l)$ *if $\alpha = 1$;*
  - $\epsilon^{(l)} = \Theta(1)$ *if $\alpha > 1$.*

- *If $\beta = 1$, then*
    - $\epsilon^{(l)} = \Theta(l^{\max(\delta, 1-\alpha)})$ *if* $1 - \delta \neq \alpha$.
    - $\epsilon^{(l)} = \Theta(l^{\delta} \log l)$ *if* $1 - \delta = \alpha$.

*Furthermore, for $\beta = -\delta = 1$, $\epsilon^{(l)} \sim l^{-1}$ if $\alpha > 2$, $\epsilon^{(l)} \sim l^{1-\alpha}$ if $\alpha < 2$, and $\epsilon^{(l)} \sim l^{\delta} \log l$ if $\alpha = 2$.*

*Proof.* We can unwind the recurrence to get

$$\epsilon^{(l)} = \sum_{m=1}^{l} m^{-\alpha} \prod_{n=m+1}^{l} \left(1 + \frac{\delta}{n^{\beta}}\right) + \epsilon^{(0)} \prod_{n=1}^{l} \left(1 + \frac{\delta}{n^{\beta}}\right)$$

Suppose $\beta > 1$. By Lemma C.10, we get

$$\epsilon^{(l)} = \Theta(1) \sum_{m=1}^{l} m^{-\alpha} + \epsilon^{(0)} \Theta(1)$$
$$= \begin{cases} \Theta(l^{1-\alpha}) & \text{if } \alpha \in (0,1) \\ \Theta(\log l) & \text{if } \alpha = 1 \\ \Theta(1) & \text{if } \alpha > 1. \end{cases}$$

Now suppose $\beta = 1$. By Lemma C.10, we get

$$\epsilon^{(l)} = \sum_{m=1}^{l} m^{-\alpha} \Theta(m^{-\delta} l^{\delta}) + \epsilon^{(0)} \Theta(l^{\delta})$$

where the constants hidden inside the $\Theta$ are the same in every term of the sum. If $\alpha > 1 - \delta$, then $m^{-\delta-\alpha} = o(m^{-1})$, so that $\sum_{m=1}^{l} m^{-\delta-\alpha} = \Theta(1)$, and

$$\epsilon^{(l)} = \Theta(l^{\delta}) + \epsilon^{(0)} \Theta(l^{\delta})$$
$$= \Theta(l^{\delta}).$$

On the other hand, if $\alpha < 1 - \delta$, then $\sum_{m=1}^{l} m^{-\delta-\alpha} = \Theta(l^{1-\delta-\alpha})$. So

$$\epsilon^{(l)} = \Theta(l^{1-\alpha}) + \epsilon^{(0)} \Theta(l^{\delta})$$
$$= \Theta(l^{1-\alpha}).$$

If $\alpha = 1 - \delta$, then $\sum_{m=1}^{l} m^{-\delta-\alpha} = \Theta(\log l)$. So

$$\epsilon^{(l)} = \Theta(l^{\delta} \log l) + \epsilon^{(0)} \Theta(l^{\delta})$$
$$= \Theta(l^{\delta} \log l).$$

Finally, if $\beta \in (0,1)$, then

$$\epsilon^{(l)} = e^{\frac{\delta}{1-\beta} l^{1-\beta} + \Theta(l^{1-2\beta})} \sum_{m=1}^{l} m^{-\alpha} e^{\frac{-\delta}{1-\beta} m^{1-\beta} + \Theta(m^{1-2\beta})} + e^{\frac{\delta}{1-\beta} l^{1-\beta} + \Theta(l^{1-2\beta})}$$

The case of $\delta = -1$ telescopes, so that the upper and lower constants hidden in $\Theta$ can both be taken to be 1. $\qquad \square$

**Lemma C.12.** *Suppose for some $\beta > 0$, a sequence $\epsilon^{(l)}$ satisfies*

$$\epsilon^{(l)} = \epsilon^{(l-1)} (1 - \mu(\epsilon^{(l-1)})^{\beta}/l), \quad \epsilon^{(0)} \in (0, \frac{1}{\mu}).$$

*Then $\epsilon^{(l)} \sim (\beta \mu \log l)^{-1/\beta}$.*

*Proof.* Consider the differential equation

$$\dot{x}_\mu = -\mu x_\mu^{\beta+1}/t$$

for constant $\mu$ has solution $x_\mu = [\beta(\mu \log t + C)]^{-1/\beta}$ for some constant $C$ determined by initial condition. Note that

$$-\mu x_\mu(t)^{\beta+1}/t \le x_\mu(t+1) - x_\mu(t) \le -\mu x_\mu(t+1)^{\beta+1}/(t+1) = -(1-o(t^{-1}))\mu x_\mu(t)^{\beta+1}/t.$$

For any small enough $\alpha > 0$, we apply Lemma C.9 with $F(\epsilon, l) = \epsilon - \mu\epsilon^{\beta+1}/l$ (which is monotonic in $\epsilon$ for small enough $\epsilon$), $c^{(l)} = x_\mu(l)$, and $b^{(l)} = x_{\mu-\alpha}(l)$ to obtain

$$x_{\mu-\alpha}(l) \le \epsilon^{(l)} \le x_\mu(l)$$

for large enough $l$ and appropriately chosen initial conditions. This shows that $\epsilon^{(l)} = \Theta(\log l^{-1/\beta})$ Taking $\alpha \to 0$, we also obtain the leading coefficient $\epsilon^{(l)} \sim [\beta\mu \log l]^{-1/\beta}$.

$\square$

**Lemma C.13.** *Suppose a sequence $u^{(l)}$ is governed by the equation*

$$u^{(l)} - u^{(l-1)} = A(u^{(l-1)} + B)^\alpha,$$

*where $\alpha \in [0,1)$ and $A > 0$. Then $u^{(l)} = K_1 l^{\frac{1}{1-\alpha}} - K_2 l^{\frac{\alpha}{1-\alpha}} \log l + o(l^{\frac{\alpha}{1-\alpha}} \log l)$, where $K_1 = [A(1-\alpha)]^{\frac{1}{1-\alpha}}$ and $K_2 = \frac{1}{2}A^{\frac{1}{1-\alpha}}(1-\alpha)^{\frac{\alpha}{1-\alpha}-1}\alpha$.*

*Proof.* **Leading term.** The differential equation

$$\dot{x}_{A,B} = A(x_{A,B} + B)^\alpha$$

has solution $x_{A,B}(l) = [A(1-\alpha)(l+S)]^{\frac{1}{1-\alpha}} - B$ for some constant $S$. Since $\dot{x}_{A,B}$ is monotonic, we have (writing $x = x_{A,B}$ for brevity)

$$A(x_{A,B}(l) + B)^\alpha = \dot{x}_{A,B}(l) \le x_{A,B}(l+1) - x_{A,B}(l) \le \dot{x}_{A,B}(l+1) \le (A+o(1))(x_{A,B}(l)+B)^\alpha$$

for large enough $l$. We apply Lemma C.9 with $F(x,l) = x + A(x+B)^\alpha$ (which is monotonic in $x$ for large $x$), $c^{(l)} = x_{A,B}(l)$, and $b^{(l)} = x_{A-\epsilon,B}(l)$ to obtain

$$x_{A-\epsilon,B}(l) \le u^{(l)} \le x_{A,B}(l)$$

for large enough $l$ and appropriate initial conditions. Therefore $\lim u^{(l)}/l^{\frac{1}{1-\alpha}} \in [[(A-\epsilon)(1-\alpha)]^{\frac{1}{1-\alpha}}, [A(1-\alpha)]^{\frac{1}{1-\alpha}}]$. Taking $\epsilon \to 0$ gives the leading term.

**Subleading term.** Now let $v^{(l)} := u^{(l)} - \aleph l^{\frac{1}{1-\alpha}}$, where $\aleph = [A(1-\alpha)]^{\frac{1}{1-\alpha}}$. Then we have the recurrence

$$v^{(l+1)} + \aleph(l+1)^{\frac{1}{1-\alpha}} - v^{(l)} - \aleph l^{\frac{1}{1-\alpha}} = A(v^{(l)} + \aleph l^{\frac{1}{1-\alpha}} + B)^\alpha$$

$$v^{(l+1)} - v^{(l)} + \aleph\left(\frac{1}{1-\alpha}l^{\frac{\alpha}{1-\alpha}} + \frac{1}{2}\left(\frac{1}{1-\alpha}\right)\left(\frac{\alpha}{1-\alpha}\right)l^{\frac{\alpha}{1-\alpha}-1} + \Theta(l^{\frac{\alpha}{1-\alpha}-2})\right)$$

$$= A[\aleph^\alpha l^{\frac{\alpha}{1-\alpha}} + \alpha(v^{(l)} + B)\aleph^{\alpha-1}l^{-1} + \Theta((v^{(l)}+B)l^{-1-\frac{1}{1-\alpha}})]$$

$$v^{(l+1)} - v^{(l)} = \frac{\alpha}{1-\alpha}v^{(l)}l^{-1} - \frac{1}{2}\aleph\left(\frac{1}{1-\alpha}\right)\left(\frac{\alpha}{1-\alpha}\right)l^{\frac{\alpha}{1-\alpha}-1} + g(l)$$

for some $g(l) = O(l^{\frac{\alpha}{1-\alpha}-2} + l^{-1})$ and where, to get the last equation, we have used $A\alpha^\alpha = \frac{1}{1-\alpha}\aleph$ to cancel the $l^{\frac{\alpha}{1-\alpha}}$ term and simplified $\alpha A\aleph^{\alpha-1} = \frac{\alpha}{1-\alpha}$.

For any $J > 0$, the differential equation $\dot{v}_J(l) = \frac{\alpha}{1-\alpha}v_J(l)l^{-1} - Jl^{\frac{\alpha}{1-\alpha}-1}$ has solution $v_J(l) = C[l(1-\alpha)]^{\frac{\alpha}{1-\alpha}} - Jl^{\frac{\alpha}{1-\alpha}} \log l$. Note that the functions $F_J(z,n) = z + \frac{\alpha}{1-\alpha}zn^{-1} - Jn^{\frac{\alpha}{1-\alpha}-1}$ and $G_J(z,n) = F_J(z,n) + g(n)$ is monotonic in $z$ (for positive $n$). For large $l$, we also have $\dot{v}_J(l)$ and $F_J(v_J(l),l) = v_J(l) + \dot{v}_J(l)$ decreasing in $l$. Thus for any $\epsilon > 0$ and $l$ large enough

$$G_{J+\epsilon}(v_J(l),l) \le F_{J+\epsilon/2}(v_J(l),l) \le v_J(l)+\dot{v}_J(l+1) \le v_J(l+1) \le F_J(v_J(l),l) \le G_{J-\epsilon}(v_J(l),l).$$

Now apply Lemma C.9 with $F = G_K$, $a^{(l)} = v^{(l)}, c^{(l)} = v_{K-\epsilon}, b^{(l)} = v_{K+\epsilon}$ where $K := \frac{1}{2}\aleph(\frac{1}{1-\alpha})(\frac{\alpha}{1-\alpha}) = \frac{1}{2}A^{\frac{1}{1-\alpha}}(1-\alpha)^{\frac{\alpha}{1-\alpha}-1}\alpha$, with appropriately chosen initial conditions. This yields $\lim_{l\to\infty} v^{(l)}/(l^{\frac{\alpha}{1-\alpha}}\log l) \in [-K-\epsilon, -K+\epsilon]$ for every $\epsilon > 0$, and there it must be equal to $K$. We have thus obtained the asymptotic expansion

$$u^{(l)} = [A(1-\alpha)l]^{\frac{1}{1-\alpha}} - \frac{1}{2}A^{\frac{1}{1-\alpha}}(1-\alpha)^{\frac{\alpha}{1-\alpha}-1}\alpha l^{\frac{\alpha}{1-\alpha}}\log l + o(l^{\frac{\alpha}{1-\alpha}}\log l).$$

$\square$

**Lemma C.14.** *Suppose a sequence $u^{(l)}$ is governed by the equation*

$$u^{(l)} - u^{(l-1)} = -A(u^{(l-1)} + B)^\alpha,$$

*where $\alpha > 1$ and $A > 0$. Then $u^{(l)} \sim [A(\alpha-1)l]^{\frac{1}{1-\alpha}}$.*

*Proof.* Similar to Lemma C.13. $\square$

**Lemma C.15.** *Suppose a sequence $u^{(l)}$ is governed by the equation*

$$u^{(l)} - u^{(l-1)} = A(u^{(l-1)} + B)^\alpha + C,$$

*where $\alpha \in (0,1)$. Then $u^{(l)} = K_1 l^{\frac{1}{1-\alpha}} + R(l)$, where the remainder $R(l)$ is*

$$R(l) \sim \begin{cases} -K_2 l^{\frac{\alpha}{1-\alpha}}\log l & \text{if } \alpha > \frac{1}{2} \\ (C-K_2)l\log l & \text{if } \alpha = \frac{1}{2} \text{ and } K_2 \neq C \\ \frac{C(1-\alpha)}{1-2\alpha}l & \text{if } \alpha < \frac{1}{2} \end{cases}$$

*where $K_1 = [A(1-\alpha)]^{\frac{1}{1-\alpha}}, K_2 = \frac{1}{2}A^{\frac{1}{1-\alpha}}(1-\alpha)^{\frac{\alpha}{1-\alpha}-1}\alpha$ as in Lemma C.13.*

*Proof.* $u$ is bounded below by the dynamics $v^{(l)} - v^{(l-1)} = A(v^{(l-1)} + B)^\alpha$ and bounded above by the dynamics $w^{(l)} - w^{(l-1)} = (A + o(1))(w^{(l-1)} + B)^\alpha$. By Lemma C.13, both $v$ and $w$ are asymptotic to $u^{(l)} \sim [A(1-\alpha)l]^{\frac{1}{1-\alpha}}$, which gives the result.

Now define $v^{(l)} = u^{(l)} - [A(1-\alpha)l]^{\frac{1}{1-\alpha}}$, and similar to the proof of Lemma C.13, we find

$$v^{(l+1)} - v^{(l)} = \frac{\alpha}{1-\alpha}v^{(l)}l^{-1} - Kl^{\frac{\alpha}{1-\alpha}-1} + C + g(l)$$

where $K = \frac{1}{2}A^{\frac{1}{1-\alpha}}(1-\alpha)^{\frac{\alpha}{1-\alpha}-1}\alpha$ and $g(l) = O(l^{\frac{\alpha}{1-\alpha}-2} + l^{-1})$. If $\frac{\alpha}{1-\alpha} > 1 \iff \alpha > \frac{1}{2}$, then $C + g(l) = o(l^{\frac{\alpha}{1-\alpha}-1})$ and we can proceed as in the proof of Lemma C.13 to find $v^{(l)} \sim Kl^{\frac{\alpha}{1-\alpha}}\log l$. If $\frac{\alpha}{1-\alpha} = 1 \iff \alpha = 1$ and $K \neq C$, then $v^{(l+1)} - v^{(l)} = \frac{\alpha}{1-\alpha}v^{(l)}l^{-1} - (K-C)l^{\frac{\alpha}{1-\alpha}-1} + g(l)$, so that the technique used in Lemma C.13 would obtain $v^{(l)} \sim (K-C)l^{\frac{\alpha}{1-\alpha}}\log l = (K-C)l\log l$. If $\frac{\alpha}{1-\alpha} < 1 \iff \alpha < \frac{1}{2}$, then $v^{(l+1)} - v^{(l)} = \frac{\alpha}{1-\alpha}v^{(l)}l^{-1} + C + o(1)$, then by using the differential equation $\dot{v}_J(l) = \frac{\alpha}{1-\alpha}v_J(l)l^{-1} + J$ to approximate the difference equation solution and applying Lemma C.9 as in the proof of Lemma C.13, we obtain $v^{(l)}(l) \sim \frac{C(1-\alpha)}{1-2\alpha}l$. $\square$

### C.3 Forward Dynamical Equations

Here we derive the recurrences governing the forward length and correlation quantities $\mathbf{p}, \mathbf{q}, \boldsymbol{\lambda}, \boldsymbol{\gamma}$. We start with reduced residual networks.

**Lemma B.1.** *Suppose $\phi$ is antisymmetric. Then in an RRN, $\mathbf{p}$ and $\mathbf{q}$ satisfy the recurrence*

$$\mathbf{q} = \sigma_w^2 \underline{\mathbf{p}} + \sigma_b^2$$
$$\mathbf{p} = \mathrm{V}\phi(\mathbf{q}) + \underline{\mathbf{p}}.$$

*Proof.* We have

$$\mathbf{q} = \langle h_j^2 \rangle = \left\langle \sum_i (w_{ji}\underline{x}_i + b_j)^2 \right\rangle$$

$$= \langle b_j^2 \rangle + \sum_i \langle w_{ji}^2 \underline{x}_i^2 \rangle + 2\sum_i \langle w_{ji}\underline{x}_i b_j \rangle + 2\sum_{j \neq l} \langle w_{ji}w_{li}x_i^2 \rangle$$

But $w_{ji}, w_{li}, \underline{x}$, and $b_j$ form an independency, so the last two sums are 0, and the terms in the first sum split multiplicatively. Therefore

$$\mathbf{q} = \sigma_b^2 + \sum_i \langle w_{ji}^2 \rangle \langle \underline{x}_i^2 \rangle$$

$$= \sigma_b^2 + N \cdot \frac{\sigma_w^2}{N} \underline{\mathbf{p}}$$

$$= \sigma_b^2 + \sigma_w^2 \underline{\mathbf{p}}.$$

For the recurrence of $\mathbf{p}$, we have

$$\mathbf{p} = \langle x_i^2 \rangle = \langle (\phi(h_i) + \underline{x}_i)^2 \rangle$$

$$= \langle \phi(h_i)^2 \rangle + \langle \underline{x}_i^2 \rangle + 2\langle \phi(h_i)\underline{x}_i \rangle$$

As $N \to \infty$, the coefficient $w_{ii}$ of $\underline{x}_i$ in $h_i$ has vanishing covariance, so $h_i$ and $\underline{x}_i$ become independent. Therefore $\langle \phi(h_i)\underline{x}_i \rangle = \langle \phi(h_i) \rangle \langle \underline{x}_i \rangle$. Because $h_i$ is the sum of a large number of independent random variables, by CLT, $h_i$ is a Gaussian with mean $\sum_i \langle w_{ji} \rangle \langle \underline{x}_i \rangle + \langle b_j \rangle = 0$ since $\langle w_{ji} \rangle = \langle b_j \rangle = 0$. Our antisymmetry assumption on $\phi$ then implies $\langle \phi(h_i) \rangle = 0$. Therefore,

$$\mathbf{p} = \langle \phi(h_i)^2 \rangle + \langle \underline{x}_i^2 \rangle$$

$$= \mathrm{V}\phi(\mathbf{q}) + \underline{\mathbf{p}}$$

as desired. $\qquad\square$

**Theorem B.3.** *Suppose $\phi$ is antisymmetric. Then in an RRN, $\boldsymbol{\lambda}$ and $\boldsymbol{\gamma}$ satisfy the recurrence*

$$\boldsymbol{\lambda} = \sigma_w^2 \underline{\boldsymbol{\gamma}} + \sigma_b^2$$

$$\boldsymbol{\gamma} = \mathrm{W}\phi(\mathbf{q}, \boldsymbol{\lambda}) + \underline{\boldsymbol{\gamma}}.$$

*Proof.* Similar to <span style="color:red">Lemma B.1</span>. $\qquad\square$

Now, for the full residual networks, the proofs are similar, but we no longer need to assume that $\phi$ is antisymmetric because of the randomization via the extra sets of weights.

**Theorem B.8.** *For any nonlinearity $\phi$ in an FRN,*

$$\mathbf{q} = \sigma_w^2 \underline{\mathbf{p}} + \sigma_b^2$$

$$\mathbf{p} = \sigma_v^2 \mathrm{V}\phi(\mathbf{q}) + \sigma_a^2 + \underline{\mathbf{p}}$$

*Proof.*

$$\mathbf{q} = \langle h_j^2 \rangle = \langle (w_j^i \underline{x}_i + b_j)^2 \rangle = \langle (w_j^i \underline{x}_i)^2 \rangle + \langle b_j^2 \rangle$$

$$= \sigma_w^2 \langle \underline{x}_i^2 \rangle + \sigma_b^2$$

$$= \sigma_w^2 \underline{\mathbf{p}} + \sigma_b^2$$

$$\mathbf{p} = \langle x_i^2 \rangle = \langle (v_i^j \phi(h_j) + \underline{x}_i + a_i)^2 \rangle$$

$$= \sigma_v^2 \langle \phi(h_i)^2 \rangle + \langle \underline{x}_i^2 \rangle + \sigma_a^2$$

$$= \sigma_v^2 \mathrm{V}\phi(\mathbf{q}) + \sigma_a^2 + \underline{\mathbf{p}}$$

where in the third equality for $\mathbf{p}$, we are now using the independence of $v_i^j$ from all other variables to cancel out the terms, whereas before we had to rely on $\phi$ being antisymmetric. $\qquad\square$

**Theorem B.10.** *For any nonlinearity $\phi$, in an FRN*

$$\boldsymbol{\lambda} = \sigma_w^2 \underline{\boldsymbol{\gamma}} + \sigma_b^2$$
$$\boldsymbol{\gamma} = \sigma_v^2 W\phi(\mathbf{q}, \boldsymbol{\lambda}) + \sigma_a^2 + \underline{\boldsymbol{\gamma}}$$

*Proof.* Similar to Thm B.8. □

### C.4 Backward Dynamical Equations

Here we derive the recurrences governing the gradient quantities $\boldsymbol{\chi}$ and $\boldsymbol{\chi}_\bullet$ for different $\bullet$, all under the gradient independence assumption. Write $\beta_i^{(l)} = \frac{\partial E}{\partial x_i^{(l)}}$ for a cost function $E$.

**Theorem B.5.** *For any nonlinearity $\phi$ in an RRN, under assumptions Axiom 3.1 and Axiom 3.2, whenever $\dot{\phi}^2(\zeta)$ has finite variance for Gaussian variable $\zeta$,*

$$\underline{\boldsymbol{\chi}} = (\sigma_w^2 V\dot{\phi}(\mathbf{q}) + 1)\boldsymbol{\chi}, \qquad \boldsymbol{\chi}_b = \boldsymbol{\chi} V\dot{\phi}(\mathbf{q}), \qquad \boldsymbol{\chi}_w = \boldsymbol{\chi} V\dot{\phi}(\mathbf{q})\underline{\mathbf{p}}.$$

*Proof.* For a reduced residual network, we have the following derivative computation:

$$\frac{\partial x_i}{\partial \underline{x}_j} = \delta_{ji} + \dot{\phi}(h_i)\frac{\partial h_i}{\partial \underline{x}_j}, \quad \frac{\partial x_i}{\partial h_j} = \delta_{ji}\dot{\phi}(h_j), \quad \frac{\partial h_i}{\partial \underline{x}_j} = w_{ij}, \quad \frac{\partial h_i}{\partial w_{ij}} = \underline{x}_j, \quad \frac{\partial h_i}{\partial b_j} = \delta_{ij}.$$

Then

$$\underline{\beta}_j = \beta_j + \sum_i \beta_i \dot{\phi}(h_i)\frac{\partial h_i}{\partial \underline{x}_j}$$

$$= \beta_j + \sum_i \beta_i \dot{\phi}(h_i)w_{ij}$$

$$\langle \underline{\beta}_j^2 \rangle = \langle [\beta_j + \sum_i \beta_i \dot{\phi}(h_i)w_{ij}]^2 \rangle$$

$$= \langle \beta_j^2 \rangle + \sum_i \langle \beta_i^2 \dot{\phi}^2(h_i)(w_{ij})^2 \rangle$$

$$+ 2\sum_{i<k} \langle \beta_i \beta_k \dot{\phi}(h_i)w_{ij}\dot{\phi}(h_k)w_{kj} \rangle + 2\sum_i \langle \beta_j \beta_i \dot{\phi}(h_i)w_{ij} \rangle$$

The last two terms of the above vanish as $w_{ij}$ is independent from $w_{kj}$, $h_i, h_k$ and $\beta_i, \beta_j, \beta_k$ by Axiom 3.2, and $\langle w_{ij} \rangle = 0$.

Therefore, applying Axiom 3.1,

$$\langle \underline{\beta}_j^2 \rangle = \sigma_w^2 \langle \beta_j^2 \rangle \langle \dot{\phi}^2(h_i) \rangle + \langle \beta_j^2 \rangle$$

$$= (\sigma_w^2 V\dot{\phi}(\mathbf{q}) + 1)\langle \beta_j^2 \rangle$$

We similarly have

$$\frac{\partial E}{\partial b_j} = \sum_i \frac{\partial E}{\partial x_i}\frac{\partial x_i}{\partial h_j} = \beta_j \dot{\phi}(h_j), \qquad \text{since } \frac{\partial x_i}{\partial h_j} = \delta_{ji}\dot{\phi}(h_j)$$

$$\langle \left(\frac{\partial E}{\partial b_j}\right)^2 \rangle = \langle \beta_j^2 \dot{\phi}(h_j)^2 \rangle = \langle \beta_j^2 \rangle V\dot{\phi}(\mathbf{q}), \qquad \text{by Axiom 3.2(b);}$$

$$\frac{\partial E}{\partial w_{ji}} = \sum_i \frac{\partial E}{\partial x_i}\frac{\partial x_i}{\partial h_j}\frac{\partial h_j}{\partial w_{ji}} = \beta_j \dot{\phi}(h_j)\underline{x}_i, \qquad \text{since } \frac{\partial x_i}{\partial h_j} = \delta_{ji}\dot{\phi}(h_j)$$

$$\langle \left(\frac{\partial E}{\partial w_{ji}}\right)^2 \rangle = \langle \beta_j^2 \dot{\phi}^2(h_j)\underline{x}_i^2 \rangle = \langle \beta_j^2 \rangle V\dot{\phi}(\mathbf{q})\underline{\mathbf{p}}, \qquad \text{by Axiom 3.2(b)}$$

In the last equation we have also used the fact that as $N \to \infty$, $h_j$ and $x_i$ become independent (they are jointly Gaussian and their correlation $\langle w_{ji}^2 \rangle$ goes to 0 with $N$). □

**Theorem B.12.** *For any nonlinearity $\phi$ in an FRN, under assumptions Axiom 3.1 and Axiom 3.2, whenever $\dot\phi(\zeta)^2$ has finite variance for Gaussian variable $\zeta$,*

$$\underline{\chi} = (\sigma_v^2\sigma_w^2\mathrm{V}\dot\phi(\mathbf{q})+1)\chi, \qquad \chi_b = \sigma_v^2\chi\mathrm{V}\dot\phi(\mathbf{q}),$$

$$\chi_w = \sigma_v^2\chi\mathrm{V}\dot\phi(\mathbf{q})\underline{\mathbf{p}}, \qquad \chi_v = \chi\mathrm{V}\phi(\mathbf{q}), \qquad \chi_a = \chi$$

*Proof.* For the full residual network, we have the following derivative computations:

$$\frac{\partial x_i}{\partial \underline{x}_j} = \delta_{ji} + \sum_k v_{ik}\dot\phi(h_k)\frac{\partial h_k}{\partial \underline{x}_j}, \quad \frac{\partial x_i}{\partial h_j} = v_{ij}\dot\phi(h_j), \quad \frac{\partial h_i}{\partial \underline{x}_j} = w_{ij}, \quad \frac{\partial h_i}{\partial w_{ij}} = \underline{x}_j, \quad \frac{\partial h_i}{\partial b_i} = 1,$$

$$\frac{\partial x_i}{\partial v_{ik}} = \phi(h_k), \qquad \frac{\partial x_i}{\partial a_i} = 1.$$

Again let $\beta_j = \frac{\partial E}{\partial x_j}$. Then

$$\underline{\beta}_j = \sum_i \beta_i(\delta_{ji} + \sum_k v_{ik}\dot\phi(h_k)\frac{\partial h_k}{\partial \underline{x}_j})$$

$$= \sum_i \beta_i(\delta_{ji} + \sum_k v_{ik}\dot\phi(h_k)w_{kj})$$

Thus,

$$\langle\underline{\beta}_j^2\rangle = \langle[\sum_i \beta_i(\delta_{ji} + \sum_k v_{ik}\dot\phi(h_k)w_{kj})]^2\rangle$$

$$= \langle\beta_j^2\rangle + \sum_{i,k}\langle v_{ik}^2\rangle\langle w_{kj}^2\rangle\mathrm{V}\dot\phi(\mathbf{q})\langle\beta_i^2\rangle$$

$$= \langle\beta_j^2\rangle(1 + \sigma_v^2\sigma_w^2\mathrm{V}\dot\phi(\mathbf{q}))$$

where in the second equality we applied the independence argument as in the proof of Thm B.5, leveraging Axiom 3.2, and in the third equality we used Axiom 3.1 to get $\langle\beta_i^2\rangle = \langle\beta_j^2\rangle$.

The other computations are similar to the proof of Thm B.12.

$\square$

## C.5 Tanh: Reduced Residual Network

### C.5.1 Forward Dynamics

**Theorem B.2.** *Suppose $\phi$ is tanh-like. Assume RRN architecture.*

- *If $\sigma_w = 0$, then $\mathbf{p}^{(l)} = l\mathrm{V}\phi(\sigma_b^2) + \mathbf{p}^{(0)}$ and $\mathbf{q}^{(l)} = \sigma_b^2$.*

- *If $\sigma_w > 0$, $\lim_{l\to\infty}\mathbf{p}^{(l)}/l = 1$ and $\lim_{l\to\infty}\mathbf{q}^{(l)}/(\sigma_w^2 l) = 1$. If $\phi = \tanh$, then we can obtain more terms of the asymptotic expansions:*

$$\mathbf{p}^{(l)} = l - 2C\sigma_w^{-1}l^{1/2} - C^2\sigma_w^{-2}\log l + O(1)$$

$$\mathbf{q}^{(l)} = \sigma_w^2 l - 2C\sigma_w l^{1/2} - C^2\log l + O(1)$$

*as $l \to \infty$, where $C = \sqrt{2/\pi}$.*

*Proof.* The case with $\sigma_w = 0$ is trivial. We assume $\sigma_w > 0$ from here on.

**p and q are asymptotically linear with $l$.** We first show that, for any $\omega < 1$,

$$l + \mathbf{p}^{(0)} \geq \mathbf{p}^{(l)} \geq \omega l$$

and

$$\sigma_w^2(l + \mathbf{p}^{(0)}) + \sigma_b^2 \geq \mathbf{q}^{(l)} \geq \sigma_w^2\omega(l-1) + \sigma_b^2,$$

so that $\mathbf{p}^{(l)} \sim l$ and $\mathbf{q}^{(l)} \sim \sigma_w^2 l$.

The upper bounds are trivial, given $\mathrm{V}\phi(\mathbf{q}) \leq 1$ for any $\mathbf{q}$. We show the lower bounds for any $\omega < 1$.
For any $\epsilon > 0$, define $\aleph_\epsilon$ by $\phi^2(\aleph_\epsilon) = \exp(-\epsilon)$. Then

$$\mathrm{V}\phi(\mathbf{q}) \geq \exp(-\epsilon) \Pr[z \notin [-\aleph_\epsilon, \aleph_\epsilon] : z \sim \mathcal{N}(0, \mathbf{q})]$$

$$\geq \exp(-\epsilon) \left(1 - \frac{2\aleph_\epsilon}{\sqrt{2\pi\mathbf{q}}}\right)$$

where the second inequality follows from an overestimate of the $\Pr[z \in [-\aleph_\epsilon, \aleph_\epsilon]]$ via the mode of $\mathcal{N}(0, \mathbf{q})$.

For any $\mathbf{q} \geq \mathbf{q}^{(0)}$, $\mathrm{V}\phi(\mathbf{q})$ is then lower bounded by

$$\phi^2\left(\sqrt{\mathbf{q}^{(0)}}\right)\left(1 - \frac{2\sqrt{\mathbf{q}^{(0)}}}{\sqrt{2\pi\mathbf{q}^{(0)}}}\right) = \phi^2\left(\sqrt{\mathbf{q}^{(0)}}\right)\left(1 - \sqrt{\frac{2}{\pi}}\right) > 0.$$

Thus $\mathbf{p}^{(l)}$ and $\mathbf{q}^{(l)}$ are unbounded with $l$.

Furthermore, as $\mathbf{q} \to \infty$, the lower bound $\exp(-\epsilon)\left(1 - \frac{2\aleph_\epsilon}{\sqrt{2\pi\mathbf{q}}}\right)$ goes to $\exp(-\epsilon)$, for any $\epsilon$. Therefore, for any $\omega < 1$, $\mathbf{p}^{(l)} \geq \omega l$ and $\mathbf{q}^{(l)} \geq \sigma_w^2 \omega(l-1) + \sigma_b^2$.

**Asymptotic expansion.** Now we repeat the following to get each successive asymptotic term of $\mathbf{p}^{(l)}$ and $\mathbf{q}^{(l)}$: We plug in the current asymptotic form of $\mathbf{q}^{(l)}$ into $\mathrm{V}\tanh(\mathbf{q}) = 1 - C\mathbf{q}^{-1/2} + \Theta(\mathbf{q}^{-3/2})$ (Lemma C.5), where $C = \sqrt{2/\pi}$. Next we take the sum $\mathbf{q}^{(l)} = \sum_{r=1}^{l} \mathrm{V}\tanh(\mathbf{q}^{(r)})$, which yields one more term in the asymptotic expansion of $\mathbf{p}$ than the last round. We then repeat until we get only constant terms.

The following exhibits a trace of this procedure, where in the summation step for $\mathbf{q}^{(l)}$, we implicitly apply

$$\mathbf{q} = \sigma_w^2 l + o(l) = \sigma_w^2 l(1 + o(1))$$

$$\mathbf{q}^{-1/2} = \sigma_w^{-1} l^{-1/2}(1 + o(1)) = \sigma_w^{-1} l^{-1/2} + o(l^{-1/2})$$

$$\mathbf{p} = \sum_{r=1}^{l} 1 - C(\mathbf{q}^{(r)})^{-1/2} + \Theta((\mathbf{q}^{(r)})^{-3/2})$$

$$= \sum_{r=1}^{l} 1 - C(\sigma_w^{-1} r^{-1/2} + o(r^{-1/2})) + \Theta(r^{-3/2})$$

$$= l - 2C\sigma_w^{-1} l^{1/2} + o(l^{1/2})$$

$$\mathbf{q} = \sigma_w^2 l - 2C\sigma_w l^{1/2} + o(l^{1/2}) = \sigma_w^2 l(1 - 2C\sigma_2^{-1} l^{-1/2} + o(l^{-1/2}))$$

$$\mathbf{q}^{-1/2} = \sigma_w^{-1} l^{-1/2}(1 + C\sigma_w^{-1} l^{-1/2} + o(l^{-1/2})) = \sigma_w^{-1} l^{-1/2} + C\sigma_w^{-2} l^{-1} + o(l^{-1})$$

$$\mathbf{p} = \sum_{r=1}^{l} 1 - C(\sigma_w^{-1} l^{-1/2} + C\sigma_w^{-2} l^{-1} + o(l^{-1})) + \Theta(l^{-3/2})$$

$$= l - 2C\sigma_w^{-1} l^{1/2} - C^2\sigma_w^{-2} \log l + o(\log l)$$

$$\mathbf{q} = \sigma_w^2 l(1 - 2C\sigma_w^{-1} l^{-1/2} - C^2\sigma_w^{-2} \frac{\log l}{l} + o(\frac{\log l}{l}))$$

$$\mathbf{q}^{-1/2} = \sigma_w^{-1} l^{-1/2}(1 + C\sigma_w^{-1} l^{-1/2} + \frac{1}{2}C^2\sigma_w^{-2} \frac{\log l}{l} + o(\frac{\log l}{l}))$$

$$\mathbf{p} = \sum_{r=1}^{l} 1 - C(\sigma_w^{-1} r^{-1/2} + C\sigma_w^{-2} r^{-1} + \frac{1}{2}C^2\sigma_w^{-3} \frac{\log r}{r^{3/2}} + o(\frac{\log r}{r^{3/2}}))) + \Theta(r^{-3/2})$$

$$= l - 2C\sigma_w^{-1} l^{1/2} - C^2\sigma_w^{-2} \log l + O(1)$$

which is what we want. $\qquad\square$

**Lemma C.16.** *Let $\phi$ is antisymmetric. Then for $\tau \in [0, \pi/2]$,*

$$\mathrm{W}\phi(q, q\cos\tau) = \lim_{t\to\tau} \frac{1}{\pi\sin t} \int_{w'\geq|w|} \mathrm{d}w\,\mathrm{d}w'\Upsilon(w, w'; \tau)\phi(\frac{\sqrt{q}}{\sqrt{2}}(w+w'))\phi(\frac{\sqrt{q}}{\sqrt{2}}(w'-w))$$

$$= \frac{1}{\pi}\int_0^\infty r\,\mathrm{d}re^{-r^2/2}\int_0^\pi \mathrm{d}\theta\Sigma(\sqrt{q}r, \theta; \tau)$$

$$= \frac{1}{\pi}\int_0^\infty s\,\mathrm{d}sq^{-1}e^{-s^2q^{-1}/2}\int_0^\pi \mathrm{d}\theta\Sigma(s, \theta; \tau)$$

$$= \frac{1}{\pi}\int_0^\pi \mathrm{d}\theta\int_0^\infty \mathrm{d}se^{-s^2q^{-1}/2}\frac{\partial}{\partial s}\Sigma(s, \theta; \tau)$$

*where* $\Upsilon(w, w'; \tau) := e^{-\frac{1}{2}(\frac{w^2}{1-c}+\frac{(w')^2}{1+c})} - e^{-\frac{1}{2}(\frac{(w')^2}{1-c}+\frac{w^2}{1+c})}$ *with* $c = \cos\tau$, *and* $\Sigma(s, \theta; \tau) := \phi(s\sin\theta)\phi(s\sin(\theta-\tau))$.

Of course, in the above lemma, the limit in the first equation is only necessary when $\tau = 0$ or $\tau = \pi/2$.

*Proof.* Let $c := \cos\tau$ and

$$\Gamma := \mathrm{W}\phi(q, cq) = \frac{1}{2\pi q\sqrt{1-c^2}}\int \mathrm{d}\mathbf{z}\exp(-\mathbf{z}^T\Sigma^{-1}\mathbf{z}/2)\phi(z)\phi(z'),$$

where $\Sigma = \begin{pmatrix} q & cq \\ cq & q \end{pmatrix}$.

Our proof will have two portions: Symmetrization of the $\Gamma$ integral and trigonometric change of variables for evaluation.

**Symmetrization.** $\Sigma$ is diagonalized by $\Omega = \frac{1}{\sqrt{2q}}\begin{pmatrix} -1 & 1 \\ 1 & 1 \end{pmatrix}$,

$$\Sigma = \Omega^T\mathrm{Diag}(1-c, 1+c)\Omega.$$

By a change of variable $\mathbf{w} = \Omega\mathbf{z}$, so that $\mathrm{d}\mathbf{w} = q^{-1}\mathrm{d}\mathbf{z}$, we have

$$\Gamma = \frac{1}{2\pi\sqrt{1-c^2}}\int \mathrm{d}\mathbf{w}\exp(-\mathbf{w}^T\mathrm{Diag}(1-c, 1+c)^{-1}\mathbf{w}/2)\phi(\frac{\sqrt{q}}{\sqrt{2}}(w'-w))\phi(\frac{\sqrt{q}}{\sqrt{2}}(w+w'))$$

$$= \frac{1}{2\pi\sqrt{1-c^2}}\int \mathrm{d}w\,\mathrm{d}w'e^{-\frac{1}{2}(\frac{w^2}{1-c}+\frac{(w')^2}{1+c})}\phi(\frac{\sqrt{q}}{\sqrt{2}}(w'-w))\phi(\frac{\sqrt{q}}{\sqrt{2}}(w+w'))$$

By a change of variable swapping $w$ with $w'$, we get

$$\Gamma = -\frac{1}{2\pi\sqrt{1-c^2}}\int \mathrm{d}w\,\mathrm{d}w'e^{-\frac{1}{2}(\frac{(w')^2}{1-c}+\frac{w^2}{1+c})}\phi(\frac{\sqrt{q}}{\sqrt{2}}(w+w'))\phi(\frac{\sqrt{q}}{\sqrt{2}}(w'-w))$$

Thus

$$2\Gamma = \frac{1}{2\pi\sqrt{1-c^2}}\int \mathrm{d}w\,\mathrm{d}w'\Upsilon(w, w'; \tau)\phi(\frac{\sqrt{q}}{\sqrt{2}}(w+w'))\phi(\frac{\sqrt{q}}{\sqrt{2}}(w'-w))$$

where

$$\Upsilon(w, w'; \tau) = e^{-\frac{1}{2}(\frac{w^2}{1-c}+\frac{(w')^2}{1+c})} - e^{-\frac{1}{2}(\frac{(w')^2}{1-c}+\frac{w^2}{1+c})}.$$

Note that, by the antisymmetry of $\phi$, the integrand $K := \Upsilon(w, w'; \tau)\phi(\ldots)\phi(\ldots)$ above has the symmetries $K(w, w') = K(w', w) = K(w, -w')$, and is everywhere nonnegative. <span style="color:red">Fig. C.10</span> displays a contour plot of $K$ for typical values of $q$ and $c$. So

$$\Gamma = \frac{1}{\pi\sqrt{1-c^2}}\int_{w'\geq|w|} \mathrm{d}w\,\mathrm{d}w'K(w, w').$$

**Figure C.10:** The integrand of $\Gamma$ after symmetrization. Here $c = .2$ and $q = 100$ and $\phi = \tanh$.

This gives the first equation in the lemma.

**Polar Coordinates.** Let $\frac{w}{\sqrt{1-c}} = r\cos\theta$, $\frac{w'}{\sqrt{1+c}} = r\sin\theta$, so that

$$w = r\cos\theta\sqrt{1-c} = \sqrt{2}r\cos\theta\sin\frac{\tau}{2}$$

$$w' = r\sin\theta\sqrt{1+c} = \sqrt{2}r\sin\theta\cos\frac{\tau}{2}$$

$$dw\,dw' = \sqrt{1-c^2}\,r\,dr\,d\theta = (\sin^2\tau)r\,dr\,d\theta.$$

Then

$$\mathbf{A} := \int_{w'\geq|w|} e^{-(\frac{w^2}{1-c}+\frac{(w')^2}{1+c})/2}\phi(\sqrt{q/2}(w+w'))\phi(\sqrt{q/2}(w'-w))\,dw\,dw'$$

$$= \sin^2\tau\int_0^\infty r\,dr e^{-r^2/2}\int_{\tau/2}^{\pi-\tau/2} d\theta\phi(\sqrt{q}r\sin(\theta+\tau/2))\phi(\sqrt{q}r\sin(\theta-\tau/2)).$$

Similarly, let $\frac{w}{\sqrt{1+c}} = r\cos\theta$, $\frac{w'}{\sqrt{1-c}} = r\sin\theta$, so that

$$w = r\cos\theta\sqrt{1+c} = \sqrt{2}r\cos\theta\cos\frac{\tau}{2}$$

$$w' = r\sin\theta\sqrt{1-c} = \sqrt{2}r\sin\theta\sin\frac{\tau}{2}$$

$$dw\,dw' = \sqrt{1-c^2}\,r\,dr\,d\theta = (\sin^2\tau)r\,dr\,d\theta,$$

and

$$\mathbf{B} = \int_{w'\geq|w|} e^{-(\frac{w^2}{1+c}+\frac{(w')^2}{1-c})/2}\phi(\sqrt{q/2}(w+w'))\phi(\sqrt{q/2}(w'-w))\,dw\,dw'$$

$$= -\sin^2\tau\int_0^\infty r\,dr e^{-r^2/2}\int_{\pi/2-\tau/2}^{\pi/2+\tau/2} d\theta\phi(\sqrt{q}r\cos(\theta+\tau/2))\phi(\sqrt{q}r\cos(\theta-\tau/2))$$

$$= -\sin^2\tau\int_0^\infty r\,dr e^{-r^2/2}\int_{-\tau/2}^{\tau/2} d\theta\phi(\sqrt{q}r\sin(\theta+\tau/2))\phi(\sqrt{q}r\sin(\theta-\tau/2)).$$

Thus

$$\Gamma = \frac{1}{\pi\sqrt{1-c^2}}(\mathbf{A}-\mathbf{B})$$

$$= \frac{1}{\pi}\int_0^\infty r\,dr e^{-r^2/2}\int_{-\tau/2}^{\pi-\tau/2} d\theta\phi(\sqrt{q}r\sin(\theta+\tau/2))\phi(\sqrt{q}r\sin(\theta-\tau/2))$$

$$= \frac{1}{\pi}\int_0^\infty r\,dr e^{-r^2/2}\int_0^{\pi} d\theta\phi(\sqrt{q}r\sin(\theta))\phi(\sqrt{q}r\sin(\theta-\tau)).$$

This gives the second equation in the lemma, and a change of variables $s = \sqrt{q}r$ gives the third. For the fourth equality, we start from the third equality, and apply integration by parts:

$$\frac{1}{\pi}\int_0^\infty s\,\mathrm{d}sq^{-1}e^{-s^2q^{-1}/2}\int_0^\pi \mathrm{d}\theta\Sigma(s,\theta;\tau)$$

$$=\frac{1}{\pi}\int_0^\pi \mathrm{d}\theta\int_0^\infty \mathrm{d}ssq^{-1}e^{-s^2q^{-1}/2}\Sigma(s,\theta;\tau)$$

$$=\frac{1}{\pi}\int_0^\pi \mathrm{d}\theta\left(-e^{-s^2q^{-1}/2}\Sigma(s,\theta;\tau)\Big|_{s=0}^\infty + \int_0^\infty \mathrm{d}se^{-s^2q^{-1}/2}\frac{\partial}{\partial s}\Sigma(s,\theta;\tau)\right)$$

$$=\frac{1}{\pi}\int_0^\pi \mathrm{d}\theta\int_0^\infty \mathrm{d}se^{-s^2q^{-1}/2}\frac{\partial}{\partial s}\Sigma(s,\theta;\tau).$$

where the last equality follows because $\Sigma(0,\theta;\tau) = 0$ and $e^{-s^2q^{-1}/2} \to 0$ as $s \to \infty$. $\qquad\square$

In the following lemmas, the "2" is not important, and can be any arbitrary finite or infinite value.

**Lemma C.17.** *Suppose a function* $f : (0,2) \to \mathbb{R}$ *is* $C^k$ *on* $(0,2)$. *If* $\lim_{x\downarrow 0} f^{(i)}(x)$ *exists and is finite for every* $i \in [0,k]$, *then* $f$ *can be extended to* $[0,2)$ *such that one sided ith derivatives exist at 0 for all* $i \in [0,k]$.

*Proof.* Consider $\overline{f^{(i)}}(0) := f^{(i)}(1) - \int_0^1 f^{(i+1)}(x)\,\mathrm{d}x$ for $i \in [0,k-1]$, which naturally is also equal to $f^{(i)}(\epsilon) - \int_0^\epsilon f^{(i+1)}(x)\,\mathrm{d}x$ for any $\epsilon > 0$. Certainly $f^{(i)}(x) \to \overline{f^{(i)}}(0)$ as $x \to 0$ if this limit exists — and by assumption it does, for $0 \le i \le k-1$. Therefore, we can define the extension of $f^{(i)}$ to $x = 0$ to be $f^{(i)}(0) := \overline{f^{(i)}}(0)$. But we need to check that for $i \in [0,k-1]$.

$$\lim_{\epsilon\to 0}\frac{1}{\epsilon}(f^{(i)}(\epsilon) - f^{(i)}(0)) = f^{(i+1)}(0)$$

so that all one sided $i$th derivatives exist. But

$$\frac{1}{\epsilon}(f^{(i)}(\epsilon) - f^{(i)}(0)) = \frac{1}{\epsilon}\int_0^\epsilon f^{(i+1)}(x)\,\mathrm{d}x$$

$$= f^{(i+1)}(0) + \int_0^1 (f^{(i+1)}(x) - f^{(i)}(0))\mathrm{I}(x \in [0,\epsilon])\,\mathrm{d}x$$

Since $\lim_{x\downarrow 0} f^{(i+1)}(x) = f^{(i+1)}(0)$, $f^{(i+1)}(x) - f^{(i+1)}(0)$ is bounded for small $x$, and by dominated convergence, $\int_0^1 (f^{(i+1)}(x) - f^{(i)}(0))\mathrm{I}(x \in [0,\epsilon])\,\mathrm{d}x \to \int_0^1 0\,\mathrm{d}x = 0$ as $\epsilon \to 0$. Thus

$$\lim_{\epsilon\to 0}\frac{1}{\epsilon}(f^{(i)}(\epsilon) - f^{(i)}(0)) = f^{(i+1)}(0)$$

as desired. $\qquad\square$

**Lemma C.18.** *If* $f : [0,2) \to \mathbb{R}$ *is* $C^k$ *on* $(0,2)$ *and has one sided derivatives at 0 up to order* $k$, *then*

$$f(\epsilon) = f(0) + \epsilon f^{(1)}(0) + \cdots + \frac{\epsilon^{i-1}}{(i-1)!}f^{(i-1)}(0) + O(\epsilon^i)$$

*for any* $i \le k$.

*Proof.* We have

$$f(\epsilon) = f(0) + \int_0^\epsilon f^{(1)}(x)\,\mathrm{d}x$$

$$= f(0) + \epsilon f^{(1)}(0) + \int_0^\epsilon f^{(1)}(x) - f^{(1)}(0)\,\mathrm{d}x$$

$$= f(0) + \epsilon f^{(1)}(0) + \int_0^\epsilon \int_0^{x_0} f^{(2)}(x_2)\,\mathrm{d}x_2\,\mathrm{d}x_1$$

$$= f(0) + \epsilon f^{(1)}(0) + \frac{\epsilon^2}{2}f^{(2)}(0) + \int_0^\epsilon \int_0^{x_1} f^{(2)}(x_2) - f^{(2)}(0)\,\mathrm{d}x_2\,\mathrm{d}x_1$$

$$\vdots$$

$$f(\epsilon) = f(0) + \epsilon f^{(1)}(0) + \cdots + \frac{\epsilon^{i-1}}{(i-1)!}f^{(i-1)}(0) + \int_0^\epsilon \mathrm{d}x_1 \int_0^{x_1} \mathrm{d}x_2 \cdots \int_0^{x_{i-1}} \mathrm{d}x_i f^{(i)}(x_i)$$

for any $i \le k$. It suffices then to bound the size of the integral. Since $f^{(i)}(x) \to f^{(i)}(0)$ as $x \downarrow 0$ by assumption, $|f^{(i)}(x_i)|$ is bounded by some constant $C$ on the integration region $\mathbb{A} := \{(x_1,\ldots,x_i) : \epsilon \ge x_1 \ge \cdots \ge x_i\}$ for small enough $\epsilon$. Therefore,

$$\int_0^\epsilon \mathrm{d}x_1 \int_0^{x_1} \mathrm{d}x_2 \cdots \int_0^{x_{i-1}} \mathrm{d}x_i f^{(i)}(x_i)$$

$$= \int f^{(i)}(x_i)\mathrm{I}(\vec{x} \in \mathbb{A})\,\mathrm{d}\vec{x}$$

$$\le C|\mathbb{A}|$$

$$= \Theta(\epsilon^i).$$

$\square$

As a corollary,

**Lemma C.19.** *If $f : (0,2) \to \mathbb{R}$ is smooth on $(0,2)$ and $\lim_{x\to 0} f^{(i)}(x)$ exists and is finite for all $i$, then $f$ can be extended to $[0,2)$ and be one-sided smooth at $0$, and*

$$f(\epsilon) = f(0) + \epsilon f^{(1)}(0) + \cdots + \frac{\epsilon^{i-1}}{(i-1)!}f^{(i-1)}(0) + O(\epsilon^i)$$

*for any $i$.*

**Lemma C.20.** *Let $\phi = \tanh$. For any fixed $c$, $\mathrm{W}\phi(q, cq)$ is smooth (infinitely differentiable) on $q \in (0,\infty)$. As a function of $Q := q^{-1}$, it can be extended smoothly to the point $Q = 0$, so that*

$$\mathrm{W}\phi(q, cq) = \lim_{q'\to\infty} \mathrm{W}\phi(q', cq') + q^{-1}\lim_{q'\to\infty} \partial\mathrm{W}\phi(q', cq')/\partial(q')^{-1} + \cdots$$

$$+ \frac{q^{-i+1}}{(i-1)!}\lim_{q'\to\infty} \partial^{i-1}\mathrm{W}\phi(q', cq')/\partial(q')^{-i+1} + O(q^{-i})$$

*for any $i \ge 0$. Furthermore, for $c$ bounded away from $1$, the constants hidden $O$ can be taken independent of $c$.*

*Proof.* **Smoothness on $(0,\infty)$.** By the third equation of Lemma C.16, for $Q \in (0,\infty) \iff q \in (0,\infty)$,

$$\frac{1}{\pi}\int_0^\infty s\,\mathrm{d}s \left|\frac{\partial^n}{\partial Q^n}\left(Qe^{-s^2 Q/2}\right)\right| \int_0^\pi \mathrm{d}\theta |\phi(s\sin\theta)\phi(s\sin(\theta - \tau))|$$

$$\le \int_0^\infty s\,\mathrm{d}s \left|\frac{\partial^n}{\partial Q^n}\left(Qe^{-s^2 Q/2}\right)\right| < \infty,$$

so by Leibniz's integral rule and a simple induction, all derivatives of $\mathrm{W}\phi(q, cq)$ against $Q$ exists for any $Q \in (0,\infty)$.

**Extension to $Q = 0$.** By Lemma C.19, it suffices to show that the limit of $\frac{\partial^k \mathrm{W}\phi(q, cq)}{\partial Q^k}$ exists and is finite as $Q \to 0$, for all $k$. Let $\tau = \arccos c$. By the fourth equation of Lemma C.16, we have explicitly

$$\frac{\partial^k \mathrm{W}\phi(q, cq)}{\partial Q^k} = \frac{1}{\pi} \int_0^\pi \mathrm{d}\theta \int_0^\infty \mathrm{d}s (-s^2/2)^k e^{-s^2 Q/2} \frac{\partial}{\partial s} \Sigma(s, \theta; \tau)$$

$$= \frac{(-2)^{-k}}{\pi} \int_0^\pi \mathrm{d}\theta \int_0^\infty \mathrm{d}s \, s^{2k} e^{-s^2 Q/2} \frac{\partial}{\partial s} \Sigma(s, \theta; \tau)$$

for any $Q \in (0, \infty)$. Note that for $\phi = \tanh$, $\dot\phi = \mathrm{sech}^2$,

$$\frac{\partial}{\partial s} \Sigma(s, \theta; \tau) = \sin\theta \dot\phi(s \sin\theta)\phi(s \sin(\theta - \tau)) + \sin(\theta - \tau)\phi(s \sin\theta)\dot\phi(s \sin(\theta - \tau)).$$

We split the integral of $\frac{\partial^k \mathrm{W}\phi}{\partial Q^k}$ as follows:

$$\frac{\partial^k \mathrm{W}\phi(q, cq)}{\partial Q^k} = \frac{(-2)^{-k}}{\pi} \int_0^\pi \mathrm{d}\theta \int_0^\infty \mathrm{d}s \, s^{2k} e^{-s^2 Q/2} \sin\theta \dot\phi(s \sin\theta)\phi(s \sin(\theta - \tau))$$

$$+ \frac{(-2)^{-k}}{\pi} \int_0^\pi \mathrm{d}\theta \int_0^\infty \mathrm{d}s \, s^{2k} e^{-s^2 Q/2} \sin(\theta - \tau)\phi(s \sin\theta)\dot\phi(s \sin(\theta - \tau))$$

We show that for each piece, the limit as $Q \to 0$ exists and is finite, for any $k$. This will prove the smooth extendability of $\mathrm{W}\phi$ to $Q = 0$. We will do this for the first piece; the second is similar.

For $Q > 0$, the integrand is absolutely integrable, so we may switch the integrals.

$$\int_0^\pi \mathrm{d}\theta \int_0^\infty \mathrm{d}s \, s^{2k} e^{-s^2 Q/2} \sin\theta \dot\phi(s \sin\theta)\phi(s \sin(\theta - \tau))$$

$$= \int_0^\infty \mathrm{d}s \, s^{2k} e^{-s^2 Q/2} \int_0^\pi \mathrm{d}\theta \sin\theta \dot\phi(s \sin\theta)\phi(s \sin(\theta - \tau))$$

We now try to bound the inner integral by an exponentially decreasing term $e^{-s\mu}$ for some $\mu$; clearly, by monotone convergence on the outer integral as $Q \to 0$, this would show the limit of the integral exists and is finite.

Because $\phi$ is odd and $\dot\phi$ is even, the inner integrand is negative on $\theta \in [0, \tau)$ and positive on $\theta \in (\tau, \pi]$. We will break up the inner integral as follows, for some fixed $\epsilon > 0$ satisfying $\tau - \epsilon > 0$ independent of $s$ (recall $\tau \in (0, \pi/2)$).

$$\int_0^\pi \mathrm{d}\theta \sin\theta \dot\phi(s \sin\theta)\phi(s \sin(\theta - \tau))$$

$$= \left( \int_0^\epsilon + \int_{\pi-\epsilon}^\pi \right) \mathrm{d}\theta \sin\theta \dot\phi(s \sin\theta)\phi(s \sin(\theta - \tau)) + \int_\epsilon^{\pi-\epsilon} \mathrm{d}\theta \sin\theta \dot\phi(s \sin\theta)\phi(s \sin(\theta - \tau))$$

Now because $\dot\phi(z) = \mathrm{sech}^2(z) \le 2e^{-z}$, and $\sin\theta \ge \sin\epsilon$ on $\theta \in [\epsilon, \pi - \epsilon]$,

$$\left| \int_\epsilon^{\pi-\epsilon} \mathrm{d}\theta \sin\theta \dot\phi(s \sin\theta)\phi(s \sin(\theta - \tau)) \right|$$

$$\le 2 \int_\epsilon^{\pi-\epsilon} \mathrm{d}\theta \exp(-s \sin\epsilon)$$

$$= 2(\pi - 2\epsilon) \exp(-s \sin\epsilon).$$

For the other part:

$$\int_{\pi-\epsilon}^\pi \mathrm{d}\theta \sin\theta \dot\phi(s \sin\theta)\phi(s \sin(\theta - \tau))$$

$$= \int_\epsilon^0 \sin(\pi - \theta)\dot\phi(s \sin\pi - \theta)\phi(s \sin(\pi - \theta - \tau)) \, \mathrm{d}(\pi - \theta)$$

$$= \int_0^\epsilon \mathrm{d}\theta \sin\theta \dot\phi(s \sin\theta)\phi(s \sin\theta + \tau)$$

so that

$$\left(\int_0^\epsilon + \int_{\pi-\epsilon}^\pi\right) d\theta \sin\theta \dot\phi(s\sin\theta)\phi(s\sin(\theta-\tau))$$

$$= \int_0^\epsilon d\theta \sin\theta \dot\phi(s\sin\theta)[\phi(s\sin(\tau+\theta)) - \phi(s\sin(\tau-\theta))]$$

But by intermediate value theorem, $\phi(s\sin(\tau+\theta)) - \phi(s\sin(\tau-\theta)) = 2\theta\partial\phi(s\sin(\tau+\theta))/\partial\theta|_{\theta=\psi} = 2\theta\dot\phi(s\sin(\tau+\psi))s\cos(\tau+\psi)$ for some $\psi \in [-\theta,\theta]$. By the assumption on $\epsilon$, $\phi(s\sin(\tau+\theta)) - \phi(s\sin(\tau-\theta)) \le 2\epsilon\dot\phi(s\sin(\tau-\epsilon))s\cos(\tau-\epsilon)$. Then

$$\int_0^\epsilon d\theta \sin\theta \dot\phi(s\sin\theta)[\phi(s\sin(\tau+\theta)) - \phi(s\sin(\tau-\theta))]$$

$$\le \int_0^\epsilon d\theta \sin\theta \dot\phi(s\sin\theta)2\epsilon\dot\phi(s\sin(\tau-\epsilon))s\cos(\tau-\epsilon)$$

$$\le 2\epsilon\dot\phi(s\sin(\tau-\epsilon))s\cos(\tau-\epsilon)O(1)$$

Because $\tau - \epsilon > 0$ by assumption on $\epsilon$, and because $\dot\phi(z) = \exp(-\Theta_+(z))$, this quantity is $\exp(-\Theta_+(z))$, as desired (here $\Theta_+$ denotes a positive quantity).

Thus

$$\int_0^\pi d\theta \sin\theta \dot\phi(s\sin\theta)\phi(s\sin(\theta-\tau))$$

$$= \left(\int_0^\epsilon + \int_{\pi-\epsilon}^\pi\right) d\theta \sin\theta \dot\phi(s\sin\theta)\phi(s\sin(\theta-\tau)) + \int_\epsilon^{\pi-\epsilon} d\theta \sin\theta \dot\phi(s\sin\theta)\phi(s\sin(\theta-\tau))$$

$$= \exp(-\Theta_+(s))$$

and similarly for the other piece of $\frac{\partial^k W\phi}{\partial Q^k}$, so that

$$\int_0^\infty ds\, s^{2k} e^{-s^2 Q/2} \int_0^\pi d\theta \sin\theta \dot\phi(s\sin\theta)\phi(s\sin(\theta-\tau))$$

$$= \int_0^\infty ds\, s^{2k} e^{-s^2\frac{Q}{2}-\Theta_+(z)}$$

$$\to \int_0^\infty ds\, s^{2k} e^{-\Theta_+(z)}$$

is finite as $Q \to 0$, by monotone convergence.

**Independence of constant hidden in** $O((q')^{-i})$**.** The constant hidden is a function of the $\epsilon$ chosen above, which depend on $\tau$, but only to the extent that it must satisfy $\tau - \epsilon > 0$. As long as we are interested in a set $\mathcal{C}$ of $c$ that is bounded away from 1, the corresponding set of $\tau$ is bounded away from 0, so $\epsilon$ can be taken to be some number smaller than all of the corresponding $\tau$.

$\square$

**Lemma C.21.** *Suppose $\phi$ is tanh-like. Then for $c \in [0,1]$,*

$$W\phi(q, cq) \le \frac{2}{\pi}\arcsin(c),$$

*and weakly increases to this upper bound as $q \to \infty$. Furthermore,*

- *If $c = 0$ or 1, then equality holds regardless of $q$.*
- *If $c \in (0,1)$ is held constant, $\frac{2}{\pi}\arcsin(c) - W\phi(q, cq) = \Theta(q^{-1})$, where the hidden constants in $\Theta$ depend on $c$. But the constants can be made independent of $c$ if $c \in [\epsilon, 1-\epsilon]$ for some $\epsilon > 0$.*

*Proof.* The cases of $c = 0$ or 1 are obvious by the definition of W. So from here on we assume $c \in (0,1)$.

**Figure C.11:** We verify empirically that the subleading term in $\mathrm{W}\tanh(q,cq)$ is linear in $q^{-1}$, for constant $c$. Indeed, observe that the curve of of $\mathrm{W}\tanh$ intersects the y-axis at an angle.

Let $\tau := \arccos c$. By the first equation of Lemma C.16 and the assumption that $\phi$ is tanh-like, it is immediate that $\mathrm{W}\phi(q,cq)$ is nondecreasing in $q$. By dominated convergence, using the second equation of Lemma C.16, we get

$$
\lim_{q\to\infty} \mathrm{W}\phi(q,cq) = \frac{1}{\pi}\int_0^\infty r\,\mathrm{d}re^{-r^2/2}(\pi - 2\tau)
$$
$$
= \frac{\pi - 2\tau}{\pi}
$$
$$
= \frac{2}{\pi}\arcsin c.
$$

Then the convergence rate is $O(q^{-1})$ by Lemma C.20 and Taylor's theorem. Thus to show the convergence rate is $\Theta(q^{-1})$, it suffices to show that $\mathbf{D} := \frac{\partial \mathrm{W}\phi(q,cq)}{\partial Q} < 0$. But this is apparent from the first equation of Lemma C.16: For $\tau \in (0,\pi/2)$,

$$
\mathbf{D} = \frac{1}{\pi\sin\tau}\int_{w'\geq|w|}\mathrm{d}w\,\mathrm{d}w'\Upsilon(w,w';\tau)(-\frac{1}{2\sqrt{2}}Q^{-3/2})
$$
$$
\times [\dot\phi(\sqrt{q/2}(w+w'))\phi(\sqrt{q/2}(w'-w))
$$
$$
+ \phi(\sqrt{q/2}(w+w'))\dot\phi(\sqrt{q/2}(w'-w))]
$$
$$
< 0
$$

since $\Upsilon$ is positive on the integration domain, and $\dot\phi$ and $\phi$ are both positive for positive arguments, by the assumption of $\phi$ being tanh-like.

**Independence of the constants in $\Theta(q^{-1})$ from $c$ when $c \in [\epsilon, 1-\epsilon]$.** By Lemma C.20, the upper constant can be made independent from $c$. Since $\mathbf{D}$ is monotonically decreasing in $c$ (or monotonically increasing in $\tau$) and $|\mathbf{D}|$ is monotonically increasing in $c$ (or monotonically decreasing in $\tau$), we have $|\mathbf{D}| > |\mathbf{D}|\Big|_{c=\epsilon}$, which can be taken to be the lower constant in $\Theta(q^{-1})$. $\qquad\square$

Fig. C.11 verifies empirically that the subleading term in $\mathrm{W}\tanh(q,cq)$ is linear in $q^{-1}$, for constant $c$.

**Theorem B.4.** *Suppose $\phi$ is a tanh-like nonlinearity in an RRN. Assume $\mathbf{e}^{(0)} < 1$.*

- *If $\sigma_w = 0$, then $\boldsymbol{\gamma}^{(l)} = l\mathrm{W}\phi(\sigma_b^2, \sigma_b^2) + \boldsymbol{\gamma}^{(0)} = l\mathrm{V}\phi(\sigma_b^2) + \boldsymbol{\gamma}^{(0)}$ and $\boldsymbol{\lambda}^{(l)} = \sigma_b^2$, so that $\mathbf{e}^{(l)} \to 1$ and $1 - \mathbf{e}^{(l)} = \Theta(l^{-1})$. As a result, $\mathbf{s}^{(l)} = \mathbf{p}^{(l)}(1-\mathbf{e}^{(l)}) = \Theta(1)$.*

- *If $\sigma_w > 0$, then $\boldsymbol{\gamma}^{(l)} = \check\Theta(l^{\frac{2}{\pi}})$, and $\mathbf{e}^{(l)} \to 0$ like $\check\Theta(l^{\frac{2}{\pi}-1})$. Thus $\mathbf{s}^{(l)} = \Theta(\mathbf{p}^{(l)}) = \Theta(l)$.*

*Proof.* We have by Lemma C.21,

$$
\boldsymbol{\gamma} = \frac{2}{\pi}\arcsin(\boldsymbol{\lambda}/\mathbf{q}) - \Theta(\mathbf{q}^{-1}) + \underline{\boldsymbol{\gamma}}.
$$

Since $\mathbf{q} = \sigma_w^2 \underline{\mathbf{p}} + \sigma_b^2$ by Thm B.2, and $\boldsymbol{\lambda} = \sigma_w^2 \underline{\boldsymbol{\gamma}} + \sigma_b^2$ by Thm B.3,

$$\boldsymbol{\gamma} = \frac{2}{\pi} \arcsin\left(\frac{\sigma_w^2 \underline{\boldsymbol{\gamma}} + \sigma_b^2}{\sigma_w^2 \underline{\mathbf{p}} + \sigma_b^2}\right) - \Theta(\mathbf{q}^{-1}) + \underline{\boldsymbol{\gamma}}.$$

We claim that $\boldsymbol{\gamma}^{(l)} \to \infty$ as $l \to \infty$. Otherwise, there is some $C$ such that $\boldsymbol{\gamma}^{(l)} \leq C$ for all $l$. For large enough $l$, $\mathbf{p}^{(l)} \geq \omega l$ for any $\omega < 1$ and $\arcsin\left(\frac{C}{\sigma_w^2 \mathbf{p}^{(l-1)} + \sigma_b^2}\right) = \Theta(1/l)$ by linearization of arcsin. Thus $\boldsymbol{\gamma}^{(l)} = \Theta(\log l)$, but this contradicts our assumption that $\boldsymbol{\gamma}$ is bounded. This proves our claim.

Therefore, for large enough $l$,

$$\frac{\sigma_w^2 \underline{\boldsymbol{\gamma}} + \sigma_b^2}{\sigma_w^2 \underline{\mathbf{p}} + \sigma_b^2} = \underline{\boldsymbol{\gamma}}/\underline{\mathbf{p}} + \Theta(l^{-1}).$$

Fig. C.12 shows $\frac{2}{\pi} \arcsin x$ vs $x$. One sees that 1 is an unstable fixed point; if $\mathbf{e} < 1 - \epsilon$, then $\frac{2}{\pi} \arcsin \mathbf{e} < 1 - \epsilon - \delta$ for some $\delta$. Thus $c$ drops monotonically until some threshold under which the linearization of arcsin, $\arcsin x = x + \Theta(x^3)$, is applicable. So for large enough $l$,

$$\boldsymbol{\gamma} - \underline{\boldsymbol{\gamma}} = \frac{2}{\pi} \arcsin(\underline{\boldsymbol{\gamma}}/\underline{\mathbf{p}} + \Theta(l^{-1})) - \Theta(l^{-1})$$
$$= \frac{2}{\pi} \underline{\boldsymbol{\gamma}}/\underline{\mathbf{p}} + O(l^{-1})$$

As $\mathbf{p}^{(l)} \sim l$ by Thm B.2, this difference equation has solution $\boldsymbol{\gamma} = \Omega(l^{\frac{2}{\pi} - \epsilon}), O(l^{\frac{2}{\pi} + \epsilon})$ for any $\epsilon$ by using the dynamics of Lemma C.11 to upper and lower bound this difference equation. □

### C.5.2 Backward Dynamics

**Theorem B.6.** *For $\phi = \tanh$ in an RRN,*

- *If $\sigma_w = 0$, $\boldsymbol{\chi}^{(m)} = \boldsymbol{\chi}^{(l)}$ for all $l, m$.*

- *If $\sigma_w > 0$,*

$$\log(\boldsymbol{\chi}^{(m)}/\boldsymbol{\chi}^{(l)}) = \mathcal{A}(\sqrt{l} - \sqrt{m}) + \mathcal{B}(\log l - \log m) + O(1)$$

*where $\mathcal{A} = \frac{4}{3}\sqrt{\frac{2}{\pi}}\sigma_w$ and $\mathcal{B} = \frac{4}{3\pi} - \sigma_w^2 \frac{4}{9\pi}$.*

*Proof.* The $\sigma_w = 0$ case is obvious. We will assume $\sigma_w > 0$ from here on.

Let $\mathbf{p}^{(l)} = b_0 l + b_1 l^{1/2} + b_2 \log l + O(1)$. Then for $D = \frac{2}{3}\sqrt{\frac{2}{\pi}}$, we have (implicitly applying Lemma C.4 and Lemma C.8),

$$\mathbf{q}^{-1/2} = \sigma_w^{-1} b_0^{-1/2} l^{-1/2} (1 - b_1 b_0^{-1} 2^{-1} l^{-1/2} - b_2 b_0^{-1} 2^{-1} l^{-1} \log l + O(l^{-1}))$$

$$\mathrm{V}\dot{\phi}(\mathbf{q}) = D\mathbf{q}^{-1/2} + \Theta(\mathbf{q}^{-3/2})$$

$$= D\sigma_w^{-1} b_0^{-1/2} l^{-1/2} (1 - b_1 b_0^{-1} 2^{-1} l^{-1/2} - b_2 b_0^{-1} 2^{-1} l^{-1} \log l + O(l^{-1}))$$

$$\log(B\mathrm{V}\dot{\phi}(\mathbf{q}) + 1) = BD\sigma_w^{-1} b_0^{-1/2} l^{-1/2}$$
$$- (BD\sigma_w^{-1} b_0^{-3/2} b_1 2^{-1} + B^2 D^2 \sigma_w^{-2} b_0^{-1} 2^{-1}) l^{-1} + \Theta(l^{-3/2} \log l)$$

$$\sum_{r=1}^{l} \log(B\mathrm{V}\dot{\phi}(\mathbf{q}^{(r)}) + 1) = 2BD\sigma_w^{-1} b_0^{-1/2} l^{1/2}$$
$$- (BD\sigma_w^{-1} b_0^{-3/2} b_1 2^{-1} + B^2 D^2 \sigma_w^{-2} b_0^{-1} 2^{-1}) \log l + O(1)$$

<div align="center">39</div>

In our case, we have $b_0 = 1, b_1 = -2C\sigma_w^{-1}, b_2 = C^2\sigma_w^{-2}, B = \sigma_w^2, C = \sqrt{\frac{2}{\pi}}$, which gives

$$\sum_{r=1}^{l} \log(B\mathrm{V}\dot{\phi}(\mathbf{q}^{(r)}) + 1) = \frac{4}{3}\sqrt{\frac{2}{\pi}}\sigma_w l^{1/2} + (\frac{4}{3\pi} - \sigma_w^2\frac{4}{9\pi})\log l + O(1).$$

so that

$$\boldsymbol{\chi}^{(m)}/\boldsymbol{\chi}^{(l)} = \exp\left[\frac{4}{3}\sqrt{\frac{2}{\pi}}\sigma_w(\sqrt{l} - \sqrt{m}) + (\frac{4}{3\pi} - \sigma_w^2\frac{4}{9\pi})(\log l - \log m) + O(1)\right]$$

$\square$

**Theorem B.7.** *Suppose $\phi = \tanh$. Then in an RRN*

- *If $\sigma_w = 0$, $\boldsymbol{\chi}_b^{(l)} = \boldsymbol{\chi}^{(L)}\mathrm{V}\dot{\phi}(\sigma_b^2)$ and $\boldsymbol{\chi}_w^{(l)} = \boldsymbol{\chi}^{(L)}\mathrm{V}\dot{\phi}(\sigma_b^2)((l-1)\mathrm{V}\phi(\sigma_b^2) + \mathbf{p}^{(0)})$, where $L$ is the last layer.*

- *If $\sigma_w > 0$,*
$$\log(\boldsymbol{\chi}_b^{(m)}/\boldsymbol{\chi}_b^{(l)}) = \mathcal{A}(\sqrt{l} - \sqrt{m}) + \mathcal{B}_b(\log l - \log m) + O(1)$$
$$\log(\boldsymbol{\chi}_w^{(m)}/\boldsymbol{\chi}_w^{(l)}) = \mathcal{A}(\sqrt{l} - \sqrt{m}) + \mathcal{B}_w(\log l - \log m) + O(1)$$
*where $\mathcal{A} = \frac{4}{3}\sqrt{\frac{2}{\pi}}\sigma_w$ (same as $\mathcal{A}$ in Thm B.6) and $\mathcal{B}_b = \mathcal{B} + \frac{1}{2}, \mathcal{B}_w = \mathcal{B} - \frac{1}{2}$, with $\mathcal{B} = \frac{4}{3\pi} - \sigma_w^2\frac{4}{9\pi}$ (same as $\mathcal{B}$ in Thm B.6).*

*Proof.* The $\sigma_w = 0$ case is obvious. We will assume $\sigma_w > 0$ from here on.

As in the proof of Thm B.6,

$$\mathrm{V}\dot{\phi}(\mathbf{q}) = D\sigma_w^{-1}b_0^{-1/2}l^{-1/2} + \Theta(l^{-1})$$

where $D = \frac{2}{3}\sqrt{\frac{2}{\pi}}$. Thus by Thm B.5,

$$\log(\boldsymbol{\chi}^{(m)}/\boldsymbol{\chi}^{(l)}) = \frac{4}{3}\sqrt{\frac{2}{\pi}}\sigma_w(\sqrt{l} - \sqrt{m}) + (\frac{4}{3\pi} - \sigma_w^2\frac{4}{9\pi})(\log l - \log m) + O(1)$$

$$\log(\boldsymbol{\chi}_b^{(m)}/\boldsymbol{\chi}_b^{(l)}) = \frac{4}{3}\sqrt{\frac{2}{\pi}}\sigma_w(\sqrt{l} - \sqrt{m}) + (\frac{4}{3\pi} - \frac{1}{2} - \sigma_w^2\frac{4}{9\pi})(\log l - \log m) + O(1)$$

Similarly, since $\mathbf{p} = l + \Theta(\sqrt{l})$ by Thm B.2, we have

$$\log(\boldsymbol{\chi}_w^{(m)}/\boldsymbol{\chi}_w^{(l)}) = \frac{4}{3}\sqrt{\frac{2}{\pi}}\sigma_w(\sqrt{l} - \sqrt{m}) + (\frac{4}{3\pi} + \frac{1}{2} - \sigma_w^2\frac{4}{9\pi})(\log l - \log m) + O(1)$$

$\square$

## C.6 Tanh: Full Residual Network

### C.6.1 Forward Dynamics

**Theorem B.9.** *Suppose $\phi$ is tanh-like. Assume the FRN architecture.*

- *If $\sigma_w = 0$, then $\mathbf{p}^{(l)} = (\sigma_v^2\mathrm{V}\phi(\sigma_b^2) + \sigma_a^2)l + \mathbf{p}^{(0)}$, and $\mathbf{q}^{(l)} = \sigma_b^2$.*

- *If $\sigma_w > 0$, then $\mathbf{p}^{(l)} = b_0 l + b_1 l^{1/2} + b_2\log l + O(1)$, where*
$$b_0 = \sigma_v^2 + \sigma_a^2$$
$$b_1 = \frac{-2C\sigma_v^2\sigma_w^{-1}}{\sqrt{\sigma_v^2 + \sigma_a^2}}$$
$$b_2 = \frac{-C^2\sigma_v^4\sigma_w^{-2}}{(\sigma_v^2 + \sigma_a^2)^2}$$
*and $C = \sqrt{\frac{2}{\pi}}$. Additionally, $\mathbf{q}^{(l)} = \sigma_w^2 b_0 l + \sigma_w^2 b_1 l^{1/2} + \sigma_w^2 b_2\log l + O(1)$.*

*Proof.* The $\sigma_w = 0$ case is obvious. We will assume $\sigma_w > 0$ from here on.

As in [Thm B.2](#), $\mathbf{p}$ will have expansion $\mathbf{p} = b_0 l + b_1 l^{1/2} + b_2 \log l + O(1)$. Then, for $C = \sqrt{\frac{2}{\pi}}$,

$$\mathbf{q}^{-1/2} = \sigma_w^{-1} b_0^{-1/2} l^{-1/2} (1 - b_1 b_0^{-1} 2^{-1} l^{-1/2} - b_2 b_0^{-1} 2^{-1} l^{-1} \log l + O(l^{-1}))$$

$$\sum_{r=1}^{l} \mathrm{V}\phi(\mathbf{q}^{(r)}) = \sum_{r=1}^{l} 1 - C(\mathbf{q}^{(r)})^{-1/2} + \Theta((\mathbf{q}^{(r)})^{-3/2})$$

$$= l - 2C\sigma_w^{-1} b_0^{-1/2} l^{1/2} + C\sigma_w^{-1} b_1 b_0^{-3/2} 2^{-1} \log l + O(1)$$

$$\mathbf{p}^{(l)} = \sigma_v^2 \sum_{r=1}^{l} + \sigma_a^2 l$$

$$= (\sigma_v^2 + \sigma_a^2) l - 2C\sigma_v^2 \sigma_w^{-1} b_0^{-1/2} l^{1/2} + C\sigma_v^2 \sigma_w^{-1} b_1 b_0^{-3/2} 2^{-1} \log l + O(1)$$

which yields

$$b_0 = \sigma_v^2 + \sigma_a^2$$

$$b_1 = -2C\sigma_v^2 \sigma_w^{-1} b_0^{-1/2} = \frac{-2C\sigma_v^2 \sigma_w^{-1}}{\sqrt{\sigma_v^2 + \sigma_a^2}}$$

$$b_2 = \frac{-C^2 \sigma_v^4 \sigma_w^{-2}}{(\sigma_v^2 + \sigma_a^2)^2}$$

$\square$

**Lemma C.22.** *Suppose $\phi$ is tanh-like. Then*

$$\boldsymbol{\gamma} \leq \sigma_v^2 \frac{2}{\pi} \arcsin(\boldsymbol{\lambda}/\mathbf{q}) + \sigma_a^2 + \underline{\boldsymbol{\gamma}},$$

*and*

$$\sigma_v^2 \frac{2}{\pi} \arcsin(\boldsymbol{\lambda}/\mathbf{q}) + \sigma_a^2 + \underline{\boldsymbol{\gamma}} - \boldsymbol{\gamma} = \Theta(\mathbf{q}^{-1}).$$

*Proof.* Similar to the proof of [Lemma C.21](#). $\square$

**Lemma C.23.** *Let $u^* \in [0, 1)$. Let $f_t : [0, 1) \to [0, 1]$ be a continuous function for each $t \in \mathbb{N}$, to each of which we associate two numbers $0 \leq a_t \leq u^* \leq b_t \leq 1$. Suppose for each $t$, $f_t(u) > u$ for all $u \in [0, a_t)$ and $f_t(u) < u$ for all $u \in (b_t, 1)$. Assume that for each $u$, $f_t(u) - u \to 0$ as $t \to \infty$ uniformly over $u$. If $a_t \nearrow u^*$ and $b_t \searrow u^*$, then for any $u_0 \in [0, 1)$, the dynamics $u_t = f_t(u_{t-1})$ has a limiting point. Furthermore, either $u_t \to u^*$ or $u_t$ eventually converges monotonically (decreasing or increasing) to a limit point.*

*Proof.* Fix a $u_0 \in [0, 1)$. If $u_t \to u^*$ then we are done. Otherwise, suppose there is a neighborhood $[u^* - \epsilon, u^* + \epsilon]$ such that for an infinite sequence $t_1, t_2, \ldots$, $u_{t_i} \notin [u^* - \epsilon, u^* + \epsilon]$. WLOG assume $u_{t_i} < u^* - \epsilon$ for all $i$ and $(t_i)_i$ is the sequence of all $t$s that satisfy this inequality.

If $(t_i)_i$ contains $\{s : s \geq N\}$ for some $N$, then for some $M > N$, for every $t > M$, $a_t > u^* - \epsilon > u_t$. By assumption, $u_t$ is monotonic for all $t > M$ but is bounded above. Thus $u_t$ has a fixed point $\hat{u} \leq u^* - \epsilon$ as desired.

Now assume there are infinite $i$s such that $t_i - 1 \neq t_{i-1}$ (i.e. $t_i - 1$ is not part of the sequence $(t_i)_i$). We will show that this case is contradictory. Take $T$ large enough such that $a_t > u^* - \epsilon/2$ and $|f_t(u) - u| < \epsilon/4$ for all $u$ and for all $t \geq T$ ($T$ exists by premise). Let $j$ be the smallest index such that $t_j > T$ and $t_j - 1 \neq t_{j-1}$. By the definition of $j$, $u_{t_j - 1} \geq u^* - \epsilon$. If $u_{t_j - 1} \geq u^* - \epsilon/2$, then by definition of $T$, $u^* - \epsilon > u_{t_j} = f_{t_j}(u_{t_j - 1}) > u_{t_j - 1} - \epsilon/4 > u^* - 3\epsilon/4 > u^* - \epsilon$, a contradiction. If $u^* - \epsilon \leq u_{t_j - 1} \leq u^* - \epsilon/2$, then by the definition of $T$, $u_{t_j - 1} \leq a_{t_j - 1}$ so that $u_{t_j} = f_{t_j}(u_{t_j - 1}) > u_{t_j - 1} \geq u^* - \epsilon$, a contradiction.

The "furthermore" claim is clear from our proof above. $\square$

**Theorem B.11.** *Assume $\phi = \tanh$ in an FRN. Suppose $\mathbf{e}^{(0)} < 1$.*

- *If $\sigma_w = 0$, then $\boldsymbol{\lambda}^{(l)} = \sigma_b^2$ and $\boldsymbol{\gamma}^{(l)} = l(\sigma_v^2 \mathrm{W}\phi(\sigma_b^2, \sigma_b^2) + \sigma_a^2) + \boldsymbol{\gamma}^{(0)} = l(\sigma_v^2 \mathrm{V}\phi(\sigma_b^2) + \sigma_a^2) + \boldsymbol{\gamma}^{(0)}$. Thus $\mathbf{e}^{(l)} \to 1$ and $1 - \mathbf{e}^{(l)} = \Theta(l^{-1})$. As a result, $\mathbf{s}^{(l)} = \mathbf{p}^{(l)}(1 - \mathbf{e}^{(l)}) = \Theta(1)$.*

- *If $\sigma_w > 0$, then $\mathbf{e}^{(l)}$ converges to the unique fixed point $\mathbf{e}^* \neq 1$ determined by the equation*

$$\mathbf{e}^* = \frac{1}{\sigma_v^2 + \sigma_a^2}[\sigma_v^2 \frac{2}{\pi} \arcsin(\mathbf{e}^*) + \sigma_a^2].$$

*Furthermore, $\mathbf{e}^{(l)}$ converges to $\mathbf{e}^*$ polynomially: $|\mathbf{e}^{(l)} - \mathbf{e}^*|$ is $\check{\Theta}(l^{-\delta^*})$, where*

$$\delta^* := 1 - \frac{2}{\pi}\frac{1}{\sqrt{1 - (\mathbf{e}^*)^2}}\frac{\sigma_v^2}{\sigma_v^2 + \sigma_a^2} \in [\frac{2}{\pi} - 1, \frac{1}{2})$$

*Since $\mathbf{e}^* < 1$, $\mathbf{s}^{(l)} = \Theta(\mathbf{p}^{(l)}) = \Theta(l)$.*

*Proof.* The $\sigma_w = 0$ case is obvious. We will assume $\sigma_w > 0$ from here on.

If $\sigma_a = 0$, then $\mathbf{e}^*$ as defined above is 0, and $\mathbf{e} = \frac{\boldsymbol{\gamma}}{\mathbf{p}}$ decreases as $\Theta(l^{\frac{2}{\pi} - 1})$ to 0, by the same reason as before.

So from now on suppose $\sigma_a > 0$. We apply Lemma C.23 first to show that $\mathbf{e}$ converges. We have

$$\sigma_v^2 \mathrm{W}\phi(\mathbf{q}, c q) + \sigma_a^2 = \mathbf{e}\mathbf{p} - \underline{\mathbf{e}}\underline{\mathbf{p}}$$

$$= \mathbf{e}\mathbf{p} - \mathbf{e}\underline{\mathbf{p}} + \mathbf{e}\underline{\mathbf{p}} - \underline{\mathbf{e}}\underline{\mathbf{p}}$$

$$= (\mathbf{e} - \underline{\mathbf{e}})\underline{\mathbf{p}} + \mathbf{e}(\mathbf{p} - \underline{\mathbf{p}})$$

$$= (\mathbf{p} - \underline{\mathbf{p}})[(\mathbf{e} - \underline{\mathbf{e}})\frac{\underline{\mathbf{p}}}{\mathbf{p} - \underline{\mathbf{p}}} + \mathbf{e}]$$

$$\frac{\sigma_v^2 \mathrm{W}\phi(\mathbf{q}, c q) + \sigma_a^2}{\sigma_v^2 \mathrm{V}\phi(\mathbf{q}) + \sigma_a^2} = (\mathbf{e} - \underline{\mathbf{e}})\frac{\underline{\mathbf{p}}}{\mathbf{p} - \underline{\mathbf{p}}} + \underline{\mathbf{e}}$$

$$\frac{\mathbf{p} - \underline{\mathbf{p}}}{\underline{\mathbf{p}}}\left[\frac{\sigma_v^2 \mathrm{W}\phi + \sigma_a^2}{\sigma_v^2 \mathrm{V}\phi + \sigma_a^2} - \underline{\mathbf{e}}\right] = \mathbf{e} - \underline{\mathbf{e}}$$

If we define $f_l(u) := \frac{\mathbf{p}^{(l)} - \mathbf{p}^{(l-1)}}{\mathbf{p}^{(l)}}\left[\frac{\sigma_v^2 \mathrm{W}\phi(\mathbf{q}^{(l)}, c^{(l)}\mathbf{q}^{(l)}) + \sigma_a^2}{\sigma_v^2 \mathrm{V}\phi(\mathbf{q}^{(l)}) + \sigma_a^2} - u\right] + u$ (the LHS of the above), then $f_l(u) - u = O(l^{-1})$ uniformly for all $u$ because $\mathbf{p}^{(l)} = \Theta(l)$, $\mathbf{p}^{(l)} - \mathbf{p}^{(l-1)} = \Theta(1)$, and the part in the bracket is $O(1)$, with constants all (able to be taken) independent of $u$. We divide $[0, 1)$ into the following intervals $I_1 = [1, 1/2), I_2 = [1/2, 3/4), I_3 = [3/4, 7/8), \ldots$. For each $I_k$, it is clear that the trajectories of $\mathbf{e}^{(l)} = f_l(\mathbf{e}^{(l-1)})$ with $\mathbf{e}^{(0)} \in I_k$ will fall into some interval $J_k$ bounded away from 1 for all $l \geq L$, for large enough $L$ (dependent on $k$). Then we can apply Lemmas C.1, C.5 and C.21 to get $f_l(u) = \frac{\mathbf{p}^{(l)} - \mathbf{p}^{(l-1)}}{\mathbf{p}^{(l)}}\left[\frac{\sigma_v^2 \frac{2}{\pi} \arcsin(u) + \sigma_a^2}{\sigma_v^2 + \sigma_a^2} - u + o(1)\right] + u$ where the constants in $o(1)$ is uniform for all $\mathbf{e}^{(0)} \in I_k$. For $u < \mathbf{e}^*$ (as defined in the theorem statement), $\frac{\sigma_v^2 \frac{2}{\pi} \arcsin(u) + \sigma_a^2}{\sigma_v^2 + \sigma_a^2} > u$ and for $u > \mathbf{e}^*$, $\frac{\sigma_v^2 \frac{2}{\pi} \arcsin(u) + \sigma_a^2}{\sigma_v^2 + \sigma_a^2} < u$ (see Fig. C.12). Thus as $l \to \infty$, the $o(1)$ term gets smaller and smaller, and this monotonicity holds for $f_l(u) - u = \left[\frac{\sigma_v^2 \frac{2}{\pi} \arcsin(u) + \sigma_a^2}{\sigma_v^2 + \sigma_a^2} - u + o(1)\right] > 0$ (resp. $< 0$) on larger and larger intervals $[0, a_l] \cap J_k$ (resp. $[b_l, 1) \cap J_k$). This proves all the preconditions for Lemma C.23, which yields that $I_k$ converges to a limit point. As this argument is independent of $k$, we have that for all $\mathbf{e}^{(0)} \in [0, 1)$, $\mathbf{e}^{(l)}$ converges.

Now we solve for the limit point.

Suppose $\mathbf{e}$ has limit point $\mathbf{e}^\dagger$ (possibly different from $\mathbf{e}^*$ described in the theorem); if we express $\boldsymbol{\gamma}^{(l)} = (\mathbf{e}^\dagger + \epsilon^{(l)})\mathbf{p}^{(l)}$, then

$$\sigma_v^2 \mathrm{W}\phi(\mathbf{q}, c q) + \sigma_a^2 = \boldsymbol{\gamma} - \underline{\boldsymbol{\gamma}}$$

$$= (\mathbf{e}^\dagger + \epsilon)\mathbf{p} - (\mathbf{e}^\dagger + \underline{\epsilon})\underline{\mathbf{p}}$$

$$= \mathbf{e}^\dagger(\mathbf{p} - \underline{\mathbf{p}}) + \epsilon\mathbf{p} - \underline{\epsilon}\underline{\mathbf{p}}$$

$$\frac{\sigma_v^2 \mathrm{W}\phi(\mathbf{q}, c q) + \sigma_a^2}{\sigma_v^2 \mathrm{V}\phi(\mathbf{q}) + \sigma_a^2} = \mathbf{e}^\dagger + \epsilon + (\epsilon - \underline{\epsilon})\frac{\underline{\mathbf{p}}}{\mathbf{p} - \underline{\mathbf{p}}}$$

**Figure C.12:** Graph of $y(\mathbf{e}) = \frac{1}{\sigma_v^2+\sigma_a^2}[\sigma_v^2\frac{2}{\pi}\arcsin(\mathbf{e}) + \sigma_a^2]$ for various $\sigma_v$ and $\sigma_a$.

As $l \to \infty$, $c \sim \mathbf{e} \to \mathbf{e}^\dagger$, and $\mathrm{W}\phi(\mathbf{q}, \mathbf{e}^\dagger\mathbf{q}) \to \frac{2}{\pi}\arcsin(\mathbf{e}^\dagger)$, and $\mathrm{V}\phi(\mathbf{q}) \to 1$. Additionally, $\underline{\mathbf{p}}/(\mathbf{p} - \underline{\mathbf{p}}) = \Theta(l)$ and $\epsilon = o(1)$ so that $\epsilon - \underline{\epsilon} = o(l^{-1})$. Then we have, taking limits $l \to \infty$,

$$\frac{\sigma_v^2\frac{2}{\pi}\arcsin(\mathbf{e}^\dagger) + \sigma_a^2}{\sigma_v^2 + \sigma_a^2} = \mathbf{e}^\dagger.$$

Since $f_l$ (as defined above) repels points away from 1, the only solution for $\mathbf{e}^\dagger$ when $\mathbf{e}^{(0)} < 1$ is $\mathbf{e}^\dagger = \mathbf{e}^*$ as specified in the theorem statement.

We defer the proof of the convergence rate to $\mathbf{e}^*$ to Thm C.25.

$\square$

**Lemma C.24.** *Let $\mathbf{e}^*$ be the stable fixed point determined by $\sigma_a$ and $\sigma_v$. Then as long as $\sigma_v > 0$,*

$$\frac{2}{\pi}\frac{1}{\sqrt{1-(\mathbf{e}^*)^2}}\frac{\sigma_v^2}{\sigma_v^2 + \sigma_a^2} \in (\frac{1}{2}, \frac{2}{\pi}]$$

*Proof.* Write $\rho := \frac{\sigma_a^2}{\sigma_v^2}$. By definition of $\mathbf{e}^*$, we get

$$\mathbf{e}^* = (1-\rho)\frac{2}{\pi}\arcsin\mathbf{e}^* + \rho$$

$$\rho = = \frac{\mathbf{e}^* - \frac{2}{\pi}\arcsin\mathbf{e}^*}{1 - \mathbf{e}^*}$$

Substituting $\rho$ into the expression in question, it follows that we want to show

$$\frac{2}{\pi}(1-\mathbf{e}^{*2})^{-1/2}(1+\rho)^{-1} = \frac{2}{\pi}(1-\mathbf{e}^{*2})^{-1/2}\left(\frac{1 - \frac{2}{\pi}\arcsin\mathbf{e}^*}{1 - \mathbf{e}^*}\right)^{-1} \in (\frac{1}{2}, \frac{2}{\pi}]$$

for $\mathbf{e}^* \in [0, 1)$ (the endpoint at 1 is not included since $\sigma_v > 0$. But this is

$$\frac{2}{\pi}(1-\mathbf{e}^*)^{1/2}(1+\mathbf{e}^*)^{-1/2}(1 - \frac{2}{\pi}\arcsin\mathbf{e}^*)^{-1}.$$

Set $g(\mathbf{e}^*)$ to be this expression. We could proceed by finding critical points, but a simple plot Fig. C.13 shows that $g$ is decreasing on $[0, 1)$, with extremal values at the end points:

$$g(\mathbf{e}^*) \in [\lim_{\mathbf{e}^* \to 1} g(\mathbf{e}^*), g(0)), \quad \text{for } \mathbf{e}^* \in [0, 1).$$

Obviously $g(0) = \frac{2}{\pi}$. For the limit, we note that $\arcsin\mathbf{e}^*$ has an asymptotic expansion $\frac{\pi}{2} - \sqrt{2}(1 - e)^{1/2} + \Theta((1-e)^{3/2})$ at 1, so that $(1-\mathbf{e}^*)^{1/2}(1 - \frac{2}{\pi}\arcsin\mathbf{e}^*)^{-1} \to \frac{\pi}{2\sqrt{2}}$, and $g(\mathbf{e}^*) \to \frac{1}{2}$ as $\mathbf{e}^* \to 1$.

$\square$

**Theorem C.25.** *If $\mathbf{e}^{(0)} < 1$, then $|\mathbf{e}^{(l)} - \mathbf{e}^*|$ is $\Omega(l^{-\delta^*-\varepsilon})$ and $O(l^{-\delta^*+\varepsilon})$ for any $\varepsilon > 0$, where*

$$\delta^* := 1 - \frac{2}{\pi}\frac{1}{\sqrt{1-(\mathbf{e}^*)^2}}\frac{\sigma_v^2}{\sigma_v^2 + \sigma_a^2} \in [1 - \frac{2}{\pi}, \frac{1}{2}),$$

*where the bounds on the right follow from Lemma C.24.*

**Figure C.13:** Plot of $g(\mathbf{e}^*)$ in the proof of Lemma C.24

*Proof.* Define $\omega(q, c) = \frac{2}{\pi} \arcsin(c) - \mathrm{W} \tanh(q, cq)$. By Lemma C.21, for large enough $l$, $c$ is close to $\mathbf{e}^*$ bounded away from 0 or 1, so that $\omega(\mathbf{q}, c) = \Theta(\mathbf{q}^{-1})$ with the constant hidden in $\Theta$ independent of $c$. Additionally, by Lemma C.5, $1 - \mathrm{V} \tanh(\mathbf{q}) = \Theta(\mathbf{q}^{-1/2})$. Therefore,

$$(\mathbf{e}^* + \epsilon)\mathbf{p} = \sigma_v^2 (\frac{2}{\pi} \arcsin(\mathbf{e}^* + \underline{\epsilon}) - \omega(\mathbf{q}, c)) + \sigma_a^2 + \underline{\gamma}$$

$$= \sigma_v^2 \frac{2}{\pi} [\arcsin(\mathbf{e}^*) + \frac{\epsilon}{\sqrt{1 - (\mathbf{e}^*)^2}} + \Theta(\underline{\epsilon}^2)] - \Theta(l^{-1}) + \sigma_a^2 + \underline{\gamma}$$

$$= \mathbf{e}^*(\sigma_v^2 + \sigma_a^2) + (\mathbf{e}^* + \underline{\epsilon})\underline{\mathbf{p}} + \sigma_v^2 \frac{2}{\pi} \frac{\epsilon}{\sqrt{1 - (\mathbf{e}^*)^2}} + \Theta(\underline{\epsilon}^2) - \Theta(l^{-1})$$

$$\mathbf{e}^*(\mathbf{p} - \underline{\mathbf{p}} - \sigma_v^2 - \sigma_a^2) = \underline{\epsilon}\underline{\mathbf{p}} - \epsilon\mathbf{p} + \sigma_v^2 \frac{2}{\pi} \frac{\epsilon}{\sqrt{1 - (\mathbf{e}^*)^2}} + \Theta(\underline{\epsilon}^2) - \Theta(l^{-1})$$

$$\mathbf{e}^* \sigma_v^2 (\mathrm{V}\phi(\mathbf{q}) - 1) = \underline{\epsilon}\underline{\mathbf{p}} - \epsilon\mathbf{p} + \sigma_v^2 \frac{2}{\pi} \frac{\epsilon}{\sqrt{1 - (\mathbf{e}^*)^2}} + \Theta(\underline{\epsilon}^2) - \Theta(l^{-1})$$

$$\epsilon = \frac{1}{\mathbf{p}} (\mathbf{e}^* \sigma_v^2 (1 - \mathrm{V}\phi(\mathbf{q})) + \Theta(\underline{\epsilon}^2) - \Theta(l^{-1}) + \underline{\epsilon}(\underline{\mathbf{p}} + \sigma_v^2 \frac{2}{\pi} \frac{1}{\sqrt{1 - (\mathbf{e}^*)^2}}))$$

$$= \Theta(l^{-3/2}) + \underline{\epsilon}(1 - \delta^{(l)}/l)$$

where

$$\delta^{(l)} = \frac{l}{\mathbf{p}} (\sigma_v^2 \mathrm{V}\phi(\mathbf{q}) + \sigma_a^2 - \sigma_v^2 \frac{2}{\pi} \frac{1}{\sqrt{1 - (\mathbf{e}^*)^2}}) + \Theta(\underline{\epsilon}/l)$$

$$= (1 + \Theta(l^{-1/2}))(\sigma_v^2 (1 - \Theta(l^{-1/2})) + \sigma_a^2 - \sigma_v^2 \frac{2}{\pi} \frac{1}{\sqrt{1 - (\mathbf{e}^*)^2}})/(\sigma_v^2 + \sigma_a^2) + \Theta(\underline{\epsilon}/l)$$

$$= \delta^* + O(l^{-1/2}),$$

where $\delta^* := 1 - \frac{2}{\pi} \frac{1}{\sqrt{1 - (\mathbf{e}^*)^2}} \frac{\sigma_v^2}{\sigma_v^2 + \sigma_a^2}$, which is positive by Lemma C.24. By taking the $\delta$ of Lemma C.11 to be $\delta^* + \varepsilon$ or $\delta^* - \varepsilon$ respectively for lower and upper bounding the dynamics of $\epsilon^{(l)}$, the solution $\epsilon^{(l)}$ is $\Omega(l^{-\delta^* - \varepsilon})$ and $O(l^{-\delta^* + \varepsilon})$ for any $\varepsilon > 0$ since $\frac{1}{2} > \delta^*$.

$\square$

### C.6.2 Backward Dynamics

**Theorem B.13.** *Assume* $\phi = \tanh$ *in an FRN.*

- *If* $\sigma_w = 0$, $\chi^{(m)} = \chi^{(l)}$ *for all* $l, m$.

- *If* $\sigma_w > 0$, *then for* $l \geq m \geq 0$,

$$\log(\chi^{(m)}/\chi^{(l)}) = \mathcal{A}(\sqrt{l} - \sqrt{m}) + \mathcal{B}(\log l - \log m) + O(1)$$

*where*

$$\mathcal{A} = \frac{4}{3}\sqrt{\frac{2}{\pi}} \frac{\sigma_v^2 \sigma_w}{\sqrt{\sigma_v^2 + \sigma_a^2}}$$

$$\mathcal{B} = \frac{4}{9\pi} \frac{\sigma_v^4}{\sigma_v^2 + \sigma_a^2} \left( \frac{3}{\sigma_v^2 + \sigma_a^2} - \sigma_w^2 \right)$$

*Proof.* The $\sigma_w = 0$ case is obvious. We will assume $\sigma_w > 0$ from here on.

As in the proof of Thm B.6,

$$\log(\boldsymbol{\chi}^{(m)}/\boldsymbol{\chi}^{(l)}) = 2BD\sigma_w^{-1}b_0^{-1/2}(\sqrt{l} - \sqrt{m})$$
$$- (BD\sigma_w^{-1}b_0^{-3/2}b_1 2^{-1} + B^2 D^2 \sigma_w^{-2} b_0^{-1} 2^{-1})(\log l - \log m) + O(1)$$

where $B = \sigma_v^2 \sigma_w^2, D = \frac{2}{3}\sqrt{\frac{2}{\pi}}$,

$$b_0 = \sigma_v^2 + \sigma_a^2$$
$$b_1 = \frac{-2C\sigma_v^2 \sigma_w^{-1}}{\sqrt{\sigma_v^2 + \sigma_a^2}}$$
$$b_2 = \frac{-C^2 \sigma_v^4 \sigma_w^{-2}}{(\sigma_v^2 + \sigma_a^2)^2}.$$

with $C = \sqrt{\frac{2}{\pi}}$. This simplifies to the desired form. $\qquad\square$

**Theorem B.14.** *Suppose* $\phi = \tanh$ *in an FRN.*

- *If* $\sigma_w = 0$, *then*

$$\boldsymbol{\chi}_b^{(l)} = \sigma_v^2 \boldsymbol{\chi}^{(L)} \mathrm{V}\dot{\phi}(\sigma_b^2)$$
$$\boldsymbol{\chi}_w^{(l)} = \sigma_v^2 \boldsymbol{\chi}^{(L)} \mathrm{V}\dot{\phi}(\sigma_b^2)((\sigma_v^2 \mathrm{V}\phi(\sigma_b^2) + \sigma_a^2)(l-1) + \mathbf{p}^{(0)})$$
$$\boldsymbol{\chi}_v^{(l)} = \boldsymbol{\chi}^{(L)} \mathrm{V}\phi(\sigma_b^2)$$
$$\boldsymbol{\chi}_a^{(l)} = \boldsymbol{\chi}^{(L)}.$$

- *If* $\sigma_w > 0$, *then for* $l \geq m \geq 0$,

$$\log(\boldsymbol{\chi}_b^{(m)}/\boldsymbol{\chi}_b^{(l)}) = \mathcal{A}(\sqrt{l} - \sqrt{m}) + \mathcal{B}_b(\log l - \log m) + O(1)$$
$$\log(\boldsymbol{\chi}_w^{(m)}/\boldsymbol{\chi}_w^{(l)}) = \mathcal{A}(\sqrt{l} - \sqrt{m}) + \mathcal{B}_w(\log l - \log m) + O(1)$$
$$\log(\boldsymbol{\chi}_a^{(m)}/\boldsymbol{\chi}_a^{(l)}) = \mathcal{A}(\sqrt{l} - \sqrt{m}) + \mathcal{B}(\log l - \log m) + O(1)$$
$$\log(\boldsymbol{\chi}_v^{(m)}/\boldsymbol{\chi}_v^{(l)}) = \mathcal{A}(\sqrt{l} - \sqrt{m}) + \mathcal{B}(\log l - \log m) + O(1)$$

*where* $\mathcal{A} = \frac{4}{3}\sqrt{\frac{2}{\pi}} \frac{\sigma_v^2 \sigma_w}{\sqrt{\sigma_v^2 + \sigma_a^2}}$ *and* $\mathcal{B} = \frac{4}{9\pi} \frac{\sigma_v^4}{\sigma_v^2 + \sigma_a^2} \left( \frac{3}{\sigma_v^2 + \sigma_a^2} - \sigma_w^2 \right)$ *are as in Thm B.13 and* $\mathcal{B}_b = \mathcal{B} + \frac{1}{2}$ *and* $\mathcal{B}_w = \mathcal{B} - \frac{1}{2}$.

*Proof.* Similar to Thm B.7. $\qquad\square$

### C.7  $\alpha$-ReLU: Full Residual Network

The following can be checked readily

**Lemma B.15.** *If* $\alpha > -\frac{1}{2}$, *then* $\mathrm{V}\psi_\alpha(q) = \mathsf{c}_\alpha q^\alpha$, *where* $\mathsf{c}_\alpha = \frac{1}{\sqrt{\pi}} 2^{\alpha-1} \Gamma\left(\alpha + \frac{1}{2}\right)$.

Since $\dot{\psi}_\alpha = \alpha \psi_{\alpha-1}$, we have as a corollary,

**Figure C.14:** Verification of leading term of Thm C.28 for $\alpha = 0.55$.

**Lemma C.26.** *If $\alpha > \frac{1}{2}$, then* $\mathrm{V}\dot{\psi}_\alpha(q) = \alpha^2 \mathsf{c}_{\alpha-1} q^{\alpha-1}$.

As a special case, when $\alpha = 1$, $\mathsf{c}_\alpha = \frac{1}{2}$.

The following is a trivial computation, but useful for many simplifications.

**Lemma C.27.** $\mathsf{c}_{\alpha+1}/\mathsf{c}_\alpha = 2\alpha + 1$.

### C.7.1 Forward Dynamics

**Theorem C.28.** *Suppose we have the nonlinearity $\phi = \psi_1$. Then $\mathbf{p}^{(l)} = \Theta((1 + \sigma_v^2 \sigma_w^2/2)^l)$, with the hidden constant depending on the initial condition.*

*Proof.* We have

$$\mathbf{p} = \frac{1}{2}\sigma_v^2(\sigma_w^2 \underline{\mathbf{p}} + \sigma_b^2) + \sigma_a^2 + \underline{\mathbf{p}}$$

$$= (\frac{1}{2}\sigma_v^2\sigma_w^2 + 1)\underline{\mathbf{p}} + \frac{1}{2}(\sigma_v^2\sigma_b^2 + \sigma_a^2).$$

By the standard method of characteristic equation, we get that

$$\mathbf{p}^{(l)} = A + CB^l$$

where $A = -\frac{\sigma_a^2 + \sigma_b^2\sigma_v^2}{\sigma_v^2\sigma_w^2}$, $B = 1 + \frac{\sigma_v^2\sigma_w^2}{2}$, and $C$ is a coefficient determined by initial conditions. $\qquad\square$

**Theorem C.29.** *Suppose $\alpha < 1$. We have the following asymptotic expansion*

$$\mathbf{p}^{(l)} = K_1 l^{\frac{1}{1-\alpha}} + R(l)$$

*where the remainder term*

$$R(l) \sim \begin{cases} -K_2 l^{\frac{\alpha}{1-\alpha}} \log l & \text{if } \alpha > \frac{1}{2} \\ (C - K_2)l \log l & \text{if } \alpha = \frac{1}{2} \text{ and } K_2 \neq C \\ \frac{C(1-\alpha)}{1-2\alpha} l & \text{if } \alpha < \frac{1}{2} \end{cases}$$

*where $K_1 = [\sigma_v^2 \sigma_w^{2\alpha} \mathsf{c}_\alpha (1-\alpha)]^{\frac{1}{1-\alpha}}$, $K_2 = \frac{1}{2}[\sigma_v^2 \mathsf{c}_\alpha \sigma_w^{2\alpha}]^{\frac{1}{1-\alpha}}(1-\alpha)^{\frac{\alpha}{1-\alpha}-1}\alpha$ and $C = \sigma_a^2$.*

Fig. C.14 verifies the leading coefficient and the exponent of the leading term.

*Proof.* The difference equation governing the evolution of $\mathbf{p}$ is

$$\mathbf{p} - \underline{\mathbf{p}} = A(\underline{\mathbf{p}} + B)^\alpha + C$$

where $A = \sigma_v^2 \mathsf{c}_\alpha \sigma_w^{2\alpha}$, $B = \sigma_b^2/\sigma_w^2$, and $C = \sigma_a^2$. Then Lemma C.15 yields the result. $\qquad\square$

Thm C.29 combined with Thm C.28 gives the following result.

**Figure C.15:** (a) $\mathbb{J}_\alpha$ for different $\alpha$s and the identity function. From this plot, it looks like $\mathbb{J}_\alpha(c) \geq c$ and $\dot{\mathbb{J}}_\alpha(c) \leq 1$ for all $\alpha \in (\frac{1}{2}, 1]$ with equality iff $c = 1$, but this is misleading. (b) shows $|\mathbb{J}_\alpha(c) - c|$ in log scale. Where the curves dip below the x-axis indicate points where $\mathbb{J}_\alpha(c) = c$. We see that in fact every $\mathbb{J}_\alpha$ has a solution $\mathbb{J}_\alpha(c) = c$ for a $c < 1$, when $\alpha < 1$. (c) Furthermore, at each such $c$, $\dot{\mathbb{J}}_\alpha < 1$.

**Figure C.16:** $\mathbb{J}_1$ vs identity

**Theorem B.16.** *Suppose we have the nonlinearity $\phi = \psi_\alpha$. The in an FRN: If $\alpha = 1$, then $\mathbf{p}^{(l)} = \Theta((1 + \sigma_v^2\sigma_w^2/2)^l)$, with the hidden constant depending on the initial condition. If $0 < \alpha < 1$, then $\mathbf{p}^{(l)} = \Theta(l^{\frac{1}{1-\alpha}})$. More precisely, $\lim_{l\to\infty} \mathbf{p}/l^{\frac{1}{1-\alpha}} = [\sigma_v^2\sigma_w^{2\alpha}\mathsf{c}_\alpha(1-\alpha)]^{\frac{1}{1-\alpha}}$.*

By [2], we know that $\mathrm{W}\psi_\alpha(q, qc) = \mathrm{V}\psi_\alpha(q)\mathbb{J}_\alpha(c)$, where $\mathbb{J}_\alpha(c) = J_\alpha(\arccos c)$ and

$$J_\alpha(\theta) := \frac{1}{2\pi\mathsf{c}_\alpha}(\sin\theta)^{2\alpha+1}\Gamma(\alpha+1)\int_0^{\pi/2} \frac{\mathrm{d}\eta\cos^\alpha\eta}{(1 - \cos\theta\cos\eta)^{1+\alpha}}. \qquad (\triangle)$$

Note that $\mathbb{J}_\alpha(c) \in (-\infty, \infty)$ for $\alpha \in (-1, \infty)$ and any $c \in (0, 1)$, even though $\mathrm{V}\psi_\alpha$ is only defined for $\alpha > -1/2$.

Fig. C.15 shows a comparison of $\mathbb{J}_\alpha$ for different $\alpha$s along with the identity function. By [3, Lemma 11], $\mathbb{J}_\alpha$ is an increasing and convex function as long as $\psi_\alpha^2$ is Gaussian-integrable, which is precisely when $\alpha > -1/2$. We can compute $\mathbb{J}_\alpha(1) = \mathrm{W}\psi_\alpha(q, q)/\mathrm{V}\psi_\alpha(q) = 1$, and $\mathbb{J}_\alpha(0) = \mathrm{W}\psi_\alpha(q, 0)/\mathrm{V}\psi_\alpha(q) = \mathrm{V}\psi_{\alpha/2}(q)^2/\mathrm{V}\psi_\alpha(q) = \mathsf{c}_{\alpha/2}^2/\mathsf{c}_\alpha = \frac{1}{2\sqrt{\pi}}\frac{\Gamma(\frac{\alpha}{2}+\frac{1}{2})^2}{\Gamma(\alpha+\frac{1}{2})}$. We record these observations as a lemma.

**Lemma C.30.** *$\mathbb{J}_\alpha(c)$ is an increasing and convex function for each $\alpha > -1/2$ on $c \in [0, 1]$. $\mathbb{J}_\alpha(1) = 1$ and $\mathbb{J}_\alpha(0) = \frac{1}{2\sqrt{\pi}}\frac{\Gamma(\frac{\alpha}{2}+\frac{1}{2})^2}{\Gamma(\alpha+\frac{1}{2})}$.*

For $\alpha = 1$, Cho and Saul [2] computed

$$\mathbb{J}_1(c) = \frac{1}{\pi}(\sqrt{1 - c^2} + (\pi - \arccos(c))c).$$

Fig. C.16 shows a plot of $\mathbb{J}_1$ vs identity. It has derivative $\dot{\mathbb{J}}_1(c) = 1 - \frac{1}{\pi}\arccos c$, which shows that $\dot{\mathbb{J}}_1(c) < 1$ with equality iff $c = 1$, and consequently $\mathbb{J}_1(c) \geq c$ with equality iff $c = 1$. At the same time, $\dot{\mathbb{J}}_1(c) \geq 0$ with equality iff $c = -1$, so $\mathbb{J}_1$ is increasing on $[-1, 1]$. It has an asymptotic expansion $\mathbb{J}_1(1 - \varepsilon) = 1 - \varepsilon + \frac{2\sqrt{2}}{3\pi}\epsilon^{3/2} + \Theta(\epsilon^{5/2})$ at 1.

The zeroth Bessel function of the second kind is defined by $\mathcal{K}_0(z) = \int_1^\infty e^{-zx}(x^2 - 1)^{-1/2}\,\mathrm{d}x$. It is one of the fundamental solutions to the homogeneous differential equation $x^2\ddot{y} + x\dot{y} - x^2y = 0$. The following lemma shows that $J_\alpha$ can be expressed in terms of $\mathcal{K}_0$.

**Lemma C.31.** *For any $\alpha > -1$, $J_\alpha(\theta) = \frac{1}{2\pi\mathsf{c}_\alpha}\sin^{2\alpha+1}\theta\int_0^\infty \mathrm{d}x\mathcal{K}_0(x)e^{x\cos\theta}x^\alpha$*

*Proof.* Cho and Saul [2] gave the expression

$$2\pi\mathsf{c}_\alpha J_\alpha(\theta) = \csc\theta\int_0^\infty \mathrm{d}u\int_0^\infty \mathrm{d}v e^{-(u^2+v^2-2uv\cos\theta)/2\sin^2\theta}u^\alpha v^\alpha.$$

Note that the integrand is symmetric in $u$ and $v$. Thus, if $\mathsf{V} = \{(u,v) : u, v \geq 0 \ \& \ v \geq u\}$, then

$$2\pi \mathsf{c}_\alpha J_\alpha(\theta) = 2\csc\theta \int_{\mathsf{V}} \mathrm{d}u\,\mathrm{d}v e^{-(u^2+v^2-2uv\cos\theta)/2\sin^2\theta} u^\alpha v^\alpha.$$

Now make the change of variables from $\mathsf{V}$ to $\{(\mathbb{p},\mathbb{q}) : \mathbb{q} \geq 2\sqrt{\mathbb{p}}\}$:

$$\mathbb{p} = uv \qquad\qquad\qquad \mathrm{d}\mathbb{p} = v\,\mathrm{d}u + u\,\mathrm{d}v$$
$$\mathbb{q} = u + v \qquad\qquad\qquad \mathrm{d}\mathbb{q} = \mathrm{d}u + \mathrm{d}v$$
$$\mathrm{d}\mathbb{p}\,\mathrm{d}\mathbb{q} = (v-u)\,\mathrm{d}u\,\mathrm{d}v \qquad\qquad \mathrm{d}u\,\mathrm{d}v = (\mathbb{q}^2 - 4\mathbb{p})^{-1/2}\,\mathrm{d}\mathbb{p}\,\mathrm{d}\mathbb{q}$$

so that we have

$$2\pi\mathsf{c}_\alpha J_\alpha(\theta) = 2\csc\theta \int_0^\infty \mathrm{d}\mathbb{p}\, e^{\mathbb{p}(1+\cos\theta)\csc^2\theta}\mathbb{p}^\alpha \int_{2\sqrt{\mathbb{p}}}^\infty \mathrm{d}\mathbb{q}\, e^{-\mathbb{q}^2\csc^2\theta}(\mathbb{q}^2 - 4\mathbb{p})^{-1/2}.$$

The inner integral in $\mathbb{q}$ can be expressed in terms of $\mathcal{K}_0$ by a change of variable $x = \mathbb{q}^2/2\sqrt{\mathbb{p}}$:

$$2\pi\mathsf{c}_\alpha J_\alpha(\theta) = 2\csc\theta \int_0^\infty \mathrm{d}\mathbb{p}\, e^{\mathbb{p}(1+\cos\theta)\csc^2\theta}\mathbb{p}^\alpha \frac{1}{2}e^{-\mathbb{p}\csc^2\theta}\mathcal{K}_0(\mathbb{p}\csc^2\theta)$$

$$= \csc\theta \int_0^\infty \mathrm{d}\mathbb{p}\,\mathcal{K}_0(\mathbb{p}\csc^2\theta)e^{\mathbb{p}\cos\theta\csc^2\theta}\mathbb{p}^\alpha$$

$$= \sin^{2\alpha+1}\theta \int_0^\infty \mathrm{d}x\,\mathcal{K}_0(x)e^{x\cos\theta}x^\alpha$$

$\square$

Define $L_\alpha(\theta) = 2\pi\mathsf{c}_\alpha J_\alpha(\theta)\csc^{2\alpha+1}\theta = \int_0^\infty \mathrm{d}x\,\mathcal{K}_0(x)e^{x\cos\theta}x^\alpha$.

**Lemma C.32.** *If $\alpha > 1$, then*

$$L_\alpha(\theta) = \csc^2\theta[(2\alpha-1)\cos\theta L_{\alpha-1}(\theta) + (\alpha-1)^2 L_{\alpha-2}(\theta)].$$

*Proof.* We will prove this claim for $\theta < 1$, and by continuity this also proves the case $\theta = 1$. As remarked above, $\mathcal{K}_0(z) = \ddot{\mathcal{K}}_0(z) + z^{-1}\dot{\mathcal{K}}_0(z)$. Thus

$$L_\alpha(\theta) = \int_0^\infty \mathrm{d}x(\ddot{\mathcal{K}}_0(x) + x^{-1}\dot{\mathcal{K}}_0(x))e^{x\cos\theta}x^\alpha$$

$$= \dot{\mathcal{K}}_0 e^{x\cos\theta}x^\alpha|_0^\infty + \mathcal{K}_0 e^{x\cos\theta}x^{\alpha-1}|_0^\infty$$

$$\quad - \int \mathrm{d}x[\cos\theta e^{x\cos\theta}x^\alpha + \alpha e^{x\cos\theta}x^{\alpha-1}]\dot{\mathcal{K}}_0$$

$$\quad - \int \mathrm{d}x[\cos\theta e^{x\cos\theta}x^{\alpha-1} + (\alpha-1)e^{x\cos\theta}x^{\alpha-2}]\mathcal{K}_0$$

Asymptotically, $\mathcal{K}_0(z) \sim \sqrt{\frac{\pi}{2z}}e^{-z}$ as $z \to \infty$ and $\mathcal{K}_0(z) \sim -\ln(z)$ as $z \searrow 0$, and $\dot{\mathcal{K}}_0(z) \sim -\sqrt{\frac{\pi}{2z}}e^{-z}$ as $z \to \infty$ and $\dot{\mathcal{K}}_0(z) \sim -z^{-1}$ as $z \searrow 0$. Thus, as $\alpha > 1$,

$$\dot{\mathcal{K}}_0 e^{x\cos\theta}x^\alpha|_0^\infty = -\lim_{x\to\infty}\sqrt{\pi/2}e^{-x(1-\cos\theta)}x^{\alpha-1} + \lim_{x\searrow 0}e^{x\cos\theta}x^{\alpha-1} = 0$$

$$\mathcal{K}_0 e^{x\cos\theta}x^{\alpha-1}|_0^\infty = -\lim_{x\to\infty}\sqrt{\pi/2}e^{-x(1-\cos\theta)}x^{\alpha-2} + \lim_{x\searrow 0}e^{x\cos\theta}x^{\alpha-1}\ln x = 0$$

So

$$L_\alpha(\theta) = -\cos\theta L_{\alpha-1}(\theta) - (\alpha-1)L_{\alpha-2}(\theta) - \int \mathrm{d}x[\cos\theta e^{x\cos\theta}x^\alpha + \alpha e^{x\cos\theta}x^{\alpha-1}]\dot{\mathcal{K}}_0$$

Via another integration by parts, the integral on the right is

$$\cos\theta e^{x\cos\theta}x^\alpha\mathcal{K}_0|_0^\infty + \alpha e^{x\cos\theta}x^{\alpha-1}\mathcal{K}_0|_0^\infty$$

$$\quad - \int \mathrm{d}x[\cos^2\theta e^{x\cos\theta}x^\alpha + 2\alpha\cos\theta e^{x\cos\theta}x^{\alpha-1} + \alpha(\alpha-1)e^{x\cos\theta}x^{\alpha-2}]\mathcal{K}_0$$

$$= -[\cos^2\theta L_\alpha(\theta) + 2\alpha\cos\theta L_{\alpha-1}(\theta) + \alpha(\alpha-1)L_{\alpha-2}(\theta)]$$

where the evaluation terms vanish just like before. Altogether, we have

$$L_\alpha(\theta) = \cos^2 \theta L_\alpha(\theta) + (2\alpha - 1)\cos\theta L_{\alpha-1}(\theta) + (\alpha - 1)^2 L_{\alpha-2}(\theta)$$
$$= \csc^2 \theta[(2\alpha - 1)\cos\theta L_{\alpha-1}(\theta) + (\alpha - 1)^2 L_{\alpha-2}(\theta)]$$

$\square$

As a corollary we get

**Lemma C.33.** *Suppose $\alpha > 1$. Then*

$$J_\alpha(\theta) = \cos\theta J_{\alpha-1}(\theta) + (\alpha - 1)^2(2\alpha - 1)^{-1}(2\alpha - 3)^{-1}\sin^2\theta J_{\alpha-2}(\theta)$$
$$\mathbb{J}_\alpha(c) = c\mathbb{J}_{\alpha-1}(c) + (\alpha - 1)^2(2\alpha - 1)^{-1}(2\alpha - 3)^{-1}(1 - c^2)\mathbb{J}_{\alpha-2}(c)$$

The derivative of $J_\alpha(\theta)$ turns out to be quite simple.

**Lemma C.34.** *Suppose $\alpha > 0$. Then*

$$\dot{J}_\alpha(\theta) = -\alpha^2(2\alpha - 1)^{-1}J_{\alpha-1}(\theta)\sin\theta$$
$$\dot{\mathbb{J}}_\alpha(c) = \alpha^2(2\alpha - 1)^{-1}\mathbb{J}_{\alpha-1}(c)$$

*Proof.* We will prove the first formula. The second follows from chain rule. By Lemma C.31,

$$J_\alpha(\theta) = \frac{1}{2\pi c_\alpha}\sin^{2\alpha+1}\theta\int dx \mathcal{K}_0(x)e^{x\cos\theta}x^\alpha$$
$$\dot{J}_\alpha(\theta) = \frac{1}{2\pi c_\alpha}[(2\alpha + 1)\sin^{2\alpha}\theta\cos\theta\int dx \mathcal{K}_0(x)e^{x\cos\theta}x^\alpha$$
$$- \sin^{2\alpha+2}\theta\int dx \mathcal{K}_0(x)e^{x\cos\theta}x^{\alpha+1}]$$
$$= (2\alpha + 1)\cot\theta J_\alpha(\theta) - \frac{c_{\alpha+1}}{c_\alpha}\csc\theta J_{\alpha+1}(\theta)$$
$$= (2\alpha + 1)\csc\theta[\cos\theta J_\alpha(\theta) - J_{\alpha+1}(\theta)].$$

As $\alpha + 1 > 1$, by Lemma C.33, this is

$$- (2\alpha + 1)\csc\theta[(\alpha - 1)^2(2\alpha + 1)^{-1}(2\alpha - 1)^{-1}\sin^2\theta J_{\alpha-1}(\theta)]$$
$$= -(\alpha - 1)^2(2\alpha - 1)^{-1}\sin\theta J_{\alpha-1}(\theta).$$

$\square$

Thus $\dot{\mathbb{J}}_\alpha(1) = \alpha^2(2\alpha - 1)^{-1}\mathbb{J}_{\alpha-1}(1) = \alpha^2(2\alpha - 1)^{-1}$ for any $\alpha > 0$ by Lemma C.30. For $1/2 < \alpha \leq 1$, $\dot{\mathbb{J}}_\alpha(1) \geq 1$ with equality iff $\alpha = 1$, and for $\alpha = 1/2$, $\dot{\mathbb{J}}_\alpha(1) = \infty > 1$ by continuity of $\dot{\mathbb{J}}_\alpha(c)$ in $\alpha$. Because for $\alpha > -1/2$, $\mathbb{J}_\alpha$ is increasing and convex on $[0,1]$ and $\mathbb{J}_\alpha(0) > 0$ by Lemma C.30, $\mathbb{J}_\alpha$ intersects identity at a unique point away from 1 when $\alpha \in [1/2, 1)$. We record this as a theorem.

**Theorem C.35.** *For $\alpha \in [1/2, 1)$, $\mathbb{J}_\alpha(c) = c$ has two solutions: an unstable solution at 1 ("unstable" meaning $\dot{\mathbb{J}}_\alpha(1) > 1$) and a stable solution in $\mathbf{e}^* \in (0,1)$ ("stable" meaning $\dot{\mathbb{J}}_\alpha(\mathbf{e}^*) < 1$).*

This result confirms that pictures presented in Fig. C.17b,c are qualitatively correct, that there are indeed stable fixed points of $\mathbb{J}_\alpha$ away from 1.

**Theorem B.17.** *Suppose $\phi = \psi_1$. Then in an FRN, $\mathbf{e}^{(l)} \to 1$ and $1 - \mathbf{e}^{(l)} \sim [\frac{1}{4}\sigma_v^2\sigma_w^2 B^{-1}Ul]^{-2}$ for $B = 1 + \sigma_v^2\sigma_w^2/2$ and $U = \frac{2\sqrt{2}}{3\pi}$. As a result, $\mathbf{s}^{(l)} = (1-\mathbf{e}^{(l)})\mathbf{p}^{(l)} = \Theta(l^{-2}\exp(\Theta(l))) = \exp(\Theta(l))$.*

*Proof.* If $\underline{\mathbf{e}} < 1$, then

$$c = \frac{\sigma_w^2\underline{\gamma} + \sigma_b^2}{\sigma_w^2\underline{\mathbf{p}} + \sigma_b^2} \geq \underline{\mathbf{e}}$$
$$\mathbb{J}_1(c) \geq \mathbb{J}_1(\underline{\mathbf{e}})$$
$$\mathbf{e} = \frac{\sigma_v^2 c_\alpha \mathbf{q}^\alpha \mathbb{J}_1(c) + \sigma_b^2}{\sigma_v^2 c_\alpha \mathbf{q}^\alpha + \sigma_b^2} \geq \mathbb{J}_1(\underline{\mathbf{e}})$$

**Figure C.17:** Left-to-right: **(a)** $\mathbb{J}_\alpha$ for different $\alpha$s and the identity function (black, dashed line). $\mathbb{J}_1$ is highlighted in red. From this plot, it looks like $\mathbb{J}_\alpha(c) \geq c$ and $\dot{\mathbb{J}}_\alpha(c) \leq 1$ for all $\alpha \in (\frac{1}{2}, 1]$ with equality iff $c = 1$, but this is misleading. **(b)** shows $|\mathbb{J}_\alpha(c) - c|$ in log scale. Where the curves dip below the x-axis indicate points where $\mathbb{J}_\alpha(c) = c$. We see that in fact every $\mathbb{J}_\alpha$ has a solution $\mathbb{J}_\alpha(c) = c$ for a $c < 1$, when $\alpha < 1$. **(c)** Furthermore, at each such $c$, $\dot{\mathbb{J}}_\alpha < 1$. (b) and (c) demonstrate the existence of stable fixed points away from 1 for $\mathbb{J}_\alpha, \alpha \in (1/2, 1)$, which is confirmed rigorously by Thm C.35.

but $\mathbf{e} \geq \mathbb{J}_1(\underline{\mathbf{e}}) > \underline{\mathbf{e}}$ as noted above. Thus by monotone convergence $\mathbf{e}$ converges, and $\mathbf{e}^* = 1$ is the only possible fixed point.

By Lemma C.1, $c = \underline{\mathbf{e}}(1 + \Theta(\underline{\epsilon}\mathbf{p}^{-1})) = 1 - \underline{\epsilon} + \Theta(\underline{\epsilon}\mathbf{p}^{-1}) = 1 - u\underline{\epsilon}$ where $u := 1 - \Theta(\underline{\mathbf{p}}^{-1})$. Using the asymptotic expansion $\mathbb{J}_1(1 - \epsilon) = 1 - \epsilon + U\epsilon^{3/2} + \Theta(\epsilon^{5/2})$, we have

$$(1 - \epsilon)\mathbf{p} = \sigma_v^2 \frac{\mathbf{q}}{2}\mathbb{J}_1(1 - u\underline{\epsilon}) + \sigma_a^2 + (1 - \underline{\epsilon})\underline{\mathbf{p}}$$

$$-\epsilon\mathbf{p} = \sigma_v^2 \frac{\mathbf{q}}{2}(\mathbb{J}_1(1 - u\underline{\epsilon}) - 1) - \underline{\epsilon}\underline{\mathbf{p}}$$

$$= \sigma_v^2 \frac{\mathbf{q}}{2}[-u\underline{\epsilon} + Uu^{3/2}\underline{\epsilon}^{3/2} + \Theta(u^{5/2}\underline{\epsilon}^{5/2})] - \underline{\epsilon}\underline{\mathbf{p}}$$

$$\epsilon = \underline{\epsilon}\frac{1}{\mathbf{p}}[\underline{\mathbf{p}} + \sigma_v^2 \frac{\mathbf{q}}{2}(u - Uu^{3/2}\underline{\epsilon}^{1/2} + \Theta(u^{5/2}\underline{\epsilon}^{3/2}))]$$

$$= \underline{\epsilon}\frac{1}{\mathbf{p}}[\mathbf{p} - \sigma_a^2 + \sigma_v^2 \frac{\mathbf{q}}{2}(\Theta(\underline{\mathbf{p}}^{-1}) - Uu^{3/2}\underline{\epsilon}^{1/2} + \Theta(u^{5/2}\underline{\epsilon}^{3/2}))]$$

$$= \underline{\epsilon}[1 + \frac{-\sigma_a^2 + \sigma_v^2 \frac{\mathbf{q}}{2}(\Theta(\underline{\mathbf{p}}^{-1}) - Uu^{3/2}\underline{\epsilon}^{1/2} + \Theta(u^{5/2}\underline{\epsilon}^{3/2}))}{\mathbf{p}}]$$

$$= \underline{\epsilon}[1 + \frac{-\sigma_a^2\mathbf{q}^{-1} + \frac{1}{2}\sigma_v^2(\Theta(\underline{\mathbf{p}}^{-1}) - Uu^{3/2}\underline{\epsilon}^{1/2} + \Theta(u^{5/2}\underline{\epsilon}^{3/2}))}{\mathbf{p}\mathbf{q}^{-1}}]$$

Let the content of the bracket on the RHS be $\aleph$. We have $\mathbf{pq}^{-1} = (1 + o(1))B/\sigma_w^2$. If $\epsilon = O(\mathbf{p}^{-1})$, then $\aleph = 1 - O(\mathbf{p}^{-1})$, but because $\mathbf{p}$ is exponentially decreasing, this means $\epsilon = \Theta(1)$ and does not converge to 0 — this is a contradiction. Therefore, $\underline{\epsilon} = \omega(\underline{\mathbf{p}}^{-1})$, and

$$\epsilon = \underline{\epsilon}[1 - \frac{1}{2}B^{-1}\sigma_v^2\sigma_w^2U\underline{\epsilon}^{1/2}(1 + o(1))]$$

$$\epsilon - \underline{\epsilon} = -\frac{1}{2}B^{-1}\sigma_v^2\sigma_w^2U\underline{\epsilon}^{3/2}(1 + o(1))$$

Using Lemma C.14 to upper and lower bound our dynamics, we get that $\epsilon^{(l)} \sim [\frac{1}{4}\sigma_v^2\sigma_w^2B^{-1}Ul]^{-2}$. $\square$

**Lemma C.36.** *Let $\phi$ be any nonlinearity. Suppose $\mathrm{W}\phi(r, rd) = \mathrm{V}\phi(r)\mathbb{K}(d)$ for some twice differentiable function $\mathbb{K}(d)$ independent of $\mathbf{q}$, where $\mathbb{K}(1) = 1$ naturally. Suppose further that*

- *$\mathbb{K}(d) = d$ has a solution $d = \mathbf{e}^* > 0$ where $\dot{\mathbb{K}}(\mathbf{e}^*) = \delta < 1$;*

- *$\mathbb{K}(d) > d$ for all $d < \mathbf{e}^*$ and $\mathbb{K}(d) < d$ for all $1 > d > \mathbf{e}^*$; and*

- *$\mathbb{K}$ is nondecreasing.*

*Let $\epsilon^{(l)} := \mathbf{e}^{(l)} - \mathbf{e}^*$ and suppose $\mathbf{e}^{(0)} < 1$. If $\gamma^{(l)} \to \infty$ and $\mathrm{V}\phi(\mathbf{q}^{(l)}) \to \infty$, then $\epsilon^{(l)} \to 0$ and satisfies*

$$\epsilon = \underline{\epsilon}\left(1 - \frac{\sigma_a^2 + (1 - \delta + O(\epsilon))\sigma_v^2\mathrm{V}\phi(\mathbf{q})}{\mathbf{p}}\right) + \mathrm{V}\phi(\mathbf{q})\Theta(\gamma^{-1}\mathbf{p}^{-1}).$$

*Proof.* First we note that because $\mathbf{e}^*$ is the only stable fixed point of the dynamics $x \mapsto \mathbb{K}(x)$, with the basin of attraction $[0, 1)$, we can show $\mathbf{e}^{(l)} \to \mathbf{e}^*$ as in the proof of Thm B.11 (using Lemma C.23).

Write $V^{(l)} := \mathrm{V}\phi(\mathbf{q}^{(l)})$. We first show that $\mathbf{e}^{(l)} \to \mathbf{e}^*$. When $l$ is large,

$$c = \frac{\sigma_w^2 \underline{\gamma} + \sigma_b^2}{\sigma_w^2 \underline{\mathbf{p}} + \sigma_b^2} = \underline{\mathbf{e}}(1 + O(\boldsymbol{\gamma}^{-1}))$$

$$\mathbf{e} = \frac{\sigma_v^2 V \mathbb{K}(c) + \sigma_a^2}{\sigma_v^2 V + \sigma_a^2} = \mathbb{K}(c)(1 + O(V^{-1}\mathbb{K}(c)^{-1})).$$

If $\boldsymbol{\gamma}^{(l)}$ is bounded for all $l$, then $\mathbf{e} \to 0$ because $\mathbf{p}^{(l)} \to \infty$. Since $\mathbb{K}(c) > 0$ for $c \in [0, 1]$ and $V^{(l)} \to \infty$, we have that in the limit $l \to \infty$, $\lim_{l \to \infty} \mathbf{e} = 0 = \mathbb{K}(\lim_{l \to \infty} \mathbf{e}) = \mathbb{K}(0)$ (by the continuity of $\mathbb{K}$), which is impossible by our assumptions. Thus $\boldsymbol{\gamma}^{(l)} \to \infty$, and we have $\lim_{l \to \infty} \mathbf{e} = \mathbb{K}(\lim_{l \to \infty} \mathbf{e})$. By our assumptions, $\mathbf{e}^*$ is the only stable fixed point of $\mathbb{K}$ with basin of attraction $[0, 1)$, so this shows that $\mathbf{e} \to \mathbf{e}^*$ as desired.

Now we derive the equation in question. Note that $c = \underline{\mathbf{e}}(1 + \Theta(\boldsymbol{\gamma}^{-1}))$ because $\mathbf{e}^* < 1$. We use the Taylor expansion $\mathbb{K}(\mathbf{e}^* + \epsilon) = \mathbf{e}^* + \delta\epsilon + O(\epsilon^2)$.

$$(\mathbf{e}^* + \epsilon)\mathbf{p} = \sigma_v^2 V \mathbb{K}\left((\mathbf{e}^* + \underline{\epsilon})(1 + \Theta(\boldsymbol{\gamma}^{-1}))\right) + \sigma_a^2 + (\mathbf{e}^* + \underline{\epsilon})\mathbf{p}$$

$$= \sigma_v^2 V(\mathbf{e}^* + \delta(\underline{\epsilon} + \Theta(\boldsymbol{\gamma}^{-1})) + O(\underline{\epsilon}^2)) + \sigma_a^2 + (\mathbf{e}^* + \underline{\epsilon})\mathbf{p}$$

$$\epsilon\mathbf{p} = \sigma_v^2 V(\delta(\underline{\epsilon} + \Theta(\boldsymbol{\gamma}^{-1})) + O(\underline{\epsilon}^2)) + \underline{\epsilon}\mathbf{p}$$

$$\epsilon = \underline{\epsilon}(1 - \frac{\sigma_a^2 + (1 - \delta + O(\underline{\epsilon}))\sigma_v^2 V}{\mathbf{p}}) + \Theta(V\boldsymbol{\gamma}^{-1}\mathbf{p}^{-1})$$

$\square$

**Theorem B.18.** *Suppose $\phi = \psi_\alpha$ for $0 < \alpha < 1$ in an FRN. Then $\mathbf{e}$ converges to the unique nonunit fixed point $\mathbf{e}^*$ of $\mathbb{J}_\alpha$, and $|\mathbf{e}^* - \mathbf{e}^{(l)}|$ is $\check{\Theta}(l^{-\mu})$, where $\mu = (1 - \dot{\mathbb{J}}_\alpha(\mathbf{e}^*))/(1 - \alpha)$. Additionally, $\mathbf{s}^{(l)} = \Theta(\mathbf{p}^{(l)}) = \Theta(l^{1/(1-\alpha)})$.*

*Proof.* We apply Lemma C.36. We first check the conditions of the lemma, with $\mathbb{K} = \mathbb{J}_\alpha$. The following conditions were already verified.

- $\mathbb{J}_\alpha$ has a fixed point $\mathbf{e}^*$ less than but very close to 1, where its slope is $\upsilon := \dot{\mathbb{J}}_\alpha(\mathbf{e}^*) < 1$. (Thm C.35)

- $\mathbb{J}_\alpha(d) > d$ for all $d < \mathbf{e}^*$ and $\mathbb{J}_\alpha(d) < d$ for all $d > \mathbf{e}^*$. (By the convexity shown in Lemma C.30)

- $\mathbb{J}_\alpha$ is nondecreasing (Lemma C.30). Furthermore, from its integral formula (Eq. ($\triangle$)), we see easily that $\mathbb{J}_\alpha$ is smooth at $\mathbf{e}^* < 1$.

We also proved the following

- $\mathbf{p}^{(l)} \sim [\sigma_v^2 \sigma_w^{2\alpha} \mathsf{c}_\alpha(1-\alpha)]^{\frac{1}{1-\alpha}} l^{\frac{1}{1-\alpha}}$ (Thm C.29) and $\boldsymbol{\gamma}^{(l)}$ is asymptotically a constant fraction of $\mathbf{p}^{(l)}$ (Lemma C.36), so both go to $\infty$.

- $\mathrm{V}\psi_\alpha(\mathbf{q}) = \mathsf{c}_\alpha \mathbf{q}^\alpha = \mathsf{c}_\alpha(\sigma_w^2 \mathbf{p} + \sigma_b^2)^\alpha = \Theta(l^{\alpha/(1-\alpha)})$, so goes to $\infty$. (Lemma B.15)

Thus, for $\upsilon = \dot{\mathbb{J}}(\mathbf{e}^*)$,

$$\frac{\sigma_a^2 + (1 - \upsilon + O(\underline{\epsilon}))\sigma_v^2 \mathrm{V}\phi(\mathbf{q})}{\mathbf{p}} \sim \frac{(1 - \upsilon)\sigma_v^2 \sigma_w^{2\alpha} \mathsf{c}_\alpha}{\mathbf{p}^{1-\alpha}}$$

$$= l^{-1}(1 - \upsilon)/(1 - \alpha).$$

Now, $\mathrm{V}\phi(\mathbf{q})\boldsymbol{\gamma}^{-1}\mathbf{p}^{-1} = \Theta(l^{-\frac{1}{1-\alpha}-1})$. By using the dynamics of Lemma C.11 to upper and lower bound our dynamics, we have $\epsilon^{(l)} = \Omega(l^{-\mu-\epsilon}), O(l^{-\mu+\epsilon})$ for any $\epsilon > 0$, where $\mu = \min((1-v)/(1-\alpha), 1/(1-\alpha)) = (1-v)/(1-\alpha)$.

$\square$

### C.7.2 Backward Dynamics

**Lemma C.37.** *Suppose random variable $X \sim \mathcal{N}(0, \sigma^2)$, and $Y = \psi_{-\beta}(X)$ for some $\beta > 0$, where $\psi_\alpha$ is $\alpha$-ReLU. Then for $\xi > 0$, $Y$ has density*

$$\Pr[Y \in [\xi, \xi + \mathrm{d}\xi]] = \frac{1}{\beta\sqrt{2\pi\sigma^2}}\xi^{-\frac{1}{\beta}-1}e^{-\xi^{-2/\beta}/2\sigma^2}.$$

*At $\xi = 0$, $Y$ has density given by a Dirac delta of mass $\frac{1}{2}$.*

*Furthermore, $Y$ has finite second moment iff $\beta < \frac{1}{2}$.*

*Proof.* We have

$$\Pr[Y \in [\xi, \infty]] = \Pr[X \in [0, \xi^{-1/\beta}]]$$

$$= \frac{1}{\sqrt{2\pi\sigma^2}}\int_0^{\xi^{-1/\beta}} e^{-x^2/2\sigma^2}\,\mathrm{d}x.$$

Differentiating the RHS against $\xi$ using Leibniz's rule, we get

$$d\Pr[Y \in [\xi, \infty)]/d\xi = \frac{1}{\sqrt{2\pi\sigma^2}}e^{-\xi^{-2/\beta}/2\sigma^2}\frac{d}{d\xi}\xi^{-1/\beta}$$

$$= \frac{-1}{\beta\sqrt{2\pi\sigma^2}}\xi^{-\frac{1}{\beta}-1}e^{-\xi^{-2/\beta}/2\sigma^2}.$$

Negating both sides gives the density $f_Y$ of $Y$ for $\xi > 0$. For $\xi = 0$, observe that $\lim_{\xi \to 0} f_Y(\xi) = 0$ because, while $\xi^{-\frac{1}{\beta}-1}$ blows up polynomially, $e^{-\xi^{-2/\beta}/2\sigma^2}$ blows up exponentially. Thus the contribution of $Y$'s mass at $Y = 0$ from $X > 0$ is 0. On the other hand, all $X < 0$ gets mapped to $Y = 0$, so $f_Y(0) = \frac{1}{2}\delta_0$, where $\delta_0$ is the Dirac delta.

For the second assertion, observe that

$$f_Y(\xi) \sim \frac{1}{\beta\sqrt{2\pi\sigma^2}}\xi^{-\frac{1}{\beta}-1}\text{as } \xi \to \infty.$$

Thus, $\xi^2 f_Y(\xi)$ is integrable iff $2 - \frac{1}{\beta} - 1 < -1 \iff \beta < \frac{1}{2}$. $\square$

**Theorem B.19.** *Suppose we have the nonlinearity $\psi_\alpha$ in an FRN. $\mathrm{Var}(\dot\psi_\alpha(\zeta)^2)$ diverges for any Gaussian variable $\zeta$ with mean 0 if $\alpha \le \frac{3}{4}$ but is finite if $\alpha > \frac{3}{4}$.*

*Proof.* Note that $\dot\psi_\alpha \propto \psi_{\alpha-1}$, so it suffices to show that $\mathrm{Var}(\psi_{\alpha-1}(\zeta)^2) = \mathrm{Var}(\psi_{2\alpha-2}(\zeta))$ is infinite for $\zeta \sim \mathcal{N}(0, \sigma^2)$. By Lemma C.37 with $\beta = 2 - 2\alpha$, $\psi_{2\alpha-2}(\zeta)$ has finite variance iff $\beta < \frac{1}{2} \iff \alpha > \frac{3}{4}$. $\square$

**Theorem B.20.** *Suppose we have the nonlinearity $\psi_\alpha$ in an FRN. If $\alpha = 1$, then $\boldsymbol{\chi}^{(l-m)} = \boldsymbol{\chi}^{(l)}\left(\frac{1}{2}\sigma_v^2\sigma_w^2 + 1\right)^m$. If $\alpha \in (\frac{3}{4}, 1)$, then $\boldsymbol{\chi}^{(l-m)} = \Theta(1)\boldsymbol{\chi}^{(l)}(l/(l-m))^R$ for $R = \frac{\alpha^2}{(1-\alpha)(2\alpha-1)}$, where the constants in $\Theta(1)$ do not depend on $l$ or $m$.*

*Proof.* If $\alpha = 1$, then

$$\underline{\boldsymbol{\chi}} = \boldsymbol{\chi}(1 + \frac{1}{2}\sigma_v^2\sigma_w^2).$$

So $\boldsymbol{\chi}^{(l-m)}/\boldsymbol{\chi}^{(l)} = \Theta(1)B^m$ for $B = 1 + \frac{1}{2}\sigma_v^2\sigma_w^2$.

If $\frac{1}{2} < \alpha < 1$, then $\underline{\chi}/\chi - 1$ is

$$
\begin{aligned}
&\sigma_v^2 \sigma_w^2 \mathrm{V}\dot{\phi}(\mathbf{q}) \\
&= \sigma_v^2 \sigma_w^2 \alpha^2 \mathsf{c}_{\alpha-1} \mathbf{q}^{\alpha-1} \\
&= \sigma_v^2 \sigma_w^2 \alpha^2 \mathsf{c}_{\alpha-1} (\sigma_w^2 \mathbf{p})^{\alpha-1} + \Theta(\mathbf{p}^{\alpha-2}) \\
&= \sigma_v^2 \sigma_w^{2\alpha} \alpha^2 \mathsf{c}_{\alpha-1} (K_1 l^{\frac{1}{1-\alpha}} - K_2 l^{\frac{\alpha}{1-\alpha}} \log l + o(l^{\frac{\alpha}{1-\alpha}} \log l))^{\alpha-1} + \Theta(l^{\frac{\alpha-2}{1-\alpha}}) \qquad \text{by Thm C.29} \\
&= \sigma_v^2 \sigma_w^{2\alpha} \alpha^2 \mathsf{c}_{\alpha-1} [K_1^{\alpha-1} l^{-1} + \Theta(l^{-2} \log l)] + O(l^{-3}) \\
&= \sigma_v^2 \sigma_w^{2\alpha} \alpha^2 \mathsf{c}_{\alpha-1} K_1^{\alpha-1} l^{-1} + \Theta(l^{-2} \log l) \\
&= R l^{-1} + \Theta(l^{-2} \log l)
\end{aligned}
$$

where $R = \sigma_v^2 \sigma_w^{2\alpha} \alpha^2 \mathsf{c}_{\alpha-1} K_1^{\alpha-1} = \frac{\alpha^2}{(1-\alpha)(2\alpha-1)}$ and $K_1 = [\sigma_v^2 \sigma_w^{2\alpha} \mathsf{c}_\alpha (1-\alpha)]^{\frac{1}{1-\alpha}}$. So

$$\underline{\chi} = \chi \exp(R l^{-1} + \Theta(l^{-2} \log l))$$

$$\boldsymbol{\chi}^{(l-m)} = \Theta(1) \boldsymbol{\chi}^{(l)} \left( \frac{l}{l-m} \right)^R$$

as desired. $\qquad\square$

**Theorem B.21.** *If $\phi = \psi_1$ in an FRN, then for $l \ge m \ge 0$,*

$$
\begin{aligned}
\boldsymbol{\chi}_b^{(l-m)} &= \Theta(1) \boldsymbol{\chi}^{(l)} B^m, & \boldsymbol{\chi}_w^{(l-m)} &= \Theta(1) \boldsymbol{\chi}^{(l)} B^l, \\
\boldsymbol{\chi}_v^{(l-m)} &= \Theta(1) \boldsymbol{\chi}^{(l)} B^l, & \boldsymbol{\chi}_a^{(l-m)} &= \Theta(1) \boldsymbol{\chi}^{(l)} B^m.
\end{aligned}
$$

*where $B = 1 + \sigma_v^2 \sigma_w^2 / 2$.*

*If $\phi = \psi_\alpha$ in an FRN, for $\alpha < 1$, then for $l \ge m \ge 0$,*

$$
\begin{aligned}
\boldsymbol{\chi}_b^{(l-m)} &= \Theta(1) \boldsymbol{\chi}^{(l)} l^R (l-m)^{-R-1}, & \boldsymbol{\chi}_w^{(l-m)} &= \Theta(1) \boldsymbol{\chi}^{(l)} l^R (l-m)^{\frac{\alpha}{1-\alpha}-R}, \\
\boldsymbol{\chi}_v^{(l-m)} &= \Theta(1) \boldsymbol{\chi}^{(l)} l^R (l-m)^{\frac{\alpha}{1-\alpha}-R}, & \boldsymbol{\chi}_a^{(l-m)} &= \Theta(1) \boldsymbol{\chi}^{(l)} (l/(l-m))^R.
\end{aligned}
$$

*Proof.* The proof is similar to that of Thm B.7. $\qquad\square$