[Reviews · NeurIPS 2017]

Reviewer 1



This paper analytically investigates the properties of Resnets with random weights using a mean field approximation. The approach used is an extension of previous analysis of feed forward neural networks. The authors show that in contrast to feed forward networks Resnets do exhibit a sub-exponential behavior (polynomial or logarithmic) when inputs are propagated forward or gradients are propagated backwards through the layers. The results are very interesting because they give an analytic justification and intuition of why Resnets with a large number of layers can be trained reliably. The paper also extends the mean field technique for studying neural network properties, which is of value for analyzing other architectures. The presentation of the results is overall clear and while the paper is well written. Overall this is an interesting paper with convincing results.